# Cortical feedback loops bind distributed representations of working memory

Ivan Voitov[1,2 ✉] & Thomas D. Mrsic-Flogel[1]

Working memory—the brain's ability to internalize information and use it flexibly to guide behaviour—is an essential component of cognition. Although activity related to working memory has been observed in several brain regions[1-3], how neural populations actually represent working memory[4-7] and the mechanisms by which this activity is maintained[8-12] remain unclear[13-15]. Here we describe the neural implementation of visual working memory in mice alternating between a delayed non-match-to-sample task and a simple discrimination task that does not require working memory but has identical stimulus, movement and reward statistics. Transient optogenetic inactivations revealed that distributed areas of the neocortex were required selectively for the maintenance of working memory. Population activity in visual area AM and premotor area M2 during the delay period was dominated by orderly low-dimensional dynamics[16,17] that were, however, independent of working memory. Instead, working memory representations were embedded in high-dimensional population activity, present in both cortical areas, persisted throughout the inter-stimulus delay period, and predicted behavioural responses during the working memory task. To test whether the distributed nature of working memory was dependent on reciprocal interactions between cortical regions[18-20], we silenced one cortical area (AM or M2) while recording the feedback it received from the other. Transient inactivation of either area led to the selective disruption of inter-areal communication of working memory. Therefore, reciprocally interconnected cortical areas maintain bound high-dimensional representations of working memory.

## A task to isolate visual working memory

Cognition necessitates the construction of internal representations of the sensory world[21-24]. Working memory isolates internal representations from their sensory causes and enables their flexible coupling to motor output. The maintenance of sensory information in working memory has been linked to persistent sensory activity[5,6,25] and low-dimensional population dynamics[26,27], often modelled as attractor regimes within dynamical systems descriptions of the observed neural activity[8,28-30]. Recent studies, however, have implicated sparse bursts of activity or 'activity-silent' mechanisms for the maintenance of sensory information stored in working memory[7,10,31,32]. A key challenge in disambiguating the neural representations that underlie working memory is the presence of behavioural variables that are independent of the maintenance of sensory information, but are nevertheless present during decision-making tasks and recruit their own neural processes, such as timing, reward expectation and motor preparation. To address this issue, we designed a task-switching paradigm in which mice had to alternate, in blocks of several hundred trials, between performing a visual working memory (WM) task and a working memory-independent Discrimination task with matching delay, stimulus, and reward statistics (Fig. 1a and Extended Data Fig. 1a). The WM task was a modified Go/No-go delayed non-match-to-sample task that required mice to

infer the reward contingency of a visual stimulus from the previously seen stimulus, separated in time by a grey-screen delay period (delay). Sequences of delay and stimulus epochs were presented continuously (that is, no inter-trial intervals), with individual trials composed of a delay and stimulus pair, such that each trial's stimulus served as a cue to the subsequent stimulus. In the WM task, a stimulus was rewarded (target) only if it was preceded by a stimulus of the mirrored orientation (cue; Fig. 1a). Cues and targets were gratings oriented at ±45°. By contrast, the identities of the rewarded stimuli (targets) during the Discrimination task were kept constant (±45°) and were independent of the preceding stimuli (0° oriented gratings). Mice had to respond to the rewarded stimulus by licking a spout, which delivered a liquid food reward (see Methods). In both tasks, the probability of the targets was 10%, and the inter-stimulus delay period was sampled from an exponential distribution (that is, approximating a flat hazard rate) ranging from 0.8 to 4 seconds. To gauge for any uncontrolled differences between the two tasks, such as lapse rate or arousal, we introduced a common unrewarded probe stimulus to both tasks, with the same presentation probability as the targets (10%).

The key behavioural difference observed between the two tasks was that correct responses in individual trials depended on the duration of the preceding inter-stimulus delay period only when

[1]Sainsbury Wellcome Centre, University College London, London, UK. [2]Biozentrum, University of Basel, Basel, Switzerland. ✉e-mail: i.voitov@ucl.ac.uk

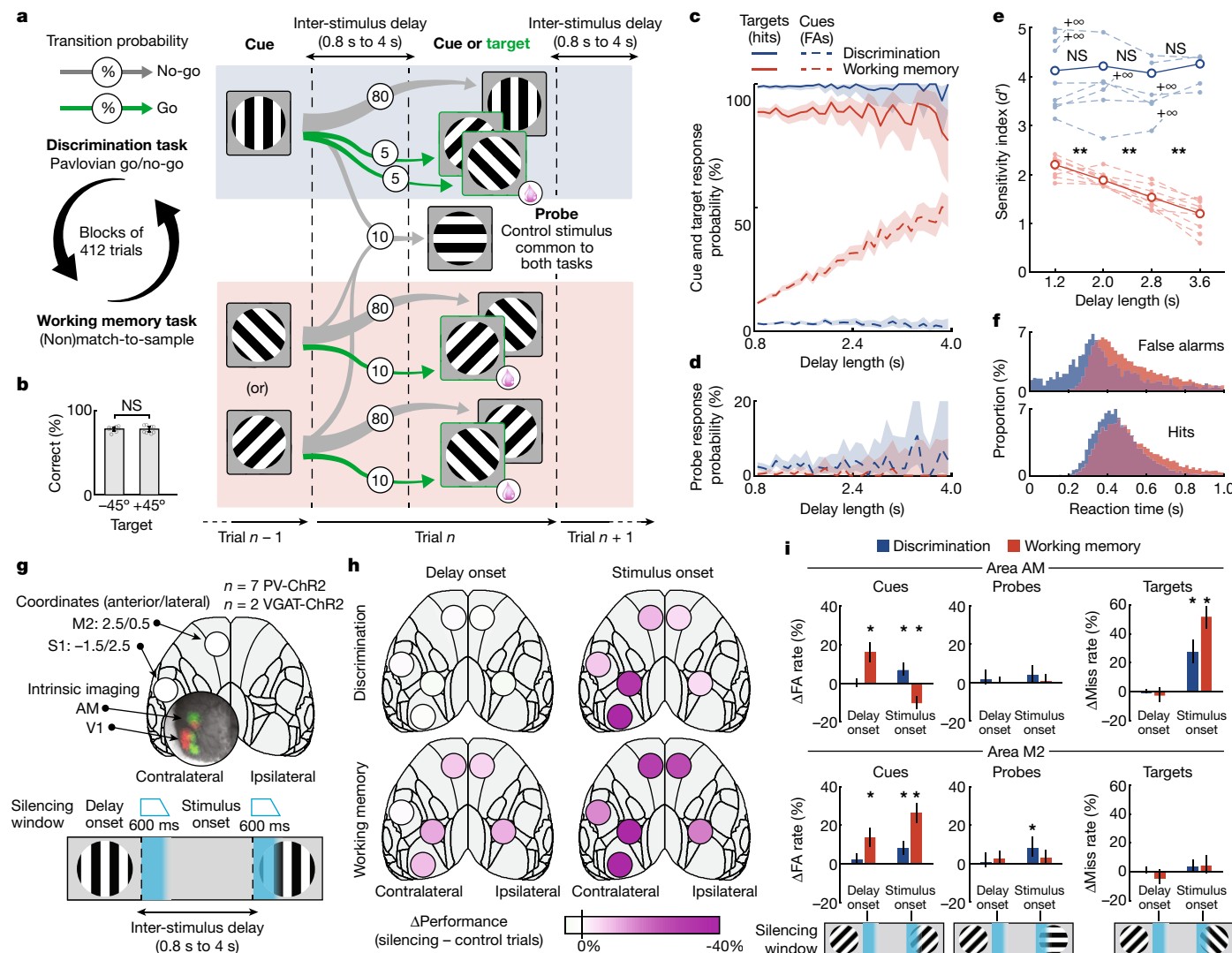

**Fig. 1 | Visual working memory is maintained by distributed neocortical regions. a**, Schematic of the task structure and transition probabilities between trial types (cues, probes and targets). Each trial was composed of a grey-screen delay period followed by a variably oriented grating stimulus. In the WM task, a stimulus was rewarded (target) only if it was preceded by a stimulus of the mirrored orientation (cue). In the Discrimination task, both +45° and −45° oriented stimuli were always rewarded (targets). **b**, Percentage of correct responses in −45° target and +45° target trials during the WM task (means are across mice; error bars are 95% CI; P = 0.65, n = 9, two-sided signed-rank test). **c**, Mean response probabilities, split by trial type, as a function of the preceding delay period length, binned at 100 ms (pooled from 9 mice; n = 150,381 trials; shaded regions are 95% CI), for the Discrimination (blue) and WM (red) tasks. FA, false alarm. **d**, Responses to probe stimuli, as in **c**. **e**, Performance of individual mice as measured by d′ (see Methods), split into delay length quartiles (dashed lines; +∞, results from quartiles containing no miss trials). Thick lines represent trials pooled from all mice. Statistical tests between adjacent quartiles (n = 9 mice; P = 0.96, P = 0.23 and P = 0.98 for the Discrimination task, and P = 9.77 × 10⁻³, P = 1.95 × 10⁻³ and P = 3.91 × 10⁻³ for the WM task, two-sided signed-rank test). *P < 0.05, **P < 0.01; NS, not significant. **f**, Normalized reaction times to cues (top) and targets (bottom), binned at 16.67 ms. Trials pooled from all 9 mice. WM task

engagement delayed peak reaction times to the cues by 163 ms and to the targets by 70 ms (P = 3.91 × 10⁻³ and P = 1.17 × 10⁻², respectively, n = 9 mice, two-sided signed-rank test; see Extended Data Fig. 2c–f). **g**, Schematic of the cortical areas targeted for the optogenetic silencing (top), and optogenetic silencing protocol (bottom). Areas V1 and AM were identified by intrinsic signal imaging (see Methods). Areas M2 and S1 were identified by coordinates (millimetres anterior/lateral of bregma). The optogenetic silencing light was flashed for 400 ms, followed by a linear ramp down over 200 ms, at the onset of the delay or stimulus. **h**, Overview of silencing effects on performance, split by silencing onset (delay, left; stimulus, right), task (Discrimination, top; WM, bottom) and area (shaded circles). Performance was defined as [100% − FA rate (%) − miss rate (%)]. Shading represents the differences in performance between control and silencing trials. **i**, Effect of optogenetic silencing on responses to cues, probes and targets, for area AM (top) and area M2 (bottom), in either task (Discrimination, blue; WM, red), and two silencing onsets (labelled). Individual bars represent the difference of FA or miss rates between silencing and control trials (pooled from 9 mice, n = 173,432 trials). Error bars represent 95% CI of silencing trials. Statistically significant silencing effects (α = 0.05) are labelled (two-sided Fisher's exact test). See Extended Data Fig. 4 for analyses of all areas.

the mice were engaged in the WM task. Specifically, false alarms (responses to the cue stimuli) during the WM task increased after longer delay lengths (Fig. 1c and Extended Data Fig. 2a,b; average slope 11.9% per second, n = 9 mice, all P < 1 × 10⁻³). Consequently, the performance of the mice, measured by d′ (see Methods), decreased

with increasing delay lengths (Fig. 1e), consistent with visual working memory duration in humans[33]. Crucially, this effect was specific to the maintenance of sensory information in memory, as responses to the probe stimuli during the WM task, despite having the same reward contingency as the cues, did not depend on the preceding

delay length (Fig. 1d and Extended Data Fig. 2b; $n = 9$ mice, all $P > 0.05$).

Consistent with the engagement of working memory leading to longer reaction times in humans[34], stimulus response latencies were longer when mice engaged in the WM task (Fig. 1f and Extended Data Fig. 2c–f). This was not linked to the response probabilities, as reaction times were longer for both false alarms to cues (higher in the WM task; $n = 9$ mice, average 163 ms lag, $P = 3.91 \times 10^{-3}$) and hits to targets (lower in the WM task; $n = 9$ mice, average 70 ms lag, $P = 1.17 \times 10^{-2}$).

Measures of movement and arousal during the inter-stimulus delay period, including running speed, halting rate and pupil diameter[35,36], were stable and not different between the two tasks (Extended Data Fig. 3a–d; $n = 9$ mice, all $P > 0.05$). Early responses during the delay were rare (3.2% of trials), but slightly more frequent during the Discrimination task (Extended Data Fig. 3b; $n = 9$ mice, $P = 3.91 \times 10^{-3}$). Thus, by developing a task to isolate working memory in mice, we were able to identify two psychometric features of working memory engagement: the disruption of internally maintained sensory information by longer delay durations, and an increase in the reaction times when engaging in the WM task.

## Working memory is maintained by distributed areas

The role of different brain regions in maintaining memory-related task variables has previously been examined by optogenetic inactivation experiments[37–42]. To assess which cortical areas were required for working memory and to distinguish between their sensory, mnemonic and motor functions, we transiently inactivated one of six different cortical areas during one of two epochs per trial by focal optogenetic stimulation of inhibitory neurons (see Methods). The silencing window (400 ms plus a 200 ms ramp down) was timed to either the onset of the inter-stimulus delay or the onset of the stimulus, chosen at random in 8% of all trials. We tested six areas of the dorsal neocortex: visual area AM, premotor area M2, primary somatosensory area S1 and primary visual area V1 contralateral to the visual stimulus, and areas AM and M2 ipsilateral to the visual stimulus (Fig. 1g).

At the onset of the delay, the inactivation of any area other than S1 increased incorrect responses during the WM task, but had no significant effect on responses during the Discrimination task (Fig. 1h). Furthermore, delay-onset inactivation only affected the responses that were dependent on the delay length (that is, false alarms to the cues in the WM task; Fig. 1i), and did not affect responses to the memory-independent probe stimuli, even in the WM task (Fig. 1i and Extended Data Fig. 4). Therefore, transient cortical inactivation during the delay did not compromise the ability of the mice to follow the task structure, and indicates that the maintenance of sensory information in working memory depends on activity in distributed areas of the neocortex.

By contrast, inactivating the neocortex during the stimulus presentation had effects that were dissociated by cortical area and task (Fig. 1i). Inactivation of the contralateral visual cortex (area AM or V1) at the onset of the stimulus led to (1) an increased miss rate to the target stimuli, more-so during the WM task; (2) an increased false alarm rate to cues during the Discrimination task; and (3) a paradoxical decrease in the false alarm rate to the cues during the WM task, revealing a divergence in the function of posterior cortical areas when engaged in working memory-guided sensory processing. On the other hand, inactivating anterior and ipsilateral cortical areas (contralateral M2 and S1, and ipsilateral AM and M2) at the onset of the stimulus had no effect on the miss rates in either task, but instead led to an increase in the false alarm rate to the cues, with a stronger effect when the mice were engaged in the WM task (Fig. 1i and Extended Data Fig. 4).

The effect of cortical inactivation on behavioural performance could not be explained by changes in measures of movement or arousal, as there was no interaction between the effects on running speed or pupil diameter and the task that the mouse was performing (Extended Data Fig. 3e–h). A bright masking light, when presented on control trials, with the same dynamics as the optogenetic inactivation light, had no effect on mouse running or stimulus responses (see Methods). An overview of all optogenetic effects on task performance is presented in Extended Data Fig. 4. Together, these inactivation experiments indicate that distributed areas of the dorsal neocortex maintain sensory information in working memory during the delay period, and become dissociated in their contribution to action generation by working memory during sensory processing.

## Working memory-agnostic neural dynamics

Previous studies of memory-guided decision making have reported persistent[4–6,17,25,43] or sequential[16,44,45] neuronal activity during the delay periods. We used two-photon calcium imaging (see Methods) to examine the neural activity recruited by working memory in two areas that are required for the maintenance of working memory information during the delay period in the WM task: higher visual area AM and premotor area M2, contralateral to the visual stimulus. As the orientations of the cue stimuli in the Discrimination and WM tasks were different, we rotated the orientations of all task stimuli +30° and −30° multiple times per session in blocks of several hundred trials, out of phase with the task blocks, to compare the neural activity elicited by common sensory input across the two tasks (see Extended Data Fig. 1b and Methods for details). Such stimulus rotations had no effect on behavioural performance (Extended Data Fig. 1c). For all subsequent comparisons of neural activity between the Discrimination and WM tasks, we limited our analyses to delay- and stimulus-evoked responses following identical sensory input.

Single cells in both AM and M2 were at least as likely to be responsive during the inter-stimulus delay periods as during the presentation of the stimuli (see Methods for selection criteria). The response profiles of individual cells were notably similar across the two tasks (Fig. 2a,c), and the cell-averaged delay period activity in either area did not significantly differ between the two tasks (Fig. 2b,d; $P = 0.67$, $n = 805$ cells). Neural populations that were responsive during the inter-stimulus delay periods exhibited sequential activation patterns similar to those previously reported in short-term memory tasks[16,44,45], whereby the population activity tiled the entire length of the delay (Fig. 2e,h). However, working memory engagement did not alter the temporal profiles of the trial-averaged activity of individual cells during the delay period (as compared to chance; Extended Data Fig. 5a,b; $n = 805$ cells, $P = 0.43$).

To identify representations specific to the WM task in the full recorded neural population, we performed dimensionality reduction on the trial-averaged responses of all active cells pooled across experiments ($n = 5,589$ cells from 16 area AM experiments, and $n = 4,023$ cells from 11 area M2 experiments, from 7 mice). Consistent with previous reports that neural population dynamics are constrained to low-dimensional modes when animals perform short-term memory tasks[46,47], most of the trial-averaged activity variance was captured within just the first three principal components (PCs; 83% in AM and 78% in M2; Fig. 2f,i; see Methods). By projecting the population activity during the Discrimination and WM tasks separately onto the first three PCs, we observed no significant difference in the Euclidean distance between the resulting activity trajectories at any point during the delay or the stimulus epochs, in either area AM or area M2, compared to a null distribution of distances obtained by shuffling the task labels (Fig. 2g,j; $n = 16$ area AM experiments and $n = 11$ area M2 experiments, all time points $P > \alpha$, two-sided $t$-test, Bonferroni-corrected $\alpha = 2.08 \times 10^{-3}$). Furthermore, the eigenspectrum of the population activity covariance did not differ significantly between the tasks (Extended Data Fig. 5c–n). Therefore, low-dimensional neural dynamics reflected processes common to both the WM and Discrimination tasks, such as timing, reward expectation or motor preparation. Nevertheless, our results

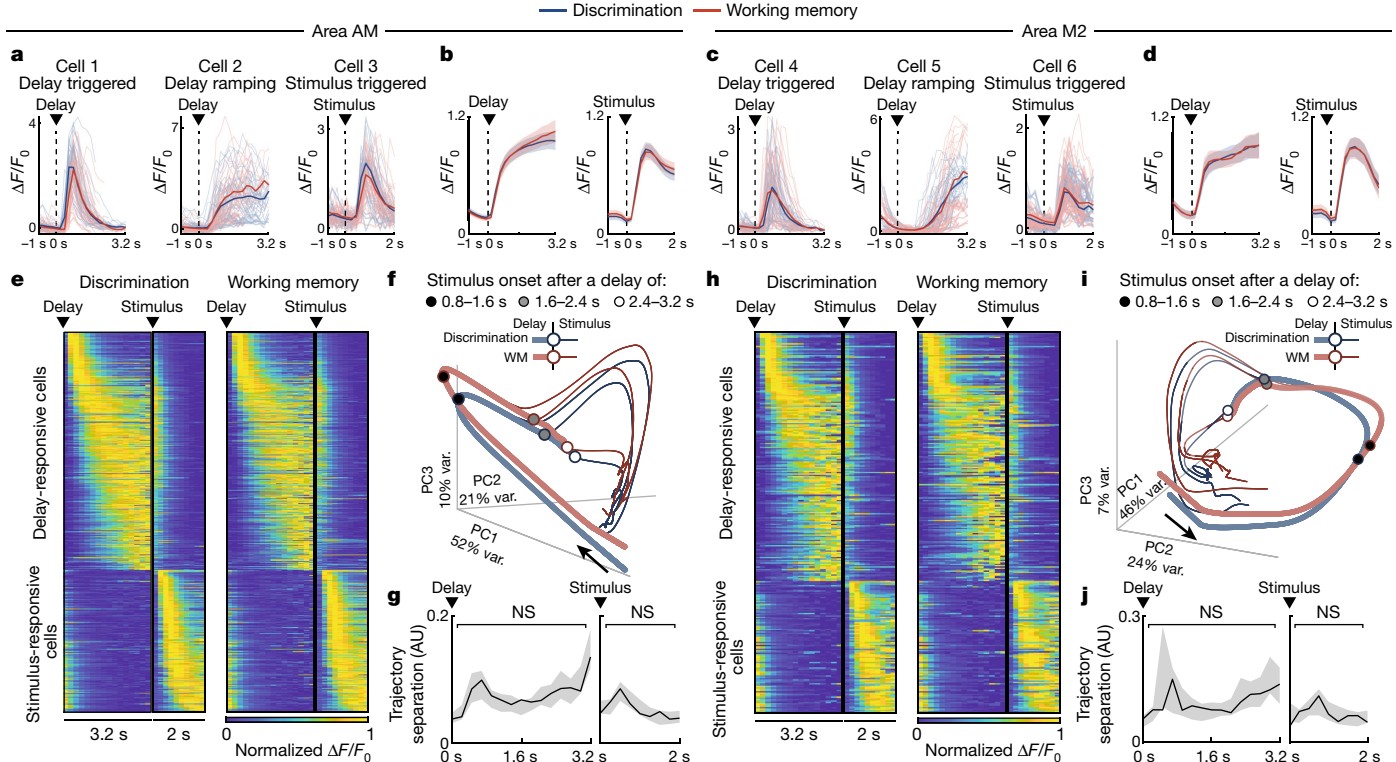

**Fig. 2 | Low-dimensional neural dynamics are independent of working memory. a**, Example single-cell responses in area AM. Saturated lines are mean responses during the Discrimination (blue) or WM (red) tasks. Dim lines are responses from 40 trials with the longest delay durations. The first two cells have their responses aligned to the delay onset, and the third cell to the stimulus. **b**, Cell- and trial-averaged responses ($\Delta F/F_0$) of the delay-responsive cells (left) and stimulus-responsive cells (right) in area AM (see Methods for selection criteria) to the delay and stimulus onsets, respectively, split by task. Cells pooled from all experiments ($n = 16$ from 7 mice). Shaded regions are 95% CI. **c,d**, As in **a,b**, for cells recorded in area M2 ($n = 11$ experiments from 7 mice). **e**, Single-cell responses for all delay-responsive and stimulus-responsive cells (individual rows) recorded in area AM ($n = 16$ experiments from 7 mice), averaged from Discrimination task (left) and WM task (right) trials, sorted by response latency (see Methods). Responses were normalized for each cell. **f**, Trial-averaged projections of area AM population activity ($n = 5,589$ cells pooled from 16 experiments) onto its first three PCs (see Methods).

The proportion of variance (var.) explained is shown along each axis. Projections were split by task (colour) and into three groups of delay duration (0.8–1.6 s, 1.6–2.4 s and 2.4–3.2 s), such that activity was plotted from trials with delays that were at least as long as the respective stimulus onset time. Arrows represent the direction of time, circles represent the minimum stimulus onset of each delay duration group. Unsaturated lines represent activity preceding stimulus onset; saturated lines represent activity after stimulus onset. **g**, The Euclidean distance between Discrimination and WM task trajectories in **f**, averaged across experiments, is plotted over the course of the delay and stimulus (all time points $P > \alpha$, Bonferroni-corrected $\alpha = 2.08 \times 10^{-3}$, two-sided $t$-test). Shaded regions are 95% CI of a trial-shuffled null distribution of task trajectory distances. AU, arbitrary units. **h–j**, As in **e–g**, but for area M2. **h**, Delay- and stimulus-responsive cells pooled across $n = 11$ experiments from 7 mice. **i,j**, PCs calculated from $n = 4,023$ pooled active cells. **j**, All time points $P > \alpha$, Bonferroni-corrected $\alpha = 2.08 \times 10^{-3}$, two-sided $t$-test. NS, not significant.

can be seen as consistent with reports of persistent delay activity in short-term memory tasks that did not necessitate the maintenance of internal sensory representations, such as delayed reaching[43] and oculomotor delay in non-human primates[4,48], or T-maze alternation in rodents[44]. Notwithstanding, the removal of activity in areas AM and M2 by optogenetic inhibition affected behaviour in a manner that was highly specific to working memory maintenance, suggesting an alternative representational scheme.

## High-dimensional embedding of working memory

Neural activity supporting sensory representations in working memory can be sparse in time and variable across trials[7,10,49,50]. We first confirmed that the distribution of single-cell differences in delay activity between the two tasks was greater than would be expected by chance, and could not be explained by confounding slow temporal factors correlated with the task blocks (for example, fluorophore bleaching or changes in motivational state; Extended Data Fig. 6). Although this analysis did not reveal distinct subpopulations of cells whose delay activity was selective to either task, we hypothesized that sparse and uncorrelated working

memory representations could still be linearly separable in the full neural activity space even in the absence of informative low-dimensional dynamics.

We used linear discriminant analysis (LDA) to identify two dimensions that capture working memory representations in the full neural activity space: a task coding dimension ($CD_{TASK}$), which captures the activity introduced by working memory engagement by contrasting the delay activity in the two tasks after identical sensory input (Fig. 3a); and a cue coding dimension ($CD_{CUE}$), which captures the particular stimulus information maintained during the delay by contrasting the two possible cue stimuli that precede the delay activity during the WM task (Fig. 3a; see Methods). By projecting the delay activity of all active cells ($310 \pm 41$ cells per experiment) onto $CD_{TASK}$, we were able to decode the task (Discrimination or WM) that the mouse was engaged in with high accuracy (average cross-validated test accuracy of 91%; Fig. 3b). Likewise, projections of delay activity onto $CD_{CUE}$ successfully separated trials on the basis of the preceding cue (average cross-validated test accuracy of 84%; Fig. 3b). These results were similar across neural populations in areas AM and M2 (Extended Data Fig. 7), albeit with a higher decoding performance in area AM.

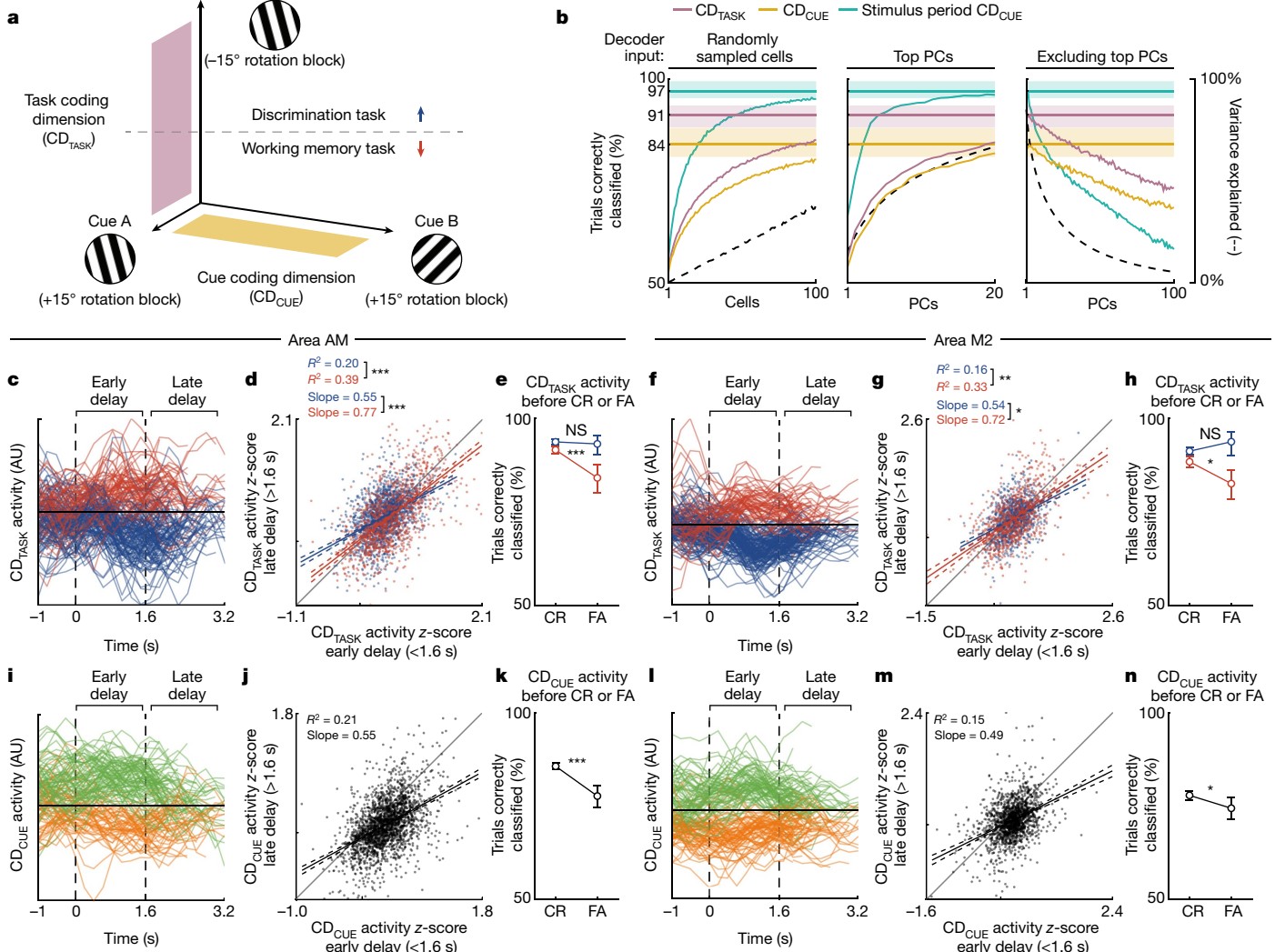

**Fig. 3 | High-dimensional embedding of working memory representations.**
**a**, Schematic of two coding dimensions ($CD_{TASK}$ and $CD_{CUE}$; see main text and Methods) used to identify working memory representations during the delay. $CD_{TASK}$ was identified from opposing rotation blocks for the two tasks, such that the preceding stimulus orientations were identical. **b**, Assessment of the embedding dimensionality of $CD_{TASK}$ (lilac), $CD_{CUE}$ (gold) and stimulus period $CD_{CUE}$ (see main text). Horizontal lines are mean test accuracies when decoding the task from $CD_{TASK}$ delay activity, the preceding cue from the $CD_{CUE}$ delay activity and the current cue from the stimulus period $CD_{CUE}$ stimulus-evoked activity (averages across $n = 31$ experiments; shaded regions are 95% CI; see Extended Data Fig. 7 for experiments split by area). Coloured incrementing lines are mean test accuracies when limiting the population activity available for decoding to a number of randomly sampled cells (left), the top PCs (centre) or excluding the top PCs (right). Dashed grey lines are the respective proportions of total population activity variance available for decoding. **c**, Single-trial population activity of an example experiment from area AM, aligned to the onset of the delay, projected onto $CD_{TASK}$. Only trials with sufficiently long delays (≥2 s) are shown. **d**, z-scored single-trial area AM $CD_{TASK}$ activity in the second half (>1.6 s) of the delay plotted against the first half (<1.6 s) of the delay (that is, each point is one trial). Trials were split by task (colour) and the Discrimination task projections had their signs inverted. Only

trials with sufficiently long delays (≥2 s) were included in this analysis ($n = 2,921$ trials collected from 18 experiments). Solid and dashed lines represent the fits and 95% CI of a regression model. The models' coefficients of determination ($R^2$) and slopes, and their differences between tasks, are shown ($P = 2.40 \times 10^{-5}$ and $P = 3.08 \times 10^{-4}$, respectively, difference greater than zero two-sided one-sample $t$-test). **e**, The proportion of cue trials in which $CD_{TASK}$ delay activity correctly classified the trials' task, split by correct rejection (CR) or false alarm (FA) to the subsequent cue ($P = 5.67 \times 10^{-4}$, $n = 3,577$ trials for the WM task; $P = 0.07$, $n = 5,781$ trials for the Discrimination task; two-sided Fisher's exact test). Error bars represent 95% CI. **f–h**, As in **c–e**, but for area M2 experiments. **g**, Differences between $R^2$ values and slopes ($n = 1,882$ trials collected from 13 experiments; $P = 6.05 \times 10^{-3}$ and $P = 4.64 \times 10^{-2}$, respectively, difference greater than zero two-sided one-sample $t$-test). **h**, Pre-CR and pre-FA difference ($P = 1.71 \times 10^{-2}$, $n = 2,573$ trials for WM task; $P = 0.47$, $n = 3,669$ trials for Discrimination task; two-sided Fisher's exact test). **i–k**, As in **c–e**, but for $CD_{CUE}$. **j**, $n = 2,210$ trials collected from 18 experiments. Cue A projections had their signs inverted. **k**, Pre-CR and pre-FA difference ($P = 9.56 \times 10^{-9}$, $n = 6,816$ trials, two-sided Fisher's exact test). **l–n**, As in **i–k** ($CD_{CUE}$), but for area M2 experiments. **m**, $n = 1,600$ trials collected from 13 experiments. **n**, Pre-CR and pre-FA difference ($P = 2.84 \times 10^{-2}$, $n = 5,063$ trials, two-sided Fisher's exact test). *$P < 0.05$, **$P < 0.01$, ***$P < 0.001$; NS, not significant.

To examine the embedding of these representations in the neural activity space[51], we systematically limited the population activity available for defining $CD_{TASK}$ and $CD_{CUE}$, and used the resulting dimensions to decode the task and cue identities. By decoding from sequentially increasing numbers of randomly sampled active cells (see Methods), we observed that at least 100 cells were required to

predict the task and cue identity in 85% and 80% of trials, respectively (Fig. 3b). To assess the sparsity of this population encoding and to dissociate the number of cells required for decoding from the total activity variance available, we performed a similar decoding sweep with each experiment's population activity projected onto its first PCs (calculated independently per experiment, from the population

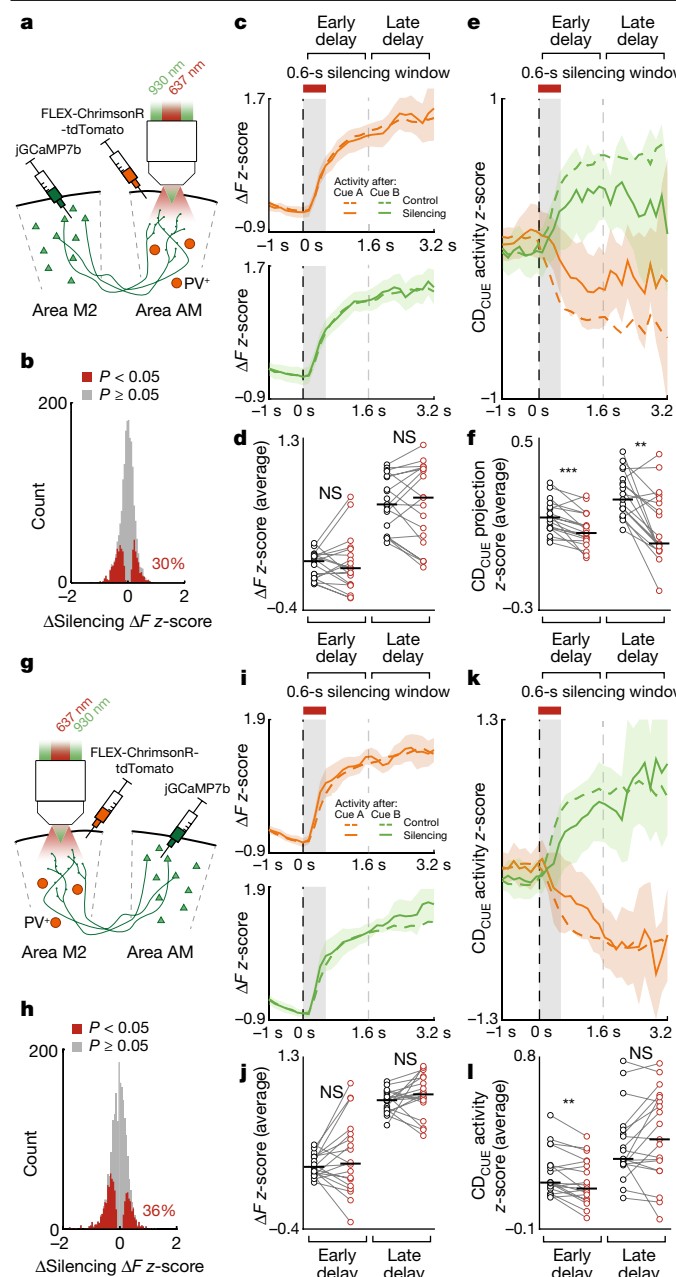

**Fig. 4 | Maintenance of working memory representations by reciprocal cortical interactions. a**, Schematic of the experiment. The activity of area M2 axons in area AM (M2 → AM) was recorded while area AM was inactivated by optogenetic stimulation of local PV⁺ cells. Silencing occurred at the onset of the delay and lasted for 600 ms (as in Fig. 1g). **b**, Effect of optogenetic silencing of area AM on the z-scored ΔF activity of single M2 → AM axons, averaged over the delay. Significantly affected axons were identified by comparison with a bootstrapped null distribution (two-sided *t*-test, α = 0.05). **c**, M2 → AM axonal ΔF responses aligned to the onset of the delay, binned at 132 ms, split by preceding cue (top and bottom) and control and silencing trials (dashed and solid lines, respectively), z-scored and averaged across experiments ($n = 19$ from 4 mice). Shaded regions are 95% CI of silencing trial responses. **d**, The responses in **c**, with the effects of optogenetic silencing, per experiment ($n = 19$), averaged separately for the early delay ($<1.6$ s; $P = 0.81$) and late delay ($\geq 1.6$ s; $P = 0.84$), two-sided signed-rank test. **e**, Activity shown as in **c**, but projected onto $CD_{CUE}$. **f**, As in **d**, for activity projected onto $CD_{CUE}$ ($n = 19$, $P = 9.67 \times 10^{-4}$ and $P = 1.12 \times 10^{-3}$, for early and late delay activity, respectively, two-sided signed-rank test). Responses following cue A were inverted. **g**, Schematic as in **a**, showing the reverse experiment, silencing area M2 while imaging the AM → M2 axons. **h–l**, as in **b–f**, but for the reverse experiments ($n = 19$ experiments from 3 mice). **j**, Early delay silencing effect ($P = 0.33$), late delay silencing effect ($P = 0.42$), two-sided signed-rank test. **l**, Early delay silencing effect ($P = 7.01 \times 10^{-3}$), late delay silencing effect ($P = 0.33$), two-sided signed-rank test. ** $P < 0.01$, *** $P < 0.001$; NS, not significant.

Single-trial dynamics of delay period activity have previously been used to inform mechanistic accounts of their persistence[48,52,53]. We projected single-trial population activity during the delay onto either the $CD_{TASK}$ ($CD_{TASK}$ activity) or the $CD_{CUE}$ ($CD_{CUE}$ activity), and found that activity in these subspaces was persistent, often spanning the full duration of the delay (Fig. 3c,f,i,l). We summarized the single-trial dynamics of this coding dimension activity by comparing it between the first and second halves of the delay, and calculating the slope and coefficient of determination ($R^2$) of this relationship. The slope of this relationship is indicative of the persistence of working memory representations over the course of the delay, and the $R^2$ is indicative of their robustness over time. Notably, these measures are agnostic of whether the cue or the task identities of the trials were correctly predicted (that is, the classification accuracy; Fig. 3b). These analyses were restricted to trials with sufficiently long ($\geq 2$ s) delay periods pooled across all experiments ($n = 2,921$ trials collected from 18 experiments in area AM, and $n = 1,882$ trials collected from 13 experiments in area M2). We found that both the $R^2$ and the slope of the $CD_{TASK}$ activity were higher when the mice were performing the WM task as compared to the Discrimination task ($P = 2.40 \times 10^{-5}$ and $P = 3.08 \times 10^{-4}$ for area AM experiments, and $P = 6.05 \times 10^{-3}$ and $P = 4.64 \times 10^{-2}$ for area M2 experiments, for $R^2$ and slope, respectively; Fig. 3d,g). This suggests that $CD_{TASK}$ activity reflects neural activity that is integral to the maintenance of working memory, and supports the role of line attractor dynamics for the maintenance of working memory representations[28,30].

A requisite for working memory representations during the delay is that they should be predictive of correct behavioural responses to the subsequent stimulus. We observed that in both areas AM and M2, a reduction of $CD_{CUE}$ activity or $CD_{TASK}$ activity during the WM task delays, as measured by the failure to correctly classify a given trial's cue or task identity, anticipated incorrect responses to the subsequent stimulus ($P = 5.67 \times 10^{-4}$ and $P = 9.56 \times 10^{-9}$ for $CD_{TASK}$ and $CD_{CUE}$ activity, respectively, from area AM experiments, and $P = 1.71 \times 10^{-2}$ and $P = 2.84 \times 10^{-2}$ for $CD_{TASK}$ and $CD_{CUE}$ activity from area M2 experiments; Fig. 3e,h,k,n). Extended Data Figure 8 shows these results in terms of the $CD_{TASK}$ and $CD_{CUE}$ activity of individual experiments. A reduction of the delay period $CD_{TASK}$ activity did not predict incorrect responses during the Discrimination task, nor were incorrect responses associated with changes in the eigenspectrum of the population activity covariance during the

activity of all trials concatenated in time; see Methods). Up to 20 of the first PCs were required to reach a similar level of decoding accuracy to that of 100 randomly sampled cells, despite explaining significantly more of the population activity variance (67% versus 39%, $n = 31$ experiments from 7 mice, $P = 1.30 \times 10^{-6}$; Fig. 3b). Furthermore, leaving out the first 100 PCs from the data available to the decoder—thus removing on average 95% of the variance in population activity—preserved a substantial decoding accuracy of the $CD_{TASK}$ (73%) and $CD_{CUE}$ (68%; Fig. 3b). By contrast, population representations of the sensory stimulus (stimulus period $CD_{CUE}$; Fig 3b), calculated similarly to the $CD_{CUE}$ but decoded from the respective stimulus-evoked responses of cues, were lower dimensional—the top 4 PCs were sufficient to predict the stimulus identify of 90% of trials, and the removal of the first 100 PCs greatly reduced the stimulus decoding accuracy (predicting only 58% of trials). Together, these results indicate that working memory representations are embedded in a high-dimensional subspace of population activity, with most cells carrying working memory information that is distributed in uncorrelated modes of activity.

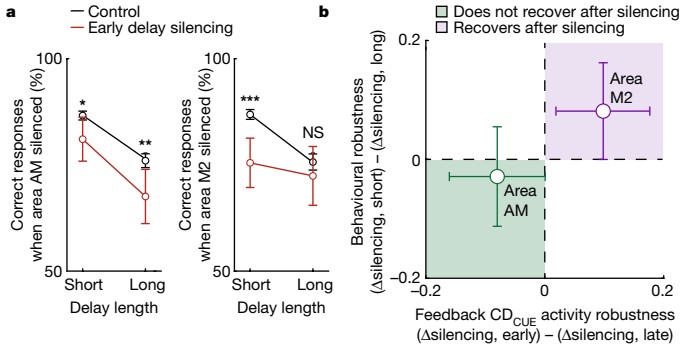

**Fig. 5 | Differential robustness of working memory to visual and premotor cortical inactivation. a**, Experiment-averaged effect of optogenetic silencing at delay onset on the proportion of correct responses to the subsequent stimuli, split by the length of the delay (<1.6 s, short; ≥1.6 s, long). Control trials in black and silencing trials in red. Axonal imaging experiments were pooled with the behaviour and optogenetics experiments (Fig. 1) to increase statistical power. Left, silencing area AM ($P = 2.89 \times 10^{-2}$ and $P = 7.22 \times 10^{-3}$, short and long delays, respectively, $n = 105$ experiments, two-sided paired-sample $t$-test). Right, silencing area M2 ($P = 1.12 \times 10^{-4}$ and $P = 0.32$, short and long delays, respectively, $n = 112$ experiments, two-sided paired-sample $t$-test). Error bars are 95% CI. **b**, Robustness of behaviour and axonal feedback $CD_{CUE}$ activity to optogenetic silencing, split by area silenced (area M2, purple; area AM, green). Robustness was measured as the difference of the silencing effect size between the early delay and late delay, per experiment. Behavioural data were collected across all experiments (as in **a**). Error bars are 95% CI along each axis.

delay (Extended Data Fig. 5c–n), which suggests that the relationship between delay period $CD_{CUE}$ and $CD_{TASK}$ activity and subsequent behavioural responses was not explained by task-agnostic premotor activity. Notably, when the delay activity failed to predict the current task (that is, trials misclassified with $CD_{TASK}$ activity), the cue information encoded in the delay activity (measured as $CD_{CUE}$ classification accuracy) was concomitantly lower (Extended Data Fig. 9i–m), revealing a link between these two coding dimensions (Pearson's $r = 0.26 \pm 0.18$ for area AM experiments and $r = 0.18 \pm 0.26$ for area M2 experiments; mean ± 95% confidence interval (CI) across experiments).

To examine the specificity of working memory representations to the WM task, we investigated the maintenance of cue information during the Discrimination task delay periods. Neural population activity encoding the preceding cues during the delay periods of the WM task ($CD_{CUE}$ activity) was largely absent during the Discrimination task, with only 58% of trials correctly classified in areas AM and M2 (compared with 89% and 80% classification accuracy in AM and M2, respectively, during the WM task; Extended Data Fig. 9a,e). As a negative control, we observed that the population activity that discriminated the cues during their presentation in the WM task (stimulus period $CD_{CUE}$; Fig. 3b) was able to correctly predict the same stimuli during the Discrimination task (95% and 79% classification accuracy for area AM and area M2, respectively; Extended Data Fig. 9a,e). Thus, unlike the delay representations of preceding cues, the sensory representations of the cues were encoded in a similar manner whether or not the mouse was engaged in the WM task. Furthermore, delay period $CD_{CUE}$ activity during the Discrimination task was less robust (measured as the $R^2$ of the first and the second half of the delay period activity; $P < 0.001$ in both areas) and decayed faster (measured as the slope between the first and second half of the delay period activity; $P < 0.05$ in area M2 and $P < 0.001$ in area AM) as compared to the delay $CD_{CUE}$ activity during the WM task, and did not predict correct behavioural responses to the subsequent stimuli during the Discrimination task ($P > 0.05$ in both areas; Extended Data Fig. 9b–d,f–h). Thus, neural populations in areas AM and M2 maintained the delay representations of preceding cue stimuli selectively during the WM task.

Together, our results indicate that distributed areas of the neocortex maintained persistent representations of working memory throughout the delay periods, which were embedded within a high-dimensional subspace of population activity, the collapse of which led to working memory-specific behavioural deficits.

## Role of cortical interactions in working memory

The integration of sensory input with internal representations, which is essential to theories such as predictive coding[54], active inference[23] and hierarchical Bayesian inference[55], has been proposed to be mapped onto the reciprocal interactions between cortical areas, such that feedback from any one area is dependent on the concurrent input to it from the other[19,20]. To test whether such mechanisms could underlie the distributed maintenance of working memory in the neocortex, we used two-photon calcium imaging to measure the activity of axons originating from area M2 and projecting to area AM, while simultaneously optogenetically silencing the activity in area AM at the onset of the delay period (Fig. 4a; see Methods). We also repeated this experiment with the areas reversed, recording area AM axonal activity in area M2 while inactivating area M2 (Fig. 4g). These experiments examine the functional feedback influences that areas AM and M2 have on each other, which may be mediated through direct corticocortical projections or indirect subcortical pathways.

Inactivating area AM at delay onset significantly affected the average delay activity of 30% of delay-active M2 axonal boutons in area AM (of $n = 2,162$ boutons from 19 experiments; Fig. 4b; $\alpha = 0.05$; see Methods), whereas inactivating area M2 affected the delay activity of 36% of AM axonal boutons in area M2 (of $n = 1,952$ boutons from 19 experiments; Fig. 4h). Consistent with previous in vitro[56] and in vivo[19] characterizations, and reports of intracortical feedback being largely modulatory[57], the net influence of inactivating either area AM or M2 on the respective feedback activity was neither inhibitory nor excitatory, and led to no significant change in the average or first PC (PC1) of the boutons' delay activity (Fig. 4c,d,i–j and Extended Data Fig. 10).

By contrast, $CD_{CUE}$ activity during the delay was disrupted in area M2 boutons when inactivating area AM (Fig. 4e), and similarly in area AM boutons when inactivating area M2 (Fig. 4k), revealing a co-dependence of the working memory representations maintained within these areas. Extended Data Figure 10 shows the effects of target area silencing on feedback activity averaged over the full delay period. Although feedback $CD_{CUE}$ activity was not completely eliminated after target area inactivation, this may reflect the partial deficits that are induced by such silencing on working memory-specific behaviour (Fig. 1i). To quantify the persistence of $CD_{CUE}$ disruption, we split the delay duration into halves and quantified the difference in $CD_{CUE}$ activity between inactivation and control trials separately for each half of the delay. Area AM → M2 boutons exhibited a transient reduction in $CD_{CUE}$ activity after inactivation of area M2, with this activity being disrupted in the first 1.6 s of the delay (Fig. 4k,l; $P = 7.01 \times 10^{-3}$) but not in the subsequent 1.6 s ($P = 0.33$), indicating that neural activity encoding the preceding cue recovered. Conversely, the $CD_{CUE}$ activity of M2 → AM boutons was disrupted for the full duration of the delay after inactivation of area AM (Fig. 4e,f; $P = 9.67 \times 10^{-4}$ and $P = 1.12 \times 10^{-3}$ for the first and second half of the delay, respectively), indicating that activity encoding the preceding cue in these axons failed to recover following delay-onset silencing.

We pooled data from all optogenetic inactivation experiments (that is, Figs. 1 and 4) to examine whether the difference in the recovery of $CD_{CUE}$ activity after delay-onset silencing was reflected in the persistence of the induced working memory-specific behavioural deficits (Fig. 5a). As predicted by the recovery of AM → M2 $CD_{CUE}$ activity, inactivation of area M2 led to an increase in incorrect responses in short delay trials (less than 1.6 s; $P = 1.12 \times 10^{-3}$), but not in long delay trials (greater than 1.6 s; $P = 0.46$). Delay-onset inactivation of area AM, on the other hand, increased incorrect responses in both short and long

delay duration trials ($P = 2.36 \times 10^{-2}$ and $P = 1.14 \times 10^{-2}$, respectively), mirroring the lack of recovery of $CD_{CUE}$ activity after such silencing. We defined a recovery index for the effect of optogenetic inactivation on behaviour and on $CD_{CUE}$ activity to summarize and contrast this relationship between areas (Fig. 5b).

Together, these results reveal first, that reciprocal corticocortical communication of working memory representations is dependent on simultaneous activity in both areas, and second, that the recovery (after inactivation of area M2), or lack of recovery (after inactivation of area AM), of these representations mirrors the delay length-dependent behavioural deficits that are induced by the inactivation.

## Discussion

By contrasting a working memory-dependent task with a working memory-independent task, with otherwise identical delay, stimulus and reward statistics, we were able to characterize key psychometric manifestations of working memory, such as slowed reaction times[34] and delay duration-dependent maintenance[33], and isolate their neural representations from other covariates of task engagement. The presence of shared, dominant modes of neural activity in both the Discrimination and the WM tasks (Fig. 2) challenges the notion that persistent delay period activity, such as the sequential firing of neurons[16,44,45], carries the sensory information that is maintained in working memory, and instead suggests that it reflects common task attributes such as the timing of the trial structure, motor preparation or reward expectation[17,27,58–60]. Our results provide support for an alternative representational scheme for the maintenance of working memory, in which dispersed cell populations encode working memory with trial-to-trial variable and uncorrelated patterns of activity. Such high-dimensional population codes have been associated with 'hot-coal'[9] or 'activity-silent'[10] theories of working memory maintenance, although further investigations with electrophysiology and targeted causal network perturbations, respectively, will be necessary to draw direct comparisons. We further observed that this high-dimensional population activity nonetheless gave rise to working memory representations that were readily decoded by a one-dimensional subspace of population activity that was robust and persistent over variable delay durations. This working memory-specific subspace of neural population activity provides a reliable encoding for reading out unreliable single-cell working memory signals[26,59,60], and may therefore determine how the network responds to and adjudicates subsequent sensory inputs[31,32].

Neural inactivation studies have reported varying degrees to which different areas of the neocortex are required during delayed decision-making tasks[37–42]. Using our two-task design, we were able to isolate the inactivation effects that were specific to visual working memory, and this revealed the contribution of multiple cortical areas (Fig. 1g–i). Nevertheless, the presence of low-dimensional neural dynamics during the delay periods in either area AM or area M2 did not underlie correct responses to the subsequent stimuli, as the recruitment of such activity (Fig. 2) and its removal by optogenetic inactivation (Fig. 1g–i) had no effect on mouse performance during the Discrimination task. Instead, errors in responses to the subsequent stimuli after optogenetic inactivation during the WM task were concomitant with the subspace-specific disruption of neural activity in cortical areas distal to the inactivated area (Fig. 5b). Our results therefore highlight the difficulty in interpreting the effects of optogenetic inactivation on behaviour[61], and demonstrate the behaviourally causal role of the subspace-specific propagation of localized inactivations across reciprocally connected cortical areas.

Inactivating the neocortex at different epochs of the trial had distinct, sometimes opposing effects on behavioural responses, and the relationship between the area silenced and the trial epoch differed between the Discrimination and the WM tasks (Fig. 1i and Extended Data Fig. 4). Furthermore, the behavioural and neural consequences of inactivation differed in their transience, with different cortical areas showing varying degrees of recovery after inactivation (Fig. 5). The ability of behaviour and neural activity to recover after inactivation of area M2, but not area AM, during the early delay period can be contrasted with the timescales of activity patterns observed locally in these areas, which have been found to be slower in anterior regions[62], but is consistent with previous reports of the rebound of preparatory population activity in the frontal cortex after inactivation[63].

The bidirectional and selective disruption of the working memory representations that are communicated between cortical areas following the inactivation of local target area activity provides direct evidence for the mechanistic role of functional cortical feedback loops in the maintenance of internally generated cognitive representations. Such mechanisms have previously been proposed to account for the influence of latent variables, such as priors or contextual information, on sensory processing in hierarchical models of visual perception[23,54,55,64]. Our results therefore suggest that common neural substrates—specifically, high-dimensional population codes—may implement internal models in the brain, independent of their sensory causes and motor consequences.

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

## Methods

### Mice and ethics

All experiments were performed under the UK Animals (Scientific Procedures) Act of 1986 (PPL PD867676F) following local ethical approval by the Sainsbury Wellcome Centre Animal Welfare Ethical Review Body. A total of 7 PV-Cre (ref. [65]) × Ai32 and 2 × VGAT-Cre × Ai32 mice (JAX 017320, JAX 016962 and JAX 024109, Jackson Laboratory; ChR2 expressed in inhibitory interneurons) were used for the behaviour and optogenetics experiments (Fig. 1). A total of seven mice—one wild type (Charles River), three Ai-148 (ref. [66]) × Cux-creER (JAX 030328 and JAX 012243, Jackson Laboratory; GCaMP6f expressed in most excitatory layer 2/3 cells under the control of tamoxifen) and three Ai-148 (Cre-dependent GCaMP6f expression in all cells) mice—were used for the cell-body imaging experiments (Figs. 2 and 3). A total of seven PV-cre mice were used for the axonal imaging experiments (Figs. 4 and 5). Mice were of either sex (10 male and 13 female) and were between 8 and 16 weeks old at the start of their experiments. Before the experiments, the three Cux-creER mice were administered tamoxifen (10 mg ml$^{-1}$) by intraperitoneal injection (1 mg per 10 g body weight), three times, every other day. Mice were co-housed with littermates in IVC cages, in reversed day–night cycle lighting conditions, with the ambient temperature and humidity set to 23 °C and 56% relative humidity, respectively.

### Surgical procedures

Before all surgeries, the mice were injected with an analgesic (carprofen 5 mg kg$^{-1}$) subcutaneously. General anaesthesia was induced with 3% isoflurane, which was then reduced to maintain a breathing rate of around 1 Hz. A custom-designed stainless steel headplate was attached to the skull using dental cement (C&B Super Bond). In some of the older mice in the behaviour and optogenetics experiments (Fig. 1), the dorsal surface of the skull was carefully thinned with a dental drill. The exposed skull was then sealed with a thin layer of light-curing dental composite (Tetric EvoFlow).

For the cell-body calcium-imaging experiments (Figs. 2 and 3), after a minimum recovery time of three days and intrinsic signal imaging (see below), a second surgery was performed to make a cranial window over areas AM and M2, identified by intrinsic signal imaging and coordinates (0.5 mm lateral, 2.5 mm anterior of bregma), respectively. A 5 mm craniotomy was made over the dorsal surface of the skull and a 300 μm thick, 5 mm diameter glass window was implanted with cyanoacrylate glue (Pattex). In the three Ai-148 mice, two 50 nl viral injections of AAV9.hSyn. Cre.WPRE.hGH (Penn Vector Core) diluted to a low titre ($5 \times 10^{10}$ vg ml$^{-1}$) in Ringer's solution were made into areas AM and M2 with a Nanoject III microinjector (Drummond Scientific). In the wild-type mouse, two 50 nl viral injections of AAV1.hSyn.GCaMP6f.WPRE.SV40 (Addgene, 100837) diluted to $5 \times 10^{12}$ vg ml$^{-1}$ was likewise made into areas AM and M2. In four of the mice, a viral injection of AAVretro.hSyn1.mCherry-2A-iCre.WPRE.SV40 ($1 \times 10^{12}$ vg ml$^{-1}$; v147 Zurich Vector Core) was made into AM (one mouse) or M2 (three mice), to help localize the respective connected areas.

For the axonal imaging experiments (Figs. 4 and 5), following a minimum recovery time of three days and intrinsic signal imaging, a second surgery was performed to make a cranial window over either area AM or M2 and perform the viral injections. In three of the mice, a 3 mm diameter craniotomy was made centred around area AM, and a smaller (less than 1 mm diameter) craniotomy was made over area M2 (identified with coordinates relative to bregma; 0.5 mm lateral, 2.5 mm anterior). Viral injections (100 nl) of AAV1.hSyn.DIO.ChrimsonR.tdTomato ($3.9 \times 10^{12}$; UNC Vector Core) and AAV1.hSyn.jGCaMP7b.WPRE (ref. [67]) ($2 \times 10^{13}$; Addgene, 104489) were then made into areas AM and M2, respectively, with a Nanoject III microinjector (Drummond Scientific). Immediately afterwards, the larger area AM craniotomy was sealed with a 3 mm glass window. In the other four mice, the same procedure was done but with areas AM and M2 reversed.

### Intrinsic signal imaging

We used intrinsic signal imaging[68] of the dorsal cortex to identify the locations of cortical areas V1 and AM. Intrinsic imaging was performed on awake mice while they were head-fixed on top of a freely rotating Styrofoam cylinder. The visual cortex was illuminated with 700 nm light, a macroscope was focused 500 μm below the cortical surface, and the collected light was bandpass-filtered centred at 700 nm (10 nm bandwidth; 67905, Edmund Optics). The images were acquired at a rate of 6.25 Hz with a 12-bit CCD camera (1300QF, VDS Vosskühler), an image acquisition board (PCI-1422, National Instruments) and custom software written in LabVIEW (National Instruments). The visual stimuli, presented on a display 22.5 cm away from the left eye, were generated using Psychophysics Toolbox[69] running in MATLAB (MathWorks), and consisted of square-wave gratings, covering a 40° visual angle, 0.08 cycles per degree, drifting at 4 Hz in eight random directions, presented on an isoluminant grey background for 2 s, with 18 s inter-stimulus intervals. The gratings were presented alternatively at two positions, at 15° elevation and either 30° or 80° azimuth. Response maps to the grating patches at either position were used to identify the centres of V1 and AM, using a reference map[70].

### Behavioural shaping and apparatus

Mice were trained for 2–6 weeks before the initiation of data acquisition. Mice were food-restricted for the full duration of the behavioural training and data acquisition, with no scheduled breaks. The maximum weight loss was limited to 80% of their pre-restriction body weight. Food restriction began at least three days after headplate implantation surgery. The mice were trained for approximately 2 h every day, once a day. For the first few days of training the mice were handled on a cloth and iteratively fed Ensure Plus strawberry milkshake (Abbott Laboratories) through a syringe to acclimate them to the behavioural training environment.

Over the next few days, the mice were trained to run on a freely rotating Styrofoam cylinder, while head-fixed, in front of the visual stimulation display (U2415, Dell; 60 Hz refresh rate), placed 22.5 cm away from their left eye and oriented at 32° relative to midline. A reward delivery spout was positioned under the snout of the mice from which a drop of Ensure Plus was occasionally delivered by the experimenter to encourage running. Licks were detected with a piezoelectric diaphragm sensor (Murata 7BB-12-9) placed under the spout.

Once the mice were running freely, they were trained to perform a simple visual detection task, in which the onset of a visual stimulus was associated with reward (a drop of Ensure Plus), delivered from the reward spout after the stimulus onset. The mouse running speed was recorded with a rotary encoder (05.2400.1122.1000, Kübler), and the mice had to run a specified distance between the stimulus presentations. This distance was variable and set such that the mice received roughly one reward per minute. Reward delivery was triggered when the mouse licked the spout any time during a response window of 1 s following the stimulus onset. If the mice failed to lick in response to the stimulus, an automatic reward was delivered at the end of the response window. The detection of licks, reward delivery, recording of data and the presentation of visual stimuli were controlled by a custom LabVIEW software (National Instruments). Visual stimuli were generated by custom software (https://github.com/Ivan-Voitov/Vizi) written in Unity (Unity Technologies). Hardware interfacing was conducted with a data acquisition board (PCIe-6321, National Instruments).

The visual stimuli were drifting square-wave gratings presented at 100% contrast, 0.025 cycles per degree, covering 60° of the visual field of the mice, centred at 15° elevation and 45° azimuth, presented on an isoluminant grey background. The luminance of the monitor was set at 0 cd m$^{-2}$, 22.5 cd m$^{-2}$ and 45 cd m$^{-2}$, at black, grey and white values, respectively. The grating stimuli were cycling in a closed loop with the mouse running speed for the first one to two weeks of training, and were then fixed at 3.5 Hz for the remainder of the experiments.

Once the mice were running comfortably on the Styrofoam cylinder and licking in response to the presentation of grating stimuli (after one to four weeks of training), the task parameters were introduced to begin training either the Discrimination or the WM tasks. The order in which the two tasks were trained varied between mice.

## Task training and design

Both the Discrimination and the WM tasks consisted of alternating delay (grey background) and stimulus (full contrast grating) periods (Fig. 1a). Delay period durations were sampled from an exponential distribution with a mean of 800 ms, and then had 800 ms added (that is, a 800 ms offset or minimum duration). The sampled delay periods durations were then capped at 4,000 ms by resampling the duration from a uniform distribution between 3,600 and 4,000 ms whenever this cap was reached (to ensure a minimal effect on the average duration of the delay). The resulting average delay duration was 1,600 ms. The duration of the stimulus period was proportional to the mouse running speed that is, was a distance to traverse), and was set to either 100 cm or 80 cm depending on the average running speed of the mouse, such that the stimulus period took a similar time to traverse by all mice if they did not stop running. Forcing the mice to traverse a certain distance to get through the stimulus period promoted persistent running in mice over the course of each session, which in turn ensured stereotyped movement within the delay periods and reduced variability between mice. The resulting average stimulus duration was 1,967 ms.

In both the Discrimination and the WM tasks, the orientation of the grating stimuli classified them as either go or no-go (Fig. 1a and Extended Data Fig. 1a). The stimuli presented were cues (no-go; 80% of trials), probes (no-go, 10% of trials) or targets (go, 10% of trials). Cue stimuli were gratings oriented at 0° (vertical) in the Discrimination task and either +45° or −45° in the WM task; probe stimuli were gratings oriented at 90° (horizontal) in both tasks; and target stimuli were gratings oriented at +45° or −45° in both tasks. The stimulus presented for each trial was sampled randomly with the aforementioned probabilities, with the exception that after a probe or a target stimulus, a cue stimulus was mandatory (100% probability; Extended Data Fig. 1a). The only difference between the Discrimination and the WM tasks was that the cues were always vertical (0°) gratings in the Discrimination task, but the cues (oriented −45° or +45°) were mirrored in orientation relative to the current targets (+45° and −45°, respectively) in the WM task. Accordingly, in the WM task, the orientations of the cues were only switched after the presentation of a target, whereas in the Discrimination task the cues were always the same (vertical gratings). Because cues were more frequent than the other stimulus types (80% probability), most trials were consecutive cues of the same orientation. In addition to serving as common no-go stimuli in both tasks, the probes ensured that the mice were not using an odd-ball detection strategy to perform either task (that is, responding to rare stimuli), as the probe presentation probability was the same as the target probability.

Sequences of delay and stimulus epochs were presented continuously (that is, no inter-trial intervals), with individual trials composed of a delay and stimulus pair, such that each trial's stimulus served as a cue to the subsequent stimulus. If mice licked the spout during a 1 s response window after the onset of the target stimuli (that is, go trials), the trials were classified as hit trials; otherwise, they were classified as miss trials. In the miss trials, the mice received an automatic reward at the end of the response window, consisting of half of the normal reward amount. The same 1 s response window was used to classify responses to the cue and probe stimuli (that is, no-go trials) as false alarms (FAs) or correct rejections (CRs). Licking during the no-go trials was not punished.

Once the mice were trained in both tasks (sequentially, with the order varying between mice), the blocked task structure was introduced, with the Discrimination and WM tasks alternating every 415 trials over the course of each session in the behavioural and optogenetic experiments, and a similar but variable number of trials (300–600) in the imaging

experiments (to accommodate a variable number of trials between the rotation blocks; see below). Mice performed between three and eight task blocks per session. Mice switched task blocks quickly (within a few trials), as the presence or absence of the Discrimination task cue stimulus (a vertical grating) was informative of the task block. Similar two-task designs have previously been used to disambiguate the neural correlates of specific cognitive processes by isolating neural representations of interest from 'condition-independent' neural activity[29,71]. One potential drawback of the two-task design is that neural activity may be recruited that would otherwise be absent if the mice were only trained on one task. Nevertheless, ethological behaviour is characterized by flexible switching between a vast repertoire of previously learned behaviours, and two-task designs therefore impose a reasonably conservative control for investigating neural correlates of cognitive processes.

For the imaging experiments (Figs. 2–5), the ±45° oriented gratings (that is, cues in the WM task and targets in both tasks) were instead oriented at ±30°, and the rotation block structure was introduced. The goal of the rotation blocks was to match the cue stimulus grating orientations between the Discrimination and WM tasks (Extended Data Fig. 1b). The rotation blocks consisted of blocks of several hundred trials, out of phase with the task blocks described above, during which all stimuli, except for the 90° oriented probes, were rotated either 15° clockwise or 15° counter-clockwise. As such, the resultant stimulus orientations for the cues and targets were −45°, −15° and +15°, and −15°, +15° and +45°, in the clockwise and counter-clockwise rotation blocks, respectively. In between two rotation blocks, the stimulus orientation angles were changed slowly in a continuous fashion (averaging around 10 min for a full 30° rotation), such that the mouse performance was not disrupted. No previous training was required for the mice to perform these rotation blocks, and there was minimal interference with the ability of the mice to alternate task blocks as the sudden presence or absence of a stimulus in between the two cues in the WM task remained an abrupt indicator of a task block switch. A typical session involved alternating between switching the task that the mice were performing and rotating the stimuli that the mice were seeing, such that the −15° and +15° oriented stimuli served as either the cues or the targets in both the Discrimination and the WM tasks, being matched in orientation across rotation blocks.

## Optogenetic inactivation of multiple cortical areas

To silence neuronal activity during behaviour (Fig. 1), we optogenetically activated ChR2-expressing inhibitory interneurons using a 473 nm laser (OBIS, Coherent) with a galvanometer scanning photostimulation system[37]. In brief, laser light was reflected off of two galvanometer scanning mirrors to target the light, expanded by two plano-convex lenses (5× magnification; LA1951-A and LA1384-A, Thorlabs) and then focused onto the brain with a 200 mm focal-length lens (AC508-200-A, Thorlabs). A polarizing beamsplitter was placed in the light path, enabling us to simultaneously image the surface of the skull (camera, 22BUC03, ImagingSource) to identify and select locations for cortical inactivation. The photostimulation and image acquisition were controlled by custom LabVIEW software and a data acquisition card (PCIe-6321; National Instruments). The laser light was pulsed at 50 Hz, with a 50% duty cycle. The laser power was set to 3 mW average (6 mW peak power) for the first 400 ms of stimulation and then linearly tapered off to 0 mW over 200 ms to minimize activity-rebound effects. The propagation of reflected light to the eyes of the mouse was blocked by either a cement wall around the visible skull or a custom 3D-printed plastic lightshield implanted during the headplate surgery. Silencing occurred in 12% of trials, at one of three epochs; the onset of the delay, the delay end (600 ms before stimulus onset) or the onset of the stimulus. Because silencing at the end of the delay was difficult to interpret, as the mice could use the silencing to predict the stimulus onset and respond pre-emptively, we discarded delay end silencing trials from all of our analyses. The cortical area to be silenced was chosen randomly trial-to-trial, and

was identified either by the coordinates relative to bregma for areas M2 and S1, or by intrinsic signal imaging for areas V1 and AM.

A 470 nm masking light, emitted from an optical fibre (FT400EMT; Thorlabs) coupled to an LED (M470F3; Thorlabs), placed 20 cm above the mouse (roughly in line with the laser light path), diffusely illuminated the head of the mouse (2 mW at the fibre tip). The masking light was flashed on each trial in the same manner as the optogenetic silencing light (400 ms plus 200 ms ramp down), at one of the three onset times (delay onset, delay end or stimulus onset), chosen randomly on control trials and at a matched onset to the optogenetic silencing light in silencing trials. This masking light therefore had the same dynamics as the laser light, and was used to both mask the presence of the laser light during the silencing trials and as a negative control for possible light-onset-induced behavioural changes during the control trials. The masking light alone (that is, during the control trials) had no effect during the 600 ms of masking light presentation, in either the Discrimination or the WM task, and at either of the onset times, on running speed ($n = 9$ mice, $P > 0.05$ for all onset times and tasks, two-sided signed-rank test) or stimulus responses ($n = 9$ mice, $P > 0.05$ for all onset times and tasks, two-sided signed-rank test).

## Two-photon calcium imaging of cell-body populations

For the cell-body imaging experiments (Figs. 2 and 3), we imaged the calcium dynamics in layer 2/3 cells of areas AM and M2 simultaneously using a wide field of view two-photon microscope[72]. The surface blood vessel pattern above the imaging sites was compared with the blood vessel pattern from the intrinsic signal imaging maps to confirm the location of area AM. Fields of view over each area were 600 μm × 600 μm and spread over four axial planes 50 μm apart. Frames from all eight fields of view were acquired at 4.68 Hz. The image acquisition software was ScanImage[73]. Two cameras (22BUC03, ImagingSource) were positioned to acquire greyscale videos of the body and left pupil at 30 Hz. The visual stimulation display was turned off during the linear phase of the resonant scanners corresponding to the image acquisition (12 kHz), so as to avoid display light spill-through into the imaging frames.

The imaging data were pre-processed using modified CaImAn software[74]. In brief, cell masks were identified as point-seeds at individual cell locations by the experimenter, using the registered mean frame image as well as a pixel-surround correlation image. The CaImAn cell segmentation and neuropil demixing algorithms (based on constrained non-negative matrix factorization) were then applied to the seeds to define the mask boundaries and extract the calcium time series from individual masks. A second round of experimenter-mediated curation was performed on these masks, and the calcium time series were re-extracted. The calcium time series were then detrended, normalized ($\Delta F/F_0$) and deconvolved using the standard CaImAn algorithms (FOOPSI[75]). For all data for which the raw $\Delta F/F_0$ activity is shown, the underlying statistical analyses (for example, estimating the latencies of delay responses for Fig. 2e,h and Extended Data Fig. 5a,b) were done on the deconvolved calcium activity. For all other statistical and population analyses, only deconvolved calcium activity was used. Imaging frames with low correlations to the average image (putative movement artefacts), or significant pupil movements (greater than five standard deviations from the mean), were discarded. Finally, individual cells were further curated using local neuropil correlations, signal-to-noise ratio and the number of calcium events, to identify cells with sufficient levels of activity for analysis, resulting in an average of 311 ± 57 active cells per area AM experiment, and 309 ± 69 active cells per area M2 experiment.

## Two-photon calcium imaging for axonal imaging and simultaneous optogenetic silencing

For the feedback imaging and local silencing experiments (Figs. 4 and 5), we imaged the axonal calcium signals with a custom-built two-photon microscope. We acquired two planes 25 μm apart in layer 1 of area AM ($n = 4$ mice) or area M2 ($n = 3$ mice), with a field of view of 400 × 400 μm

at a frame rate of 22.78 Hz. Two cameras (22BUC03, ImagingSource) were used to record the pupil and body positions at 30 Hz. For each imaging site, we also recorded a volumetric image stack to confirm the location of tdTomato-ChrimsonR transduced PV[+] cells directly underneath the recorded axons.

The imaging data were registered and pre-processed using a modified Suite2p pipeline[76]. The data were registered, bouton masks were extracted, and their calcium traces were baseline-subtracted. $F_0$ normalization was not performed owing to the very low baseline levels of fluorescence. Frames with low correlations to the registered average image or frames with significant eye movements were discarded. The boutons' time-series data were then clustered into putative axons using custom scripts written in MATLAB (MathWorks). In brief, we used independent component analysis (ICA) to extract a 40-dimensional temporal feature space from the full dimensional time series. The activity of all boutons, projected into this feature space, was then clustered using a Gaussian mixture model. The number of clusters was chosen by minimizing an adjusted Akaike information criterion error. Boutons with significant distances from their allocated cluster centre were not clustered, and all other boutons were clustered together by simply averaging their signals. This clustering procedure returned the activity time series of putative axons, each averaging eight boutons, which were then used for all analyses.

Optogenetic silencing of the area targeted by the feedback axons was achieved by stimulating the PV[+] ChrimsonR-expressing cells immediately underneath the imaging site. A 637 nm laser (OBIS, Coherent) was relayed through a 400 μm-diameter optical fibre to a 100 mm focal-length lens, which then relayed the light onto the back aperture of the objective. The optogenetic laser power, measured immediately in front of the objective, was 6 mW (average, 12 mW peak power), pulsed at 60 Hz with a 50% duty cycle. Optogenetic silencing occurred in 15% of trials, during which the light was introduced at the onset of the delay period at full power for 400 ms, and then linearly ramped down to 0 mW over the following 200 ms. The optogenetic laser and the visual stimulation display were turned off during the linear phase of the resonant scanner (12 kHz), so as to avoid light spill-through during ongoing imaging frame acquisition.

## Data analysis

Trials in which mice stopped running or licked during the inter-stimulus delay period (Extended Data Fig. 3b), trials following either targets or probes (that is, the trials that were 100% probable to be cues), and trials in which optogenetic silencing occurred at the end of the delay period (see above) were excluded from all analyses. All optogenetic silencing trials were excluded from analyses that characterized the behaviour (Fig. 1b–f and Extended Data Fig. 2). For the $d'$ analyses of per-mouse delay duration effects (Fig. 1e), positive infinities (that is, when no misses occurred) were treated as non-existent data points for statistical analysis. $d'$ was defined as:

$$d' = \varphi^{-1}_{(\text{Hit rate})} - \varphi^{-1}_{(\text{FA rate})},$$

in which $\varphi$ is the Gaussian cumulative distribution function.

Statistical analyses of optogenetic inactivation effects (Fig. 1i and Extended Data Fig. 4) were done by pooling trials from nine mice ($n = 173,432$ trials) and performing a Fisher's exact test, separately for cue, probe and target trials, and split by task. Significance levels were accordingly adjusted for multiple comparisons. Bar plot values were the trial-averaged optogenetic silencing effects subtracted from the averages of the control trials (in which no silencing occurred), and error bars represent the 95% CI of the silencing trials (that is, binomial confidence intervals).

For the imaging experiments (Figs. 2–5), although the inter-stimulus delay periods ranged from 0 to 4 s as in the behaviour and optogenetics dataset (Fig. 1), the lower numbers of trials available within each individual imaging session led to there being too few long delay duration trials to be used for neural activity analysis (owing to the exponential

distribution of delay durations). As such, all analyses were limited to delay durations ranging from 0 to 3.2 s.

As comparisons of neural activity between tasks were made across rotation blocks (Extended Data Fig. 1b), if both rotation blocks were present in both tasks within a single session, individual experiments consisted of the Discrimination and WM task blocks with the matched task stimuli (+15° or −15°) that occurred during opposite rotation blocks (that is, there were up to two experiments per session). If only one task stimulus was common to both tasks (for example, if only one task switch and stimulus rotation occurred), experiments were simply the full imaging sessions. All subsequent analyses of neural activity (Figs. 2–5) were conducted on such experiments. For depictions of single-cell responses (Fig. 2), if there were two experiments within a single session, the second experiment within the session was discarded so as to not depict the same cells multiple times.

In the cell-body imaging experiments, for analyses limited to delay- or stimulus-responsive cells (Fig. 2b,d,e,h and Extended Data Fig. 5a,b), we defined delay or stimulus responsiveness as exceeding an effect size threshold (0.2 deconvolved $\Delta F/F_0$ difference post- versus pre-delay or -stimulus) and being significantly different post- versus pre-delay or -stimulus onset (two-sided paired-sample $t$-test; $\alpha = 0.01$). For axonal imaging data, we likewise restricted analyses to axons that had a significant amount of delay-evoked activity, defined as 0.2 $z$-scored $\Delta F$ more in any one second of the delay than the last second of the preceding stimulus with a $\alpha = 0.01$ significance difference. For the analyses of the latency of single-cell responses during the delay (Fig. 2e,h and Extended Data Fig. 5a,b), all odd-numbered trials were taken out and used to estimate the response latencies (by taking the mean of Gaussian curves fit to the trial-averaged responses), and all of the remaining (even) trials were split by task and averaged for display.

For the analysis of low-dimensional neural dynamics (Fig. 2f,g,i,j), the activity (deconvolved $\Delta F/F_0$) of all cells was pooled across all experiments. First, we trial-averaged the delay and stimulus responses of all active cells in all experiments, concatenated the resulting delay and stimulus responses and calculated the PCs of these responses (that is, of the pooled pseudo-population of cells). We then separately projected the trial-averaged Discrimination task and WM task activities of all cells into the first three PCs. To plot the resultant activity dynamics (Fig. 2f,i), we further separated trials by the length of their delay period, and then interpolated and smoothed the resulting activity projections with a half-normal filter (that is, causal; $\sigma = 100$ ms) to help with visualization. The respective statistical analyses (that is, the Euclidean distances between projections; Fig. 2g,j) were performed using all trials with no interpolation or smoothing.

For all of the population analyses (Figs. 3 and 4), we identified the task or cue coding dimensions ($CD_{TASK}$ and $CD_{CUE}$, respectively) by fitting a simple linear model (using LDA) to the condition-specific (condition being the task or cue identity of each trial) delay-averaged population activity (deconvolved $\Delta F/F_0$) within each experiment (that is, a cells × trials matrix describing the activity of each cell averaged over each trial's delay period). The coding dimensions were defined as the vectors that separated the population activity during the delay periods of the Discrimination and WM tasks ($CD_{TASK}$), or the delay periods after the two cues in the WM task ($CD_{CUE}$). The coding dimensions were not orthogonalized. Incorrect trials and trials with optogenetic silencing were excluded when calculating these coding dimensions. We used the following formula to identify the discrimination vectors:

$$CD_{\overrightarrow{ab}} = \hat{\Sigma}^{-1}(\mu_a - \mu_b)$$

$$\hat{\Sigma} = \Sigma + I\gamma$$

in which $a$ and $b$ are the trial conditions (the task or cue), $\Sigma$ is the cells' covariance matrix (that is, of the cells × trials matrix) and $\gamma$ is a

regularization parameter. $\gamma$ was set to a low value ($1 \times 10^{-4}$), and served to stabilize matrix inversion; changing the value of $\gamma$ did not change the results significantly. Using other linear binary classifiers (for example, logistic regression) to identify these coding dimensions achieved very similar results. For Fig. 3b and Extended Data Figs. 5c–n and 7, the PCs were calculated from single trials, concatenated in time, from all data that were nominally used for analysis (that is, the first trial after a probe or a target and trials during the stimulus rotation periods were excluded; see above).

All reported task or cue decoding accuracies were the average cross-validation (leave-one-out) test accuracies, calculated by averaging each trial's prediction of task or cue given the coding dimensions derived from the respective experiment's remaining trials (that is, one classification accuracy was derived per experiment). All projections of the neural population activity onto the respective coding dimensions (for example, Figs. 3c,f,i,l and 4e,k) are likewise the projections of the activity of left-out single trials onto the coding dimensions calculated from their respective experiment's remaining trials. The decoding accuracies of the training sets are reported in Extended Data Fig. 7. Importantly, the reported decoding accuracies for incorrect trials (Fig. 3e,h,k,n), optogenetic silencing trials (Fig. 4e,f,k,l) and Discrimination task $CD_{CUE}$ trials (Extended Data Fig. 9a–h), which were excluded from the training sets, were calculated using the same models as those used for the reported decoding accuracies of the (left-out) correct and non-silenced $CD_{TASK}$ and $CD_{CUE}$ trials.

For the analyses decoding the task or cue before correct or incorrect behavioural responses (Fig. 3e,h,k,n), instead of projecting the average delay activity of a trial onto the coding dimensions to identify that trial's score, we instead averaged five randomly sampled imaging frames (1,068 ms of data) from the delay. This was done to eliminate any potential confounds introduced by the fact that longer delay period trials have a higher signal-to-noise ratio (that is, more frames to average) as well as a higher probability of preceding a false alarm during the WM task (Fig. 1c). Similar results were attained without this procedure, or by averaging the first five frames of each trial's delay period.

### Reporting summary

Further information on research design is available in the Nature Research Reporting Summary linked to this paper.

### Data availability

The data that support the findings of this study are available from the corresponding author upon reasonable request. Source data are provided with this paper.

### Code availability

The analysis code is publicly available at www.github.com/ivan-voitov/loops.

Acknowledgements We thank M. Li for animal husbandry and help with training the mice; R. Campbell for help with microscopy; M. Rio for help with the calcium data analysis pipeline; and the T.D.M.F. laboratory, M. Sahani, A. Akrami, M. Javadzadeh, G. Keller, A. Khan, R. Friedrich, S. Hofer, S. Furutachi and M. Lohse for feedback on the manuscript and discussions.

Author contributions I.V. designed the experiments, collected the data and analysed the data with supervision from T.D.M.F. I.V. and T.D.M.F. wrote the paper.

Competing interests The authors declare no competing interests.

Additional information

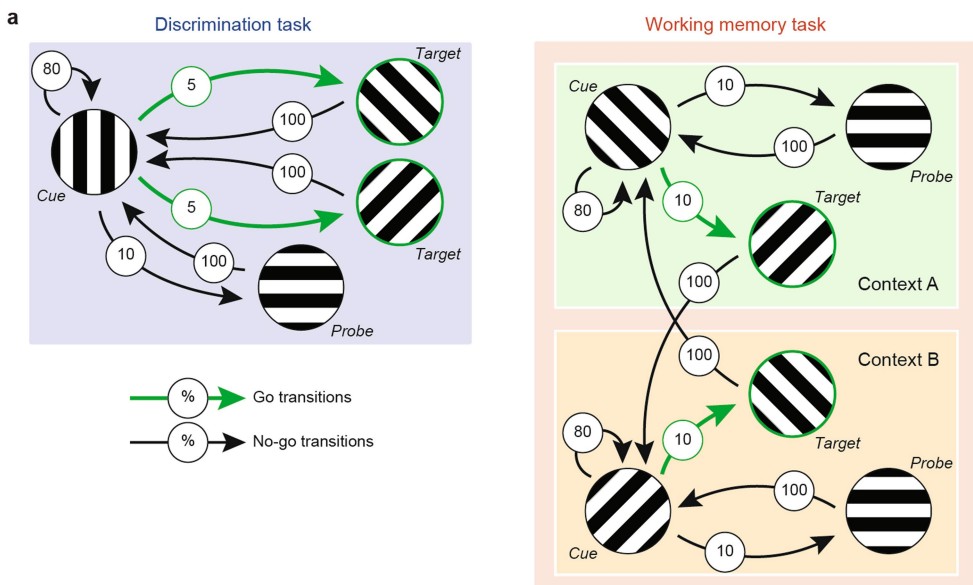

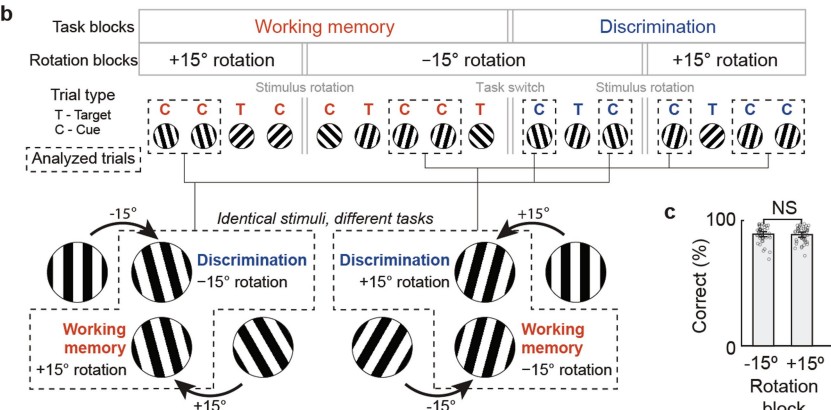

**Extended Data Fig. 1 | Alternative behavioural task schematic and rotation block design. a**, A schematic of the task design in the style of a finite-state machine. Each transition between stimuli is accompanied by a delay period (0.8 s to 4 s in length) sampled from a flat hazard rate distribution. Note that the transitions following probes and targets are non-probabilistic (are 100% likely), and were therefore excluded from analysis. **b**, A schematic of the rotation block design (related to Figs. 2–4). Over the course of each session, all cue and target stimuli were rotated 15° clockwise or 15° counter-clockwise in blocks of several hundred trials. Rotation blocks were staggered relative to task blocks, such that trials from each task coincided with each rotation. Such stimulus rotations made the cues in the Discrimination task identical in orientation to one of the cues in the WM task that was in the opposing rotation block (e.g. 0° → +15° and +30° → +15°). **c**, Percentage of correct responses in the two rotation blocks (error bars are 95% CI, $p = 0.78$, $n = 43$ sessions from 7 mice, two-sided signed-rank test).

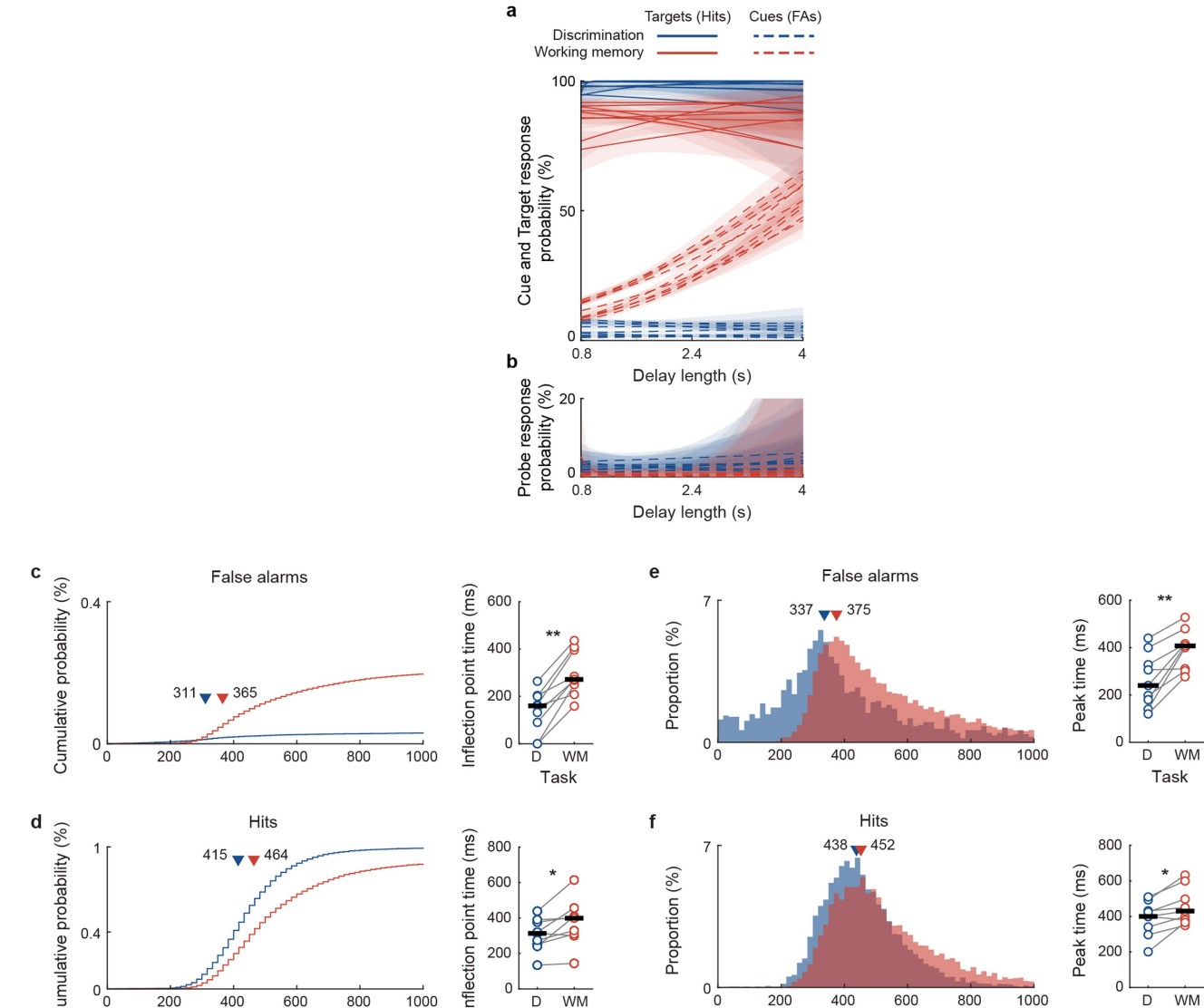

**Extended Data Fig. 2 | Psychometric consequences of working memory engagement. a–b**, Performance of each mouse as a function of the preceding delay length, as in Fig. 1c,d, as measured by responses to the cues and targets (**a**), and probes (**b**) in the Discrimination (blue) and WM (red) tasks. Responses to cues and probes are false alarms (FAs) and responses to targets are hits. Plotted curves are fits with a linear delay length dependence and a logit link function to the response probabilities. Shaded regions represent the ± 95% CI for the slope and intercept of each fit. Only responses to the cue stimulus in the WM task showed a significant relationship with the delay length (i.e. a non-zero slope, $p = 3.91 \times 10^{-3}$, $n = 9$ mice, two-sided signed-rank test). For the remaining response types: $p = 0.73$, $p = 0.57$, and $p = 0.91$ for the Discrimination task cues, targets, and probes, respectively, and $p = 0.91$ and $p = 0.91$ for the WM task targets and probes, respectively. **c**, Left, cumulative probability of false alarms (FAs) to cues at different time lags from the stimulus onset for the

Discrimination (blue) and WM (red) tasks. Data from all mice were pooled together, and binned at 16.67 ms. Arrows represent inflection point times for the two tasks (maxima of bin-to-bin differences). Right, differences between the two tasks for inflection point times, calculating separately for each mouse ($n = 9$, $p = 3.91 \times 10^{-3}$, two-sided signed-rank test). Horizontal bars represent medians. **d**, As in **c**, for hits to targets, $p = 1.95 \times 10^{-2}$, two-sided signed-rank test. **e**, Left, proportion of FAs to cues at different time lags from the stimulus onset for the Discrimination (blue) and WM (red) tasks. Data from all mice were pooled together, and binned at 16.67 ms. Arrows represent the times with the largest proportion of responses in each task (i.e., peaks of histograms). Right, differences between the two tasks for peak reaction times, calculating separately for each mouse ($n = 9$, $p = 3.91 \times 10^{-3}$, two-sided signed-rank test). Horizontal bars represent median peak reaction times. **f**, Same as **e**, for hits to targets. $p = 1.12 \times 10^{-2}$, two-sided signed-rank test.

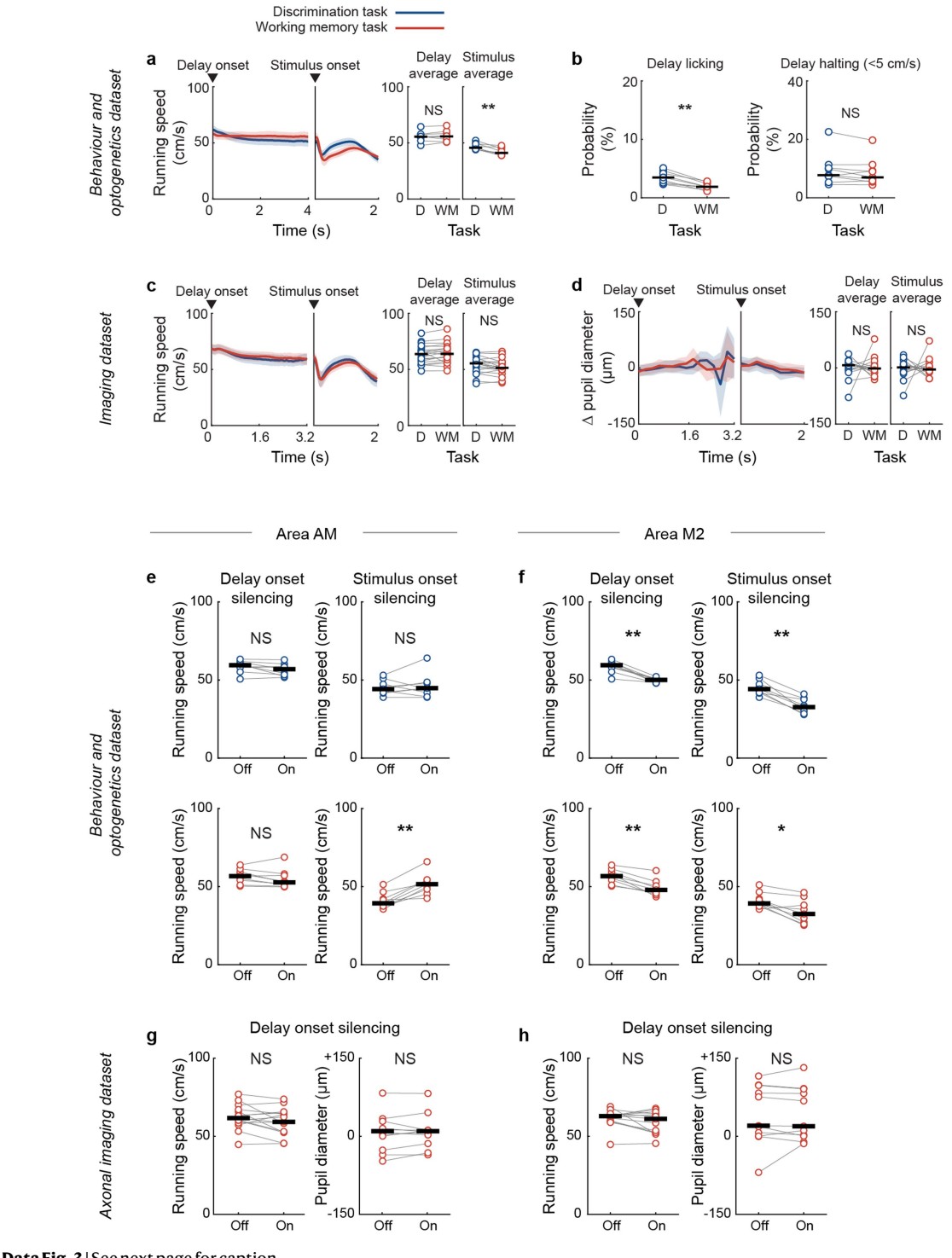

**Extended Data Fig. 3** | See next page for caption.

**Extended Data Fig. 3 | Effects of working memory engagement and optogenetic inactivation on movement and arousal. a**, Left, running speeds for the Discrimination (blue) and WM (red) tasks, averaged across mice in the behaviour and optogenetic silencing dataset (Fig. 1; $n = 9$ mice), during the delay and stimulus periods, binned at 16.67 ms. Shaded regions are ± 95% CI. Right, running speed differences between the tasks, split by animal, averaged over the duration of each delay (0 to 4 s) and stimulus (0 to 2 s) periods, $p = 0.25$ and $p = 3.91 \times 10^{-3}$, respectively, two-sided signed-rank test. No adjustments for multiple comparisons were made. Horizontal bars are the medians. **b**, Percentage of trials with responses during the delay period (left) or halts in running during the delay period (right), averaged for each mouse in the behaviour and optogenetic silencing dataset. Early responses and halts, compared between tasks, $p = 3.91 \times 10^{-3}$ and $p = 0.16$, $n = 9$ mice, two-sided signed-rank test, respectively. Horizontal bars are the medians. **c**, Same as in **a**, but for the imaging (Figs. 2 and 3) dataset, binned at 214 ms, averaged across experiments ($n = 20$, from 7 mice), $p = 0.05$ and $p = 0.07$ for the delay and stimulus periods, respectively, two-sided signed-rank test, no adjustments for multiple comparisons. **d**, Pupil diameter for the two tasks over the course of the delay and stimulus periods, as in **c**, for the imaging dataset ($n = 15$ experiments from 7 mice, $p = 0.56$ and $p = 0.85$, two-sided signed-rank test, no adjustments for multiple comparisons). Pupil diameters were mean-subtracted (means calculated from the full session) to better identify systematic differences

between the two tasks. **e**, The effect of optogenetically silencing area AM on mouse running speeds, during the 600 ms window of the silencing light and ramp down, at the onset of the delay (left column) and at the onset of the stimulus (right column), for the Discrimination (top row) and WM (bottom row) tasks, in the behaviour and optogenetics dataset (i.e., Fig. 1, $n = 9$ mice, two-sided signed-rank test). Horizontal bars are medians. A three-way ANOVA with the silencing, silencing onset, and task conditions found a significant effect of silencing onset ($p = 7.25 \times 10^{-12}$) and an interaction between the silencing onset and silencing conditions ($p = 3.76 \times 10^{-2}$), but no significant effect of silencing ($p = 0.16$), task ($p = 0.64$), or task and silencing condition interaction ($p = 0.06$). **f**, As in **e**, when silencing area M2. A three-way ANOVA with the silencing, silencing onset, and task conditions found a significant effect of silencing onset ($p = 7.20 \times 10^{-22}$) and silencing conditions ($p = 8.27 \times 10^{-11}$), but no significant effect of task ($p = 0.18$) or task and silencing condition interaction ($p = 0.19$). **g**, The effect of optogenetically silencing area AM at the onset of the delay in the axonal imaging dataset (i.e., Fig. 4, $n = 13$ sessions from 4 mice), on mouse running speeds (left; $p = 0.11$, two-sided signed-rank test) and pupil diameters (right; $p = 0.57$, two-sided signed-rank test), during the 600 ms window of the silencing light and ramp down. Pupil diameters were mean-subtracted (means calculated from the full session). **h**, As in **g**, when silencing area M2 ($n = 15$ sessions from 3 mice), running speed ($p = 0.17$) and pupil diameter ($p = 0.70$), two-sided signed-rank test.

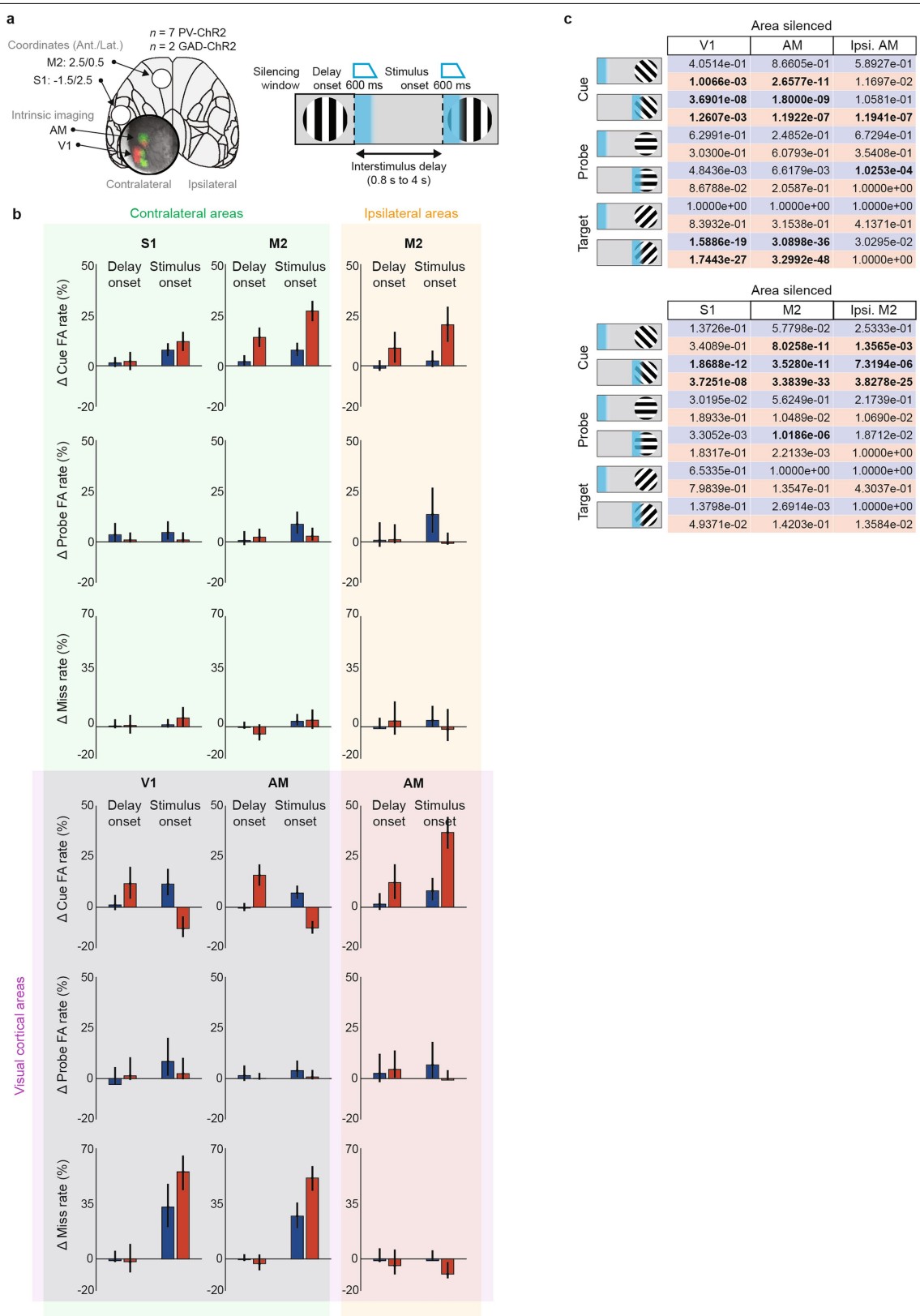

**Extended Data Fig. 4 | Optogenetic silencing effects. a**, Schematic of the optogenetic silencing design, as in Fig. 1g. **b**, Average optogenetic silencing effect on responses to cues (FAs), probes (FAs), and targets (hits) for all areas silenced (labels), during both tasks (Discrimination, blue, and WM, red), and for the two silencing onsets (at the onset of the delay, left, and at the onset of the stimulus, right). Individual bars represent the differences in mean FA or miss rate between silenced and control trials ($n = 173,432$, pooled from all 9 mice).

Error bars represent 95% CI of silencing trials. Shaded background regions group the areas silenced into contralateral, ipsilateral, and visual cortical areas. **c**, Statistical significance of optogenetic silencing effects ($p$ values) for each effect shown in **b**, two-sided Fisher's exact test, with statistically significant effects in bold font. Significance thresholds were adjusted for multiple comparisons (36 comparisons, Bonferroni correction, $\alpha = 0.0014$).

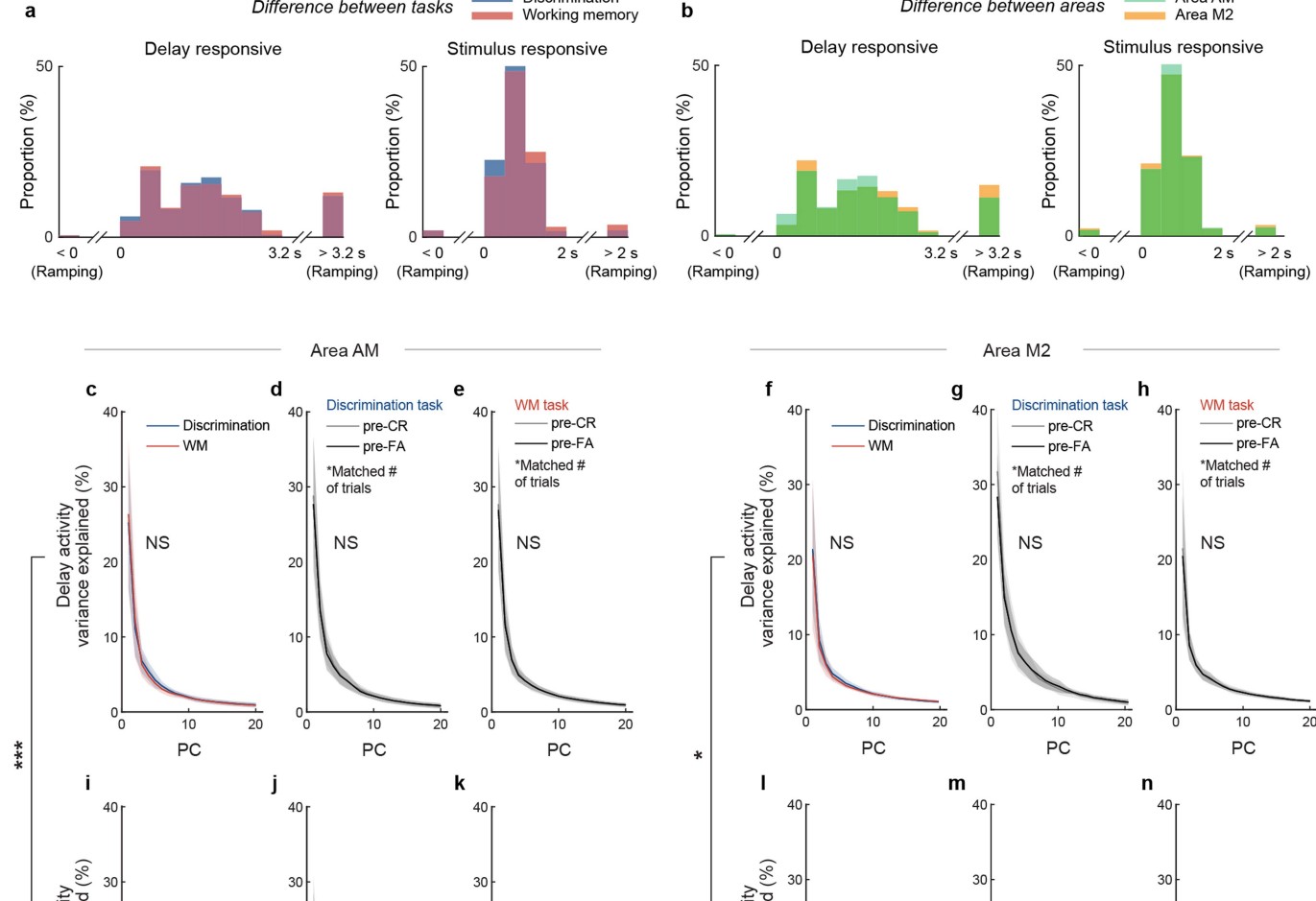

**Extended Data Fig. 5 | Temporal profiles and dimensionality of neural responses are not influenced by working memory engagement.**
**a**, The Discrimination task (blue) and the WM task (red) trial-averaged single-cell response latencies of delay-responsive cells (see Methods), binned at 400 ms, for the delay (left, $n = 805$ cells) and stimulus (right, $n = 465$ cells) evoked responses. Mean latencies outside of the delay or stimulus periods represented ramping responses and were binned separately. **b**, Same as in **a**, but comparing the response latencies of areas AM (teal) and M2 (orange). **a**–**b**, A two-way ANOVA with task and area conditions did not identify a significant difference in response latencies across tasks in their delay responses ($p = 0.43$) or stimulus responses ($p = 0.07$), a small but significantly later response latency for area M2 for the delay responses ($p = 0.01$), no significant difference between areas for stimulus responses ($p = 0.66$), and no significant interaction between the area and task conditions in either the delay ($p = 0.86$) or stimulus responses ($p = 0.50$). **c**–**e**, The eigenspectrum of the covariance of the population activity during the inter-stimulus delay period, shown as the variance explained by the first 20 PCs. PCs were calculated separately for individual experiments similarly to Fig. 3b (see Methods). Thick line is the average and shaded region is the 95% CI across experiments ($n = 18$). **c**, Trials split by task (Discrimination task in blue, WM task in red). There was no significant difference between tasks in the variance explained by the first PC ($p = 0.46$, two-sided signed-rank test). **d**–**e**, Trials split by behavioural response

to subsequent stimulus during the Discrimination (**d**) and WM (**e**) tasks (CR grey, FA black). CR trials were randomly selected to match the number of FA trials per experiment, and this was permuted 100 times and the results averaged per experiment. There was no significant difference between tasks in the variance explained by the first PC in either task ($p = 0.33$ and $p = 0.54$ for Discrimination and WM task, respectively, two-sided signed-rank test). **f**–**h**, as in **c**–**e**, but for area M2 experiments ($n = 13$, $p = 0.16$, $p = 0.19$, and $p = 0.77$, for differences between task, behavioural response during Discrimination task, and behavioural response during WM task, respectively). **i**–**k**, as in **c**–**e**, but for population activity during the stimulus ($n = 18$ experiments, $p = 0.68$, $p = 0.64$, $p = 0.85$, differences between task, behavioural response during Discrimination task, and behavioural response during WM task, respectively). **l**–**n**, as in **f**–**h**, but for population activity during the stimulus ($n = 13$ experiments, $p = 0.85$, $p = 0.42$, $p = 0.42$, differences between task, behavioural response during Discrimination task, and behavioural response during WM task, respectively). **c**–**n**, Delay activity was significantly lower dimensional than stimulus activity in area AM (26% variance versus 19% variance explained, respectively, by the first PC, averaged across 18 experiments, $p = 1.83 \times 10^{-4}$, two-sided signed-rank test) and area M2 (20% variance versus 14% variance explained by the first PC, respectively, averaged across 13 experiments, $p = 2.44 \times 10^{-2}$, two-sided signed-rank test).

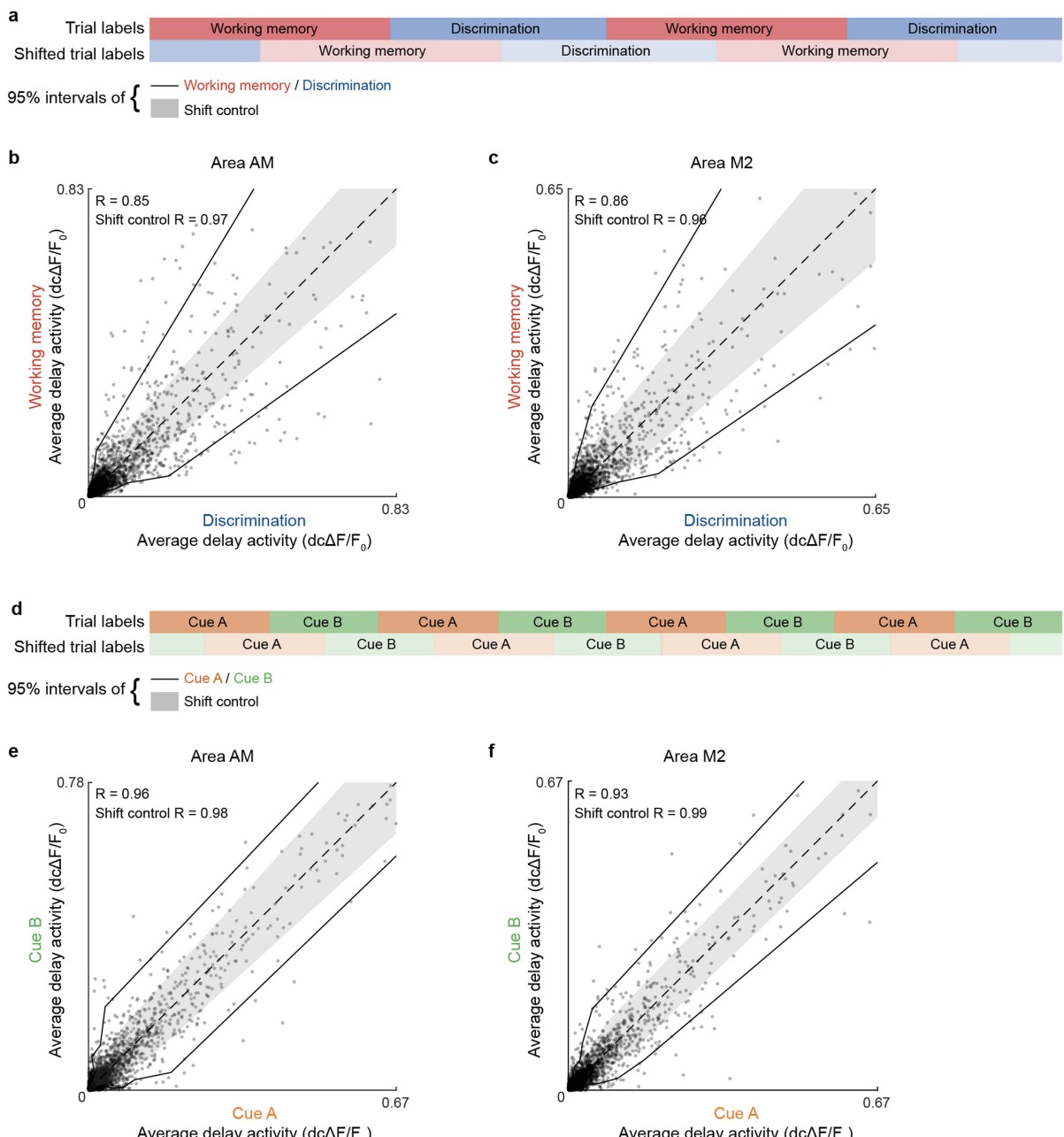

**Extended Data Fig. 6 | Dispersed populations of cells are modulated by working memory. a**, A schematic for generating the null distribution of trial-averaged delay period activity in the Discrimination and WM tasks. The task labels of all trials (top row) were shifted halfway through to the next task block (bottom row), such that the shifted identities maintained the temporal structure of the blocked task design but abolished the task-related changes in activity. **b**, Deconvolved and delay-averaged single-cell activities of all active cells from area AM ($n$ = 5,168 pooled from 7 mice), split by task. Solid lines encompass 95% of the cells in different delay activity bins (binned into 20 equal portions of cells). Shaded regions encompass a similar 95% of cells but with their activities calculated with the shifted task labels. **c**, Same as in **b**, for area M2 ($n$ = 3,794 cells from 7 mice). **d**–**e**, Same as in **a**–**c**, but contrasting the average delay activity in the WM task following a left leaning cue A or right leaning cue B stimulus.

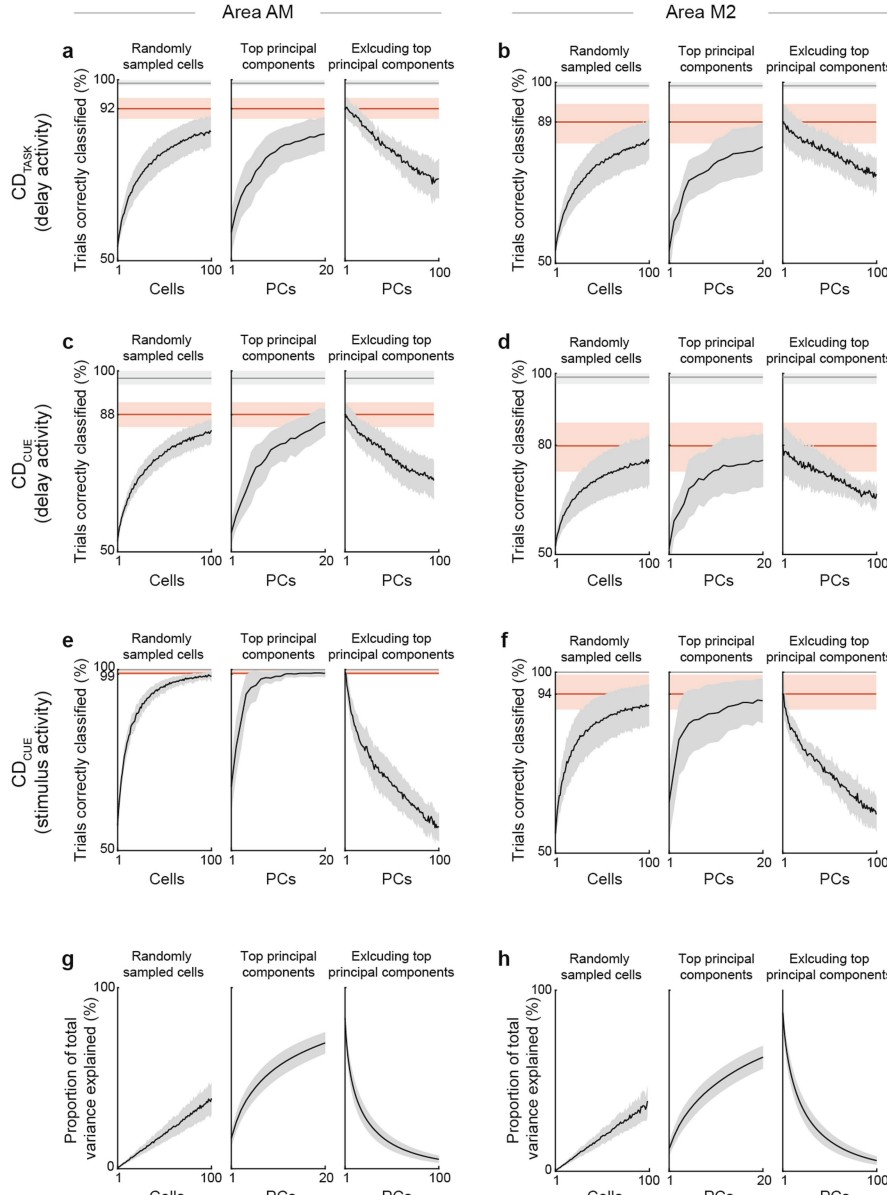

**Extended Data Fig. 7 | High-dimensional embedding of working memory representations. a**, The embedding dimensionality of the $CD_{TASK}$ averaged over all area AM experiments ($n = 18$ from 7 mice). Red horizontal lines are the mean cross-validation test accuracies of decoding the task identity from the $CD_{TASK}$ using population activity during the inter-stimulus delay period, and grey horizontal lines are the corresponding training accuracies (>98% for all data shown). Black lines are the mean test accuracies of decoding while limiting the data available to the decoder to a number of randomly sampled cells (left), the top principal components (centre), or having the top principal components

excluded (right). The $CD_{TASK}$ was recalculated for each data point (i.e. addition or removal of a cell or PC). Shaded regions are the 95% CI across experiments. **b**, As in **a**, but for area M2 experiments ($n = 13$ from 7 mice). **c**–**d**, As in **a**–**b**, but for the $CD_{CUE}$. **e**–**f**, As in **c**–**d**, but for the $CD_{CUE}$ calculated from the neural activity during the stimulus presentation (i.e. sensory responses). **g**–**h**, As in **a**–**b**, but showing the variance explained by the data made available to the decoders in the previous plots (i.e. 100% corresponds to the data used for the horizontal grey and red lines above).

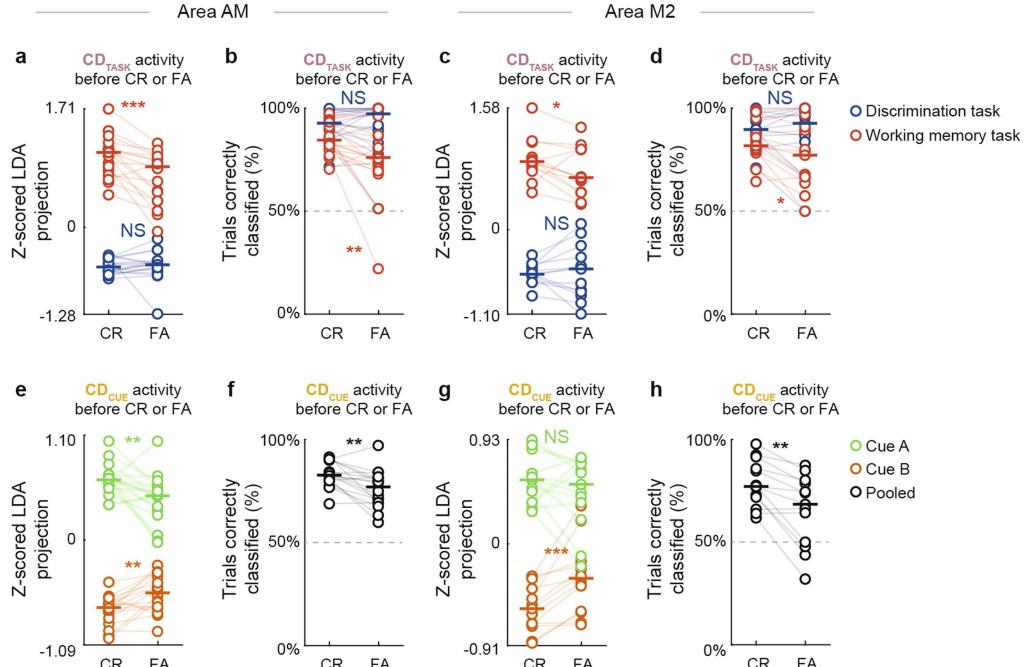

**Extended Data Fig. 8 | Working memory subspace activity predicts correct behavioural responses during the WM task. a**, Z-scored $CD_{TASK}$ delay activity of individual experiments from area AM ($n = 18$), split by behavioural outcome (CR and FA), and averaged across Discrimination task (blue) and WM task (red) trials ($p = 6.29 \times 10^{-4}$ for the WM task, $p = 0.21$ for the Discrimination task, two-sided signed-rank test). Horizontal bars represent medians. **b**, Proportion of trials with their task correctly classified using $CD_{TASK}$ delay activity prior to CRs or FAs, as in Fig. 3e, plotted for individual experiments from area AM ($n = 18$, $p = 5.68 \times 10^{-3}$ for the WM task, $p = 0.79$ for the Discrimination task, two-sided signed-rank test). **c–d**, as in **a–b**, for area M2 experiments ($n = 13$; z-scored $CD_{TASK}$ delay activity differences $p = 4.79 \times 10^{-2}$ and $p = 0.79$ for WM and

Discrimination task trials, respectively, and task classification accuracy differences $p = 1.34 \times 10^{-2}$ and $p = 0.91$ for WM and Discrimination task trials, respectively, two-sided signed-rank test). **e–h**, As in **a–d**, for $CD_{CUE}$ activity during the WM task. **e–f**, For area AM experiments ($n = 18$; z-scored $CD_{CUE}$ delay activity differences $p = 2.47 \times 10^{-3}$ and $p = 7.30 \times 10^{-3}$ for cue A and cue B trials, respectively, and cue classification accuracy difference $p = 2.13 \times 10^{-3}$, two-sided signed-rank test). **g–h**, For area M2 experiments ($n = 13$; z-scored $CD_{CUE}$ delay activity differences $p = 2.44 \times 10^{-4}$ and $p = 0.19$ for cue A and cue B trials, respectively, and cue classification accuracy difference $p = 1.22 \times 10^{-3}$, two-sided signed-rank test).

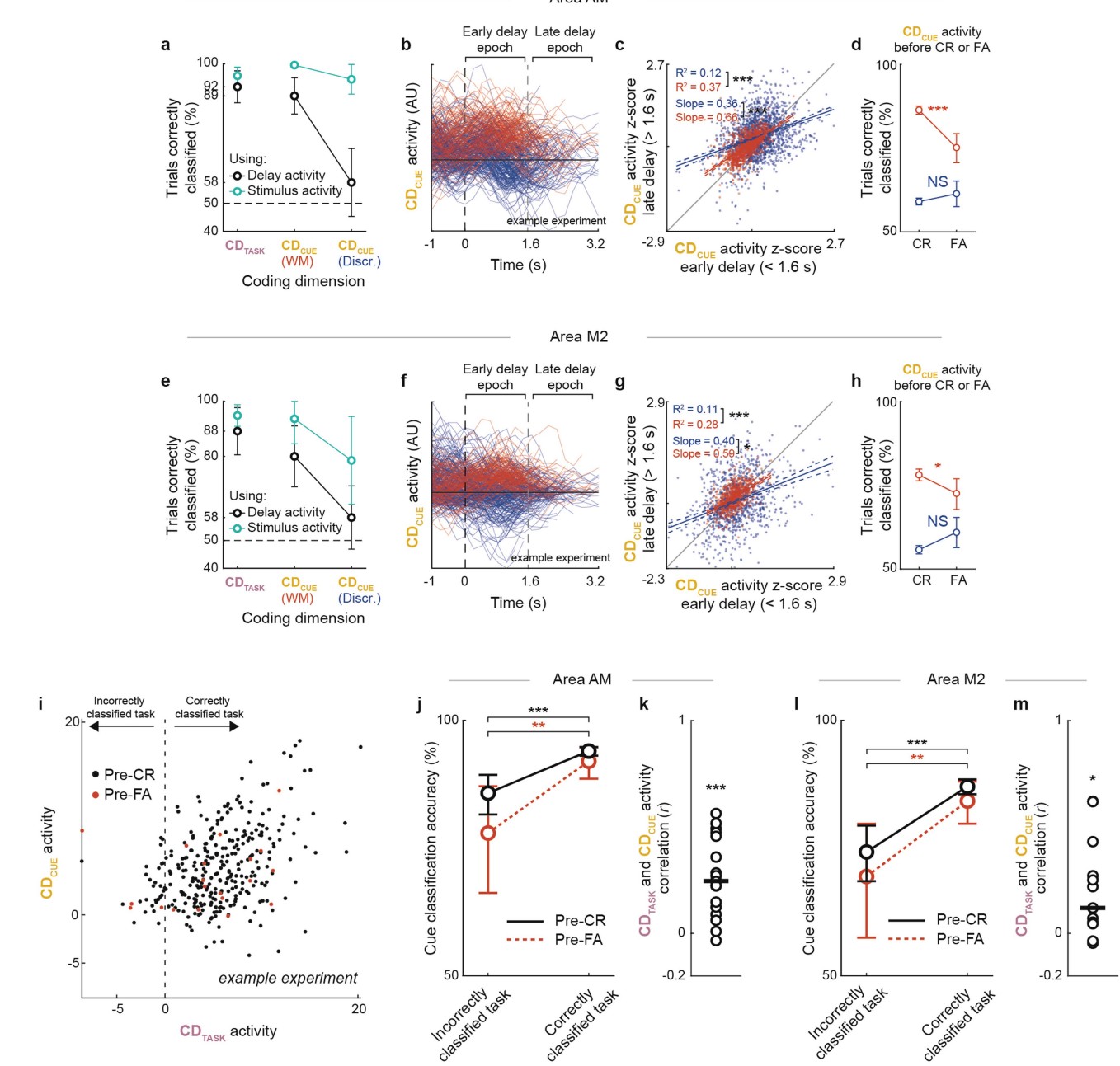

**Extended Data Fig. 9 | See next page for caption.**

**Extended Data Fig. 9 | Cue representations during the delay are selective to the WM task. a**, The percentage of trials which had their task (left column) or cue (middle and right columns) correctly classified using $CD_{TASK}$ and $CD_{CUE}$ activity, respectively. Classification was performed using activity from the trials' delay periods (black) or stimulus periods (turquoise). Trials for task classification were taken from both the Discrimination and WM tasks, as in Fig. 3, and trials for cue classification were split into left-out Discrimination task trials (right column) and the orientation-matched WM task trials (middle column) classified using the same training trials to identify $CD_{CUE}$ during the WM task. Average classification accuracy is shown for area AM experiments ($n = 18$). Error bars represent 95% CI across experiments. **b**, Single-trial population activity of an example experiment from area AM, aligned to the onset of the delay, and projected onto the $CD_{CUE}$. Trials are split into Discrimination task trials (blue lines) and the orientation-matched WM task trials (red lines). Only trials with sufficiently long delay periods ($\geq 2$ s) are shown. **c**, Z-scored single-trial $CD_{CUE}$ projections of area AM activity in the second half (>1.6 s) of the delay period plotted against the first half (<1.6 s) of delay period (i.e. each point is one trial). Trials were split by task (Discrimination task trials in blue, orientation-matched WM task trials in red). Only trials with sufficiently long delay periods ($\geq 2$ s) were included in this analysis ($n = 2,921$ trials collected from 18 experiments). Solid and dashed lines represent the slopes and 95% CI of a fit regression model. The models' coefficients of determination ($R^2$) and slopes, and their differences between the two tasks ($p = 9.18 \times 10^{-10}$ and p $= 2.80 \times 10^{-7}$, respectively, difference greater than zero, two-sided one-sample $t$-test), are printed in the top left. **d**, The proportion of cue trials in which the delay $CD_{CUE}$ activity correctly classified the preceding cue stimulus, split by task. Delay periods were split based on whether they preceded a correct rejection (CR) or false alarm (FA) to the subsequent cue ($p = 2.43 \times 10^{-8}$, $n = 3,580$ trials for the WM task, $p = 0.25$, $n = 9,649$ trials for Discrimination task, two-sided Fisher's exact test). Error bars represent 95% CI. **e–h**, As in **a–d**, but for area M2 experiments. **g**, $n = 1,882$ trials from 13 experiments. Model coefficients of determination ($R^2$) and slopes differences between the two tasks ($p = 2.88 \times 10^{-4}$ and $p = 2.30 \times 10^{-2}$, respectively, difference greater than zero, two-sided one-sample $t$-test). **h**, Pre-CR and pre-FA difference, $p = 3.04 \times 10^{-2}$ for Discrimination task and $p = 2.10 \times 10^{-2}$ for WM task, $n = 6,961$ and $n = 2,575$ trials from 13 experiments, two-sided Fisher's exact test. **i**, Delay period averaged population activity, per WM task trial, from an example experiment in area AM, projected onto the $CD_{TASK}$ (horizontal axis) and $CD_{CUE}$ (vertical axis). Black points represent individual trials' delay period activity prior to CRs and red points prior to FAs. $CD_{TASK}$ and $CD_{CUE}$ were identified independently for each trial using all remaining trials (i.e. leave-one-out cross-validated). Points below zero correspond to trials in which the task (horizontal axis) or cue (vertical axis) was incorrectly classified. **j**, The prediction of preceding cues from the population activity during the delay (measured as classification accuracy, i.e. the percentage of trials below zero on the vertical axis of **i**), when the Task was incorrectly (left column) or correctly (right column) classified, for all trials pooled from area AM experiments ($n = 3,452$ CR trials and $n = 405$ FA trials pooled from 18 experiments; $p = 3.15 \times 10^{-7}$ and $p = 1.74 \times 10^{-3}$ for CR and FA trials, respectively, two-sided Fisher's exact test). Error bars represent 95% CI. **k**, The Pearson's correlation coefficient ($r$) between $CD_{TASK}$ and $CD_{CUE}$ delay activity, per area AM experiment ($n = 18$; greater than zero, $p = 2.80 \times 10^{-4}$, two-sided one-sample signed-rank test). Horizontal bar represents median. **l**, As in **j**, but for area M2 experiments ($n = 2,378$ CR trials and $n = 395$ FA trials pooled from 13 experiments; $p = 2.07 \times 10^{-7}$ and $p = 2.21 \times 10^{-3}$ for CR and FA trials, respectively, two-sided Fisher's exact test). **m**, as in **k**, but for area M2 experiments ($n = 13$, $p = 1.71 \times 10^{-2}$, two-sided one-sample signed-rank test).

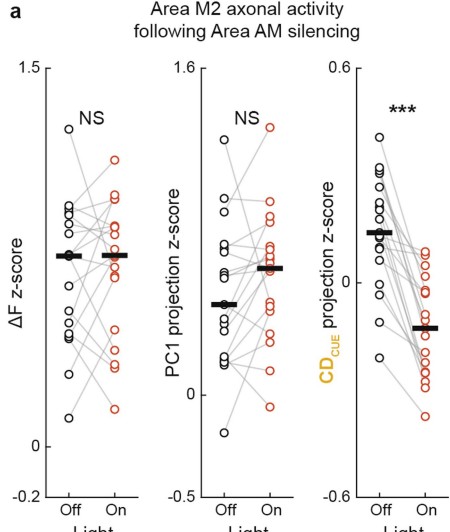

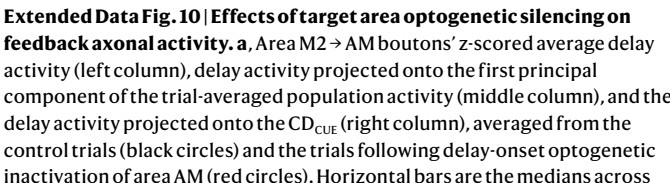

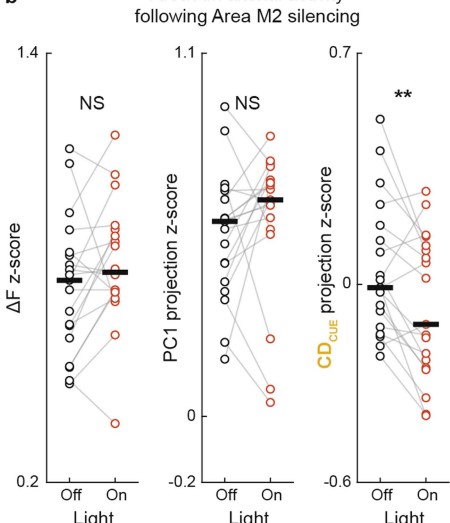

**a** Area M2 axonal activity following Area AM silencing

**b** Area AM axonal activity following Area M2 silencing

**Extended Data Fig. 10 | Effects of target area optogenetic silencing on feedback axonal activity. a**, Area M2 → AM boutons' z-scored average delay activity (left column), delay activity projected onto the first principal component of the trial-averaged population activity (middle column), and the delay activity projected onto the $CD_{CUE}$ (right column), averaged from the control trials (black circles) and the trials following delay-onset optogenetic inactivation of area AM (red circles). Horizontal bars are the medians across experiments. Average delay activity $p = 0.72$, first principal component activity $p = 0.09$, and $CD_{CUE}$ activity $p = 1.32 \times 10^{-4}$, two-sided signed-ranked test, $n = 19$ experiments from 4 mice. **b**, As in **a**, but for experiments recording area AM → M2 bouton activity while silencing area M2. Average delay activity $p = 0.10$, first principal component $p = 0.31$, and $CD_{CUE}$ activity $p = 1.00 \times 10^{-2}$, two-sided signed-ranked test, $n = 19$ experiments from 3 mice.

# Reporting Summary

## Statistics

For all statistical analyses, confirm that the following items are present in the figure legend, table legend, main text, or Methods section.

| n/a | Confirmed | |
|---|---|---|
| ☐ | ☒ | The exact sample size (*n*) for each experimental group/condition, given as a discrete number and unit of measurement |
| ☐ | ☒ | A statement on whether measurements were taken from distinct samples or whether the same sample was measured repeatedly |
| ☐ | ☒ | The statistical test(s) used AND whether they are one- or two-sided *Only common tests should be described solely by name; describe more complex techniques in the Methods section.* |
| ☐ | ☒ | A description of all covariates tested |
| ☐ | ☒ | A description of any assumptions or corrections, such as tests of normality and adjustment for multiple comparisons |
| ☐ | ☒ | A full description of the statistical parameters including central tendency (e.g. means) or other basic estimates (e.g. regression coefficient) AND variation (e.g. standard deviation) or associated estimates of uncertainty (e.g. confidence intervals) |
| ☐ | ☒ | For null hypothesis testing, the test statistic (e.g. *F*, *t*, *r*) with confidence intervals, effect sizes, degrees of freedom and *P* value noted *Give P values as exact values whenever suitable.* |
| ☒ | ☐ | For Bayesian analysis, information on the choice of priors and Markov chain Monte Carlo settings |
| ☒ | ☐ | For hierarchical and complex designs, identification of the appropriate level for tests and full reporting of outcomes |
| ☐ | ☒ | Estimates of effect sizes (e.g. Cohen's *d*, Pearson's *r*), indicating how they were calculated |

*Our web collection on statistics for biologists contains articles on many of the points above.*

## Software and code

Policy information about availability of computer code

| Data collection | LabVIEW 2013<br>ScanImage 2019 |
|---|---|
| Data analysis | MATLAB 2020a<br>CaImAn (1.8.3)<br>Suite2p (0.7.1)<br><br>The analysis code is publicly available at: www.github.com/ivan-voitov/loops. |

For manuscripts utilizing custom algorithms or software that are central to the research but not yet described in published literature, software must be made available to editors and reviewers. We strongly encourage code deposition in a community repository (e.g. GitHub). See the Nature Portfolio guidelines for submitting code & software for further information.

## Data

Policy information about availability of data

All manuscripts must include a data availability statement. This statement should provide the following information, where applicable:
- Accession codes, unique identifiers, or web links for publicly available datasets
- A description of any restrictions on data availability
- For clinical datasets or third party data, please ensure that the statement adheres to our policy

The data that support the findings of this study are available from the corresponding author upon reasonable request.

March 2021

# Field-specific reporting

Please select the one below that is the best fit for your research. If you are not sure, read the appropriate sections before making your selection.

☒ Life sciences        ☐ Behavioural & social sciences        ☐ Ecological, evolutionary & environmental sciences

For a reference copy of the document with all sections, see nature.com/documents/nr-reporting-summary-flat.pdf

# Life sciences study design

All studies must disclose on these points even when the disclosure is negative.

| | |
|---|---|
| Sample size | Data was collected from 23 animals. When possible, given the limitations such as viral vector mediated gene expression, cranial window implant condition, etc., animals were recorded from in multiple sessions. No sample size calculation was performed prior to the start of the experiments. Sample sizes were progressively increased until the onset of the COVID pandemic, when all experiments were halted. |
| Data exclusions | Animals which entered behavioural training but did not achieve adequate behavioural performance, as measured by trial completion, were excluded from all further study. Animals with cranial window implants which were in poor condition (e.g.exhibited dura regrowth) were excluded from further study. |
| Replication | Reproducibility was confirmed by having multiple animals in each experimental condition.<br><br>Multi-area optogenetic silencing experiments (Fig. 1) were replicated in 9 mice.<br>Cell-body imaging experiments (Fig. 2 and Fig. 3) were replicated in 7 mice.<br>Axonal imaging and silencing experiments (Fig. 4) were replicated in 7 mice.<br><br>All attempts at replication were successful, given the data exclusion criteria outlined above.<br><br>Preliminary data was collected at another institute in another country (Biozentrum, University of Basel). |
| Randomization | There were no comparisons across sampled animal populations in this study. All animals used in this study were chosen as the first available from the breeding populations at the local animal facility. |
| Blinding | There were no explicit control groups in this study (all experimental controls were within-animal). |

# Reporting for specific materials, systems and methods

We require information from authors about some types of materials, experimental systems and methods used in many studies. Here, indicate whether each material, system or method listed is relevant to your study. If you are not sure if a list item applies to your research, read the appropriate section before selecting a response.

### Materials & experimental systems

| n/a | Involved in the study |
|---|---|
| ☒ | ☐ Antibodies |
| ☒ | ☐ Eukaryotic cell lines |
| ☒ | ☐ Palaeontology and archaeology |
| ☐ | ☒ Animals and other organisms |
| ☒ | ☐ Human research participants |
| ☒ | ☐ Clinical data |
| ☒ | ☐ Dual use research of concern |

### Methods

| n/a | Involved in the study |
|---|---|
| ☒ | ☐ ChIP-seq |
| ☒ | ☐ Flow cytometry |
| ☒ | ☐ MRI-based neuroimaging |

## Animals and other organisms

Policy information about studies involving animals; ARRIVE guidelines recommended for reporting animal research

| | |
|---|---|
| Laboratory animals | All animals were mus musculus, C57BL/6J background. 10 male and 13 female, 8 to 16 weeks in age.<br>9 animals were PV-Cre × Ai32 mice (JAX 017320 and JAX 024109, Jackson Laboratory).<br>1 animal was a wild-type mouse (Charles River).<br>3 animals were Ai-148 × Cux-creER mice (JAX 030328 and JAX 012243, Jackson Laboratory).<br>3 animals were Ai-148 mice (JAX 030328, Jackson Laboratory).<br>7 animals were PV-Cre mice (JAX 017320, Jackson Laboratory). |
| Wild animals | There were no wild animals used in this study. |
| Field-collected samples | No field-collected samples were used in this study. |

Ethics oversight | All experiments were performed under the UK Animals (Scientific Procedures) Act of 1986, under project license PPL PD867676F, following local ethical approval by the Sainsbury Wellcome Centre Animal Welfare Ethical Review Body.

Note that full information on the approval of the study protocol must also be provided in the manuscript.

