## [Peer Review File · Nature]

Manuscript Title: Cortical feedback loops bind distributed representations of working memory

Reviewer Comments & Author Rebuttals

Reviewer Reports on the Initial Version:

Referees' comments:

Referee #1 (Remarks to the Author):

This study provides evidence for high-dimensional representation of working memory (WM) that is maintained across distributed cortical regions in mice. The authors develop a novel behavioral task that requires mice to switch between a visual discrimination task and a delayed non-match-to-sample task with similar behavioral statics that allow dissociation of WM from motor preparation and reward expectation. Focal optogenetic inactivation of visual areas and M2 during WM reveal distributed involvement of these cortical regions. 2-photon calcium imaging shows that WM information is embedded in high-dimensional representations, which differ from visual representations during stimulus presentation. Simultaneous optogenetic inactivation and axonal imaging show that WM representations are impacted by silencing of specific cortical regions within reciprocally coupled corticocortical loops, suggesting that reciprocal interactions between regions are required to maintain WM.

This is an exciting and technically impressive study that combines several cutting-edge approaches in a well-controlled behavior to gain insights into the network mechanisms of WM. Experiments and analyses are carefully done. Statistics are appropriate and descriptions of p values/error bars are accurate. The study is significant because it provides evidence for potential alternative mechanisms of WM (differing from classic persistent activity dynamics) reflected in high-dimensional dynamics that require distributed cortical regions. The results are powerful because they demonstrate a link between the discovered WM activity dimensions and behavior, even under conditions where activity is optogenetically perturbed. I have a few comments which could be addressed through additional analysis.

Major:

1) The nature of WM. A key premise of the interpretation is that the WM engaged in this study is the "maintenance of mnemonic sensory information". However, what information is maintained in WM is not completely clear from the analysis.

a. Surprisingly, the authors found that top activity modes in visual areas and M2 do not differentiate WM and discrimination task. Instead, the difference in delay activity is found in high-dimensional dynamics (quantified as activity projections on a task dimension, CD_task). However, the across-task comparison of activity dimensions encoding the stimulus, CD_cue, is not examined. Is the stimulus information present during the delay in the discrimination task, but just orthogonal to task dimension (as implied in Fig 3a)? If stimulus information is equally persistent during the delay and

the activity in the two tasks only differ in the task dimension, this could warrant a different interpretation.

b. If the delay activity in WM differs between correct reject and false alarm (FA) (suggested by Fig 3k and n), the activity difference may reflect a degradation of stimulus information maintenance or reflect an impending motor response. Can the authors discuss whether the two possibilities can be differentiated?

2) Interpretation of FA in WM task. In the behavioral experiments (Fig 1), a key behavioral feature of WM is an increase in FA rate with increased delay duration, which implies a degradation of WM maintenance over time. However, the stimuli used in the WM task (+45 and -45 deg) are the rewarded stimuli from the discrimination task. This presents a potential confound, because another interpretation could be that mice are progressively more likely to respond in blocks where a rewarding stimulus is repeatedly presented. Can the data dissociate WM degradation from an urge to respond to the rewarding stimulus? One possible solution could be to examine the WM task performance when an unrewarded stimulus is used as the cue. For example, the behavioral data from imaging experiments using rotated visual stimuli, which use non-rewarded stimuli as cues in some blocks, could clarify this question.

3) High-dimensional WM representation. The finding that informative dimensions for task and stimulus decoding are higher during WM than during stimulus presentation is very interesting (Fig 3). A few points could be better clarified.

a. How are the principal components and variance explained calculated in Fig 3b? Presumably these are calculated based on single trial activity, but the description is missing.

b. Does the higher dimensionality result from single cells responding sparsely and in uncorrelated manner as suggested by the activity-silent WM storage hypothesis? This is implied in several places in the text but could be better shown or explained.

c. The results show that stimulus information is maintained in high-dimensional dynamics during WM and collapsed activity along CD_cue predicts behavioral errors. Is the collapsed activity due to a reduction in dimensionality of delay activity? Related, does the dimensionality of stimulus activity differ between the WM and discrimination task?

Minor:

1) Do the authors have an explanation for why M2 inactivation only transiently affects behavior, but AM inactivation induces persistent effect? Slower timescale activity is usually associated with frontal cortical regions. Here, it seems the results break that expectation.

2) For axon imaging + target-region inactivation experiment, is the disruption of activity limited to CD_cue or does the inactivation also affect the CD_task activity?

3) Line 352: typo. "... following area AM inactivation (Fig. 4k)". This is likely referring to Fig 4e.

4) Line 432-434, “The selective disruption of working memory representations in corticocortical feedback projections by the inactivation of ongoing feedforward activity provides direct evidence for the mechanistic role of functional cortical feedback loops for the maintenance of internally generated cognitive representations.” I did not fully follow this conclusion. The results show that the inactivated cortical region plays a role in maintaining WM, and the inactivation also affects downstream cortical regions that project back to the inactivated region, which suggests that their interactions are required. But it is not clear if the affected cortical feedback signal itself is required for WM maintenance. Can the authors more clearly explain their interpretation?

Referee #2 (Remarks to the Author):

This study is an excellent combination of a unique, highly challenging behavioral paradigm and best-of-class physiology and activity perturbation. In brief, the authors designed a novel task, in which mice alternated between a working memory task and a discrimination task allowing to match stimuli, movement and reward statistics across tasks. This novel task helps disentangle neural representations of working memory from other behavioral variables. The authors carried out multiple perturbation experiments and showed that working memory is distributed across areas of neocortex. Simultaneous axonal recordings during inactivation of a distal cortical area further shed light on the role of interareal reciprocal interactions in the maintenance of working memory. This is highly valuable, important and timely work whose results will be of great interest to the neuroscience community. I feel that some analyses could be performed in a more straightforward way and more clearly presented, which would further improve the manuscript.

Major comments:

1. The task is great. Very impressive.

2. In the analysis of population data, the use of “high-dimensional representations” and the related conceptual model was confusing to me. “High dimensional representations” could mean representations that live in higher dimensions, or it could mean representations that are themselves high dimensional. It felt to me that on occasion the manuscript switches between those two meanings. To step back, there are two clear findings: (i) Task related information is present in population activity. (ii) Using only the first PCs that capture a large part of the variance is insufficient to fully capture this information. What this means is that the task related information is present in dimensions of activity space that don’t capture much variance. In their decoding analysis the authors find that a one dimensional decoder performs well. Therefore, the task-related dynamics themselves are not necessarily high dimensional. As an example, one can have a model in which the task-relevant dynamics are one (or low) dimensional and orthogonal to another set of dynamics that are not task relevant and capture a large amount of variance, which would be inline with the findings. The third finding is that a large number of neurons is needed to arrive at good decoding. It is related to dimensionality but not in a very straightforward way since the representation may be low dimensional yet a large number of neurons may just be necessary to estimate the dynamics in the low variance part of the population activity that allows to decode working memory. I believe the

authors have a different model in mind, one in which (I state it here simplified) working memory is distributed across many neurons, but a neuron doesn't have information in every trial, but rather only in a small subset of trials. In this scenario, trial-averaging would average away this decoding contribution, pushing it into low variance components and unless one has access to many neurons, most trials will be un-decodable, consistent with their findings (as a side note there is a potential problem in this model: since the decoder is fixed it has to assign the same weight to a neuron whether it is in a trial in which that neuron happens to be informative or not, but if the signal in informative trials is strong enough the decoder will still work). I think it is hard (and unnecessary) to argue for this model and it is best to revise statements in the text to clarify the specific findings and the possible interpretations more generally, and not just in relation to this model.

3. In terms of the CD_task, I also feel that the text and analysis follow one particular interpretation for these dynamics when the full picture might be more complex. The authors state that it "captures the activity introduced by working memory engagement by contrasting the delay activity in the two task blocks following identical sensory input". While this is a possible interpretation, another possible interpretation is that there is a context-signal that "tells" the circuit which task it is in, unrelated to the specific computations necessary for remembering the cue. The way CD_task was defined may in fact favor such a signal, as memory of the cue may be present in the discrimination blocks of the task despite the fact that it isn't behaviorally useful. I realize that the memory cannot be quite the same due to the differential results of length-of-delay on behavior and the differential perturbation effects (and that the projection has complex dynamics), but these results are not enough to justify assuming no memory. The reason it is potentially important to consider these different scenarios is in the interpretation of projections on CD_task. This interpretation differs whether one has in mind a context-signal or a reflection of the actual process of working memory. One way to get more quantitative purchase on this question is to directly test differences in the preservation of cue information through the delay epoch between the Discrimination task and the WM task (see also comment 7 below). Specifically, one possible analysis could be calculating a CD_cue that contrasts between the -15° cue and $+15^\circ$ cue in the Discrimination task and then compare the activities in their respective CD_cue in the Discrimination task and WM task. This comparison can be done for CD_cue both at the delay epoch and at the stimulus period. It would be interesting to see whether there is any consistent difference, for example, activity in the discrimination blocks along CD_cue may decay faster than in the WM task. Or perhaps there is no significant difference between the blocks which would suggest that working memory is also maintained in the Discrimination task, despite it being unnecessary for successful completion of the trial.

4. The presentation of the results of the population analysis could be made more straightforward. In particular, Fig. 3e, k, h, n are the main quantitative findings in their section and are in my opinion unnecessarily confusing. The text states: "We observed that in both areas AM and M2, disruption of the CD_cue activity or CD_task activity during the WM task, as measured by the failure to correctly classify a given trial's cue or task identity, was predictive of incorrect responses to the subsequent stimulus (Fig. 3e, h, k, n,)." It is possible I missed something, but the way the data is summarized in these plots seems unnecessarily difficult to absorb. Since the cue direction is built as a decoder of cue, the state of the decoder (on one side or another of an assumed threshold) should relate to the represented cue and a switch to the other side of the threshold should correlate with errors. It

therefore seems more straightforward to directly compare the state of the decoder to the action at the end of the delay. While there is an imbalance in trial types since 80% of the trials are match trials, the analysis still seems relevant in its simpler form. The current choice of comparing decoder accuracy separately in CR and FA trials and the claim that a reduction is expected seems a very complex presentation of this statement. I believe the idea is similar, that a mistake in decoder indicates a switch to the opposite cue and this should happen more in FA trials, and a decrease in accuracy means more errors, but the multiple inversions make it harder to understand, in my opinion. Especially given the text which conflates the activity and the decoder and uses the word disruption which can be easily confused with an active perturbation. In addition, the use of “preceding” in the axis labels is ambiguous. It could mean that the preceding trial was a CR or FA, not that the delay period preceded a CR or FA outcome.

5. The analysis of the time course of single trials in lines 296-308 is presented as highly informative of activity underlying working memory. “The slope of this relationship is indicative of the persistence of working memory representations over the course of the delay, and the R2 is indicative of their robustness over time”. While it is true that single-trial dynamics during the delay period are interesting, and the slope and R2 of a comparison between the first and second part of the projection on a decoder are relevant metrics, they are not as central as presented. For instance, in a straightforward decoding model of a comparison to threshold, the decoder projection values can become completely scrambled between the first and second part of the delay period (yielding zero R2 between the first and second parts of the delay period), yet as long as each trial doesn’t cross the threshold there is no loss in performance. It is possible that I misunderstood the analysis, and the way the different trial types were treated means that a zero R2 would necessarily mean complete mixing between condition types, but I don’t think that is the case given the way data is plotted in Fig. 3 (see also next comment). If this is true then there are simpler ways to show and describe this effect.

6. In general, it was difficult for me to follow the precise details of the population analysis. The first part of the analysis is based on a pooled pseudo-population, with trial-averaging, but then the analysis switches to single-trial dynamics and decoding (with no trial averaging). The mechanics of how single trial analysis relates to sessions (each with different trial numbers) wasn’t clearly described. I believe the authors used single session analysis that leaves one trial out and then just pooled these left out trials across all sessions, but it wasn’t clear whether the values given would be changed if averages across sessions are taken, and I am not sure I correctly followed what was performed. Clearer descriptions in the main text would be useful as well as longer explanations in the methods (and/or putting code in a code repository).

7. Regarding the point that dynamics in the high variance components of the two task blocks are very similar, one possibility (as alluded to in comment 3) is that since the animals were trained on both tasks, they adopt a strategy in which they memorize the cue in all block types. This puts the experiments in a bit of a catch 22: in order to compare the dynamics one has to train the same animal on both tasks, but if one trains the animal on both tasks, the animal could memorize the cue in all task conditions. In principle this could perhaps be addressed by animals first learning the discrimination task and then adding the working memory task (perhaps also vice versa in separate animals), while performing longitudinal imaging. I believe these experiments would be too time

consuming, especially given that the study already goes above and beyond the standard in terms of behavioral training. I therefore don't suggest the authors actually collect this data, but if the authors have existing data for the discrimination task alone (in separate animals) they could argue for its similarity to the full task data to make the scenario of working memory training changing the dynamics for both block types less plausible. The authors should also raise this option, and/or argue against it, in the discussion section.

8. Going back to the role of CD_task as in comment 3 above, I was wondering whether one can try to analyze trials in which dynamics go into the wrong state in terms of task and see whether there is a specific prediction to be made regarding different relevance for the state of CD_cue, i.e., what should one expect assuming that there might be a mistake in the assignment of the task. For instance, would a crossing of threshold in both CD_task and CD_cue compensate for each other? I believe it is not so simple since the relation between the cue and probe are different in discrimination and WM blocks, but perhaps something can be done. Related to this point, it wasn't clear to me whether CD_task and CD_cue were explicitly orthogonalized or they were just generally found to be orthogonal.

9. In the simultaneous imaging and perturbation of figure 4, even when activity remained perturbed, the 95% CI mostly don't overlap for most of the short delay, in particular for AM (as the shortest short delay is 0.8 seconds). This may imply that one shouldn't expect a behavioral effect. It is possible that the read out is more graded, but it is worth commenting on that. In addition, how the CIs were calculated is not shown.

10. I was confused by how cue rotation figured into the task block structure. If I understood correctly, the mice can switch task blocks quickly because the absence/presence of a vertical grating is informative of the task block (line 793-794). But if the vertical grating can now be rotated and appear in both tasks, do mice recognize the task because there is only one cue type? I was also confused by the statement in line 808-809 that "in between two rotation blocks, the stimulus orientation angles were changed slowly..." Does that mean the angles of the grating changed continuously? It would help to clarify these issues.

11. In the section "Cortical feedback loops maintain distributed working memory representations", it is suggested that the representation of working memory disrupted by distal inactivation might be recovered if the delay is long enough. It would be useful to generate a version of Fig. 1h in which trials are grouped according to delay length (as in Fig. 4m) and check for the effect there as well.

Minor comments:

1. Statistics for behavioral analysis and perturbations were not clear.
2. Line 166 states "... but were more diverse in their temporal profiles (Fig. 2c)". It was not quite clear to me what is meant here. The two blocks in Fig. 2c look quite similar.
3. Line 171: "Surprisingly, however, working memory engagement did not alter the temporal profile of the trial-averaged activity of individual cells" I am not sure what is the precise claim. Do no neurons have a different response, even by chance? Perhaps the statement related to changes above some chance level?
4. Line 225-7: "Furthermore, this analysis did not reveal any clear subpopulations of cells whose

delay activity was selective to either task."What exactly does this mean?

5. Line 313: "(Fig. 3e, h, k, n,)" unnecessary comma at the end.

6. Title EDF 7: "Stimulus rotations ensured identical sensory inputs across the two tasks", correct to sensory.

7. Lines 334-5: I believe a reference to Fig. 4b and Fig. 4h should be added.

8. in Fig. 1a the reward sign only appears in the WM task but not the Discrimination task, while this contingency is simpler, it is still worth having the reward sign.

9. For Fig. 2g and 2i, I expected to see three different trajectories, one for each delay group, but it seems that in the initial part of the trajectories the different delays combine into one line. Is this a plotting effect and there are actually three lines there? Or were they combined artificially? It would be good to clarify in the legends.

10. Line 352: in "...following area AM inactivation (Fig. 4k; ...)". Should "Fig. 4k" be "Fig. 4e"?

11. Line 976: the word "first" is in italics. This seems unnecessary.

12. I found the vertical axes in Extended Data Fig. 10 g and h and their description confusing. Please clarify.

Author Rebuttals to Initial Comments:

We thank the reviewers for their very constructive feedback and great suggestions for further analysis which have helped improve the manuscript by further clarifying the nature of the working memory representations in the neocortex. We have performed all of the suggested analyses and revised our manuscript accordingly, adding several new Extended Data Figures in order to accommodate these analyses.

There was one common query raised by both reviewers regarding the maintenance of sensory representations during the inter-stimulus delay period (major comment 1a from reviewer #1 and major comment 3 from reviewer #2). Specifically, the reviewers asked if it was possible to discern whether the recorded neural populations maintained the sensory information of the previous stimulus even when working memory was not behaviourally necessary (i.e. during the Discrimination task). We addressed this issue by further exploring the Cue related delay activity, operationalized as CD_{CUE} activity, during the Discrimination task, and examining whether, and how, neural representations of the preceding stimuli during the delay periods of the Discrimination task were maintained. We compiled these analyses into a new **Extended Data Fig. 13** (printed on next page) and a new paragraph in the Results section of the manuscript.

We found that the neural population activity discriminating the preceding stimuli in the WM task (CD_{CUE} activity) was largely absent during the delay period of the Discrimination task, with only 58% of trials correctly classified in areas AM and M2 (as opposed to 89% and 80% accuracy in AM and M2, respectively, for matched stimuli during the WM task; black lines in **Extended Data Fig. 13a, e**). Importantly, CD_{CUE} activity failed to maintain sensory information during the Discrimination task even when the preceding stimuli were identical to the ones in the WM task (i.e. comparing the two tasks across opposing rotation blocks, as in **Fig. 3**), and cross-validation ensured that there was no difference between how the Discrimination and WM task trials were treated with respect to the identification of the CD_{CUE} . As a negative control, we observed that the subspace of population activity discriminating the Cues *during* the presentation of the stimuli (*stimulus period* CD_{CUE} ; i.e. the turquoise lines in **Fig. 3a**), identified during the WM task, was able to correctly predict the presentation of the same stimuli during the Discrimination task (95% and 79% classification accuracy for areas AM and M2, respectively; turquoise lines in **Extended Data Fig. 13a, e**). This indicates that, in contrast to the delay period representations of preceding Cues, the sensory representations of the Cues were encoded in a similar manner whether or not the mouse was engaged in the WM task. Furthermore, delay CD_{CUE} activity during the Discrimination task was less robust (measured as the R^2 of the first and second half of the delay activity epochs, $p < 0.001$, in both areas) and decayed faster (measured as the slope of the first and second half delay activity relationship, $p < 0.05$ in area M2 and $p < 0.001$ in area AM), as compared to the delay activity along the same population activity subspace during the WM task, and did not predict correct behavioural responses to the subsequent stimuli during the Discrimination task ($p > 0.05$ in both areas; **Extended Data Fig. 13b-d, f-h**). Together, these results indicate that, during the Discrimination task, even when the delay-stimulus trial structure was identical to the WM task, the neural populations we recorded from did not maintain the representations of previous stimuli during the inter-stimulus delay periods.

The new paragraph in the results section states these results as follows:

“To examine the specificity of working memory representations to the WM task, we explored the maintenance of stimulus information during the Discrimination task delay periods. We found that the neural population activity encoding the preceding Cues during the delay periods of the WM task (CD_{CUE} activity) was largely absent during the Discrimination task, with an average of 58% of trials correctly classified in areas AM and M2 (as opposed to an 89% and 80% classification accuracy in AM and M2, respectively, during the WM task; black lines in Extended Data Fig. 13a, e). As a negative control, we observed that population activity discriminating the Cues *during* their presentation in the WM task (*Stimulus period* CD_{CUE} ; i.e.

the turquoise lines in Fig. 3b), was able to correctly predict the presentation of the same stimuli during the Discrimination task (95% and 79% classification accuracy for area AM and M2, respectively; turquoise lines in Extended Data Fig. 13a, e). This indicates that, in contrast to the delay representations of preceding Cues, the sensory representations of the Cues were encoded in a similar manner whether or not the mouse was engaged in the WM task. Furthermore, delay CD_{CUE} activity during the Discrimination task was less robust (measured as the R^2 of first and second half delay activity epochs, $p < 0.001$ in both areas) and decayed faster (measured as the slope of the first and second half delay activity relationship, $p < 0.05$ in area M2 and $p < 0.001$ in area AM) as compared to the delay CD_{CUE} activity during the WM task, and did not predict correct behavioural responses to the subsequent stimuli during the Discrimination task ($p > 0.05$ in both areas; Extended Data Fig. 13b-d, f-h).”

Extended Data Fig. 13 | Cue representations during the delay were selective to the WM task. **a**, The percentage of trials which had their Task (left column) or Cue (middle and right columns) correctly classified using CD_{TASK} and CD_{CUE} activity, respectively. Classification was performed using activity from the trials’ delay periods (black) or stimulus periods (turquoise). Trials for task classification were taken from both the Discrimination and WM tasks, as in Fig. 3, and trials for Cue classification were split into left-out Discrimination task trials (right column) and the orientation-matched WM task trials (middle column) classified using the same training trials to identify CD_{CUE} during the WM task. Average classification accuracy is shown for area AM experiments ($n = 18$). Error bars represent 95% CI across experiments. **b**, Single-trial population activity of an example experiment from area AM, aligned to the onset of the delay, and projected onto the CD_{CUE} . Trials are split into Discrimination task trials (blue lines) and the orientation-matched WM task trials (red lines). Only trials with sufficiently long delay periods (≥ 2 s) are shown. **c**, Z-scored single-trial CD_{CUE} projections of area AM activity in the second half (>1.6 s) of the delay period plotted against the first half (<1.6 s) of delay period (i.e. each point is one trial). Trials were split by task (Discrimination task trials in blue, orientation-matched WM task trials in red). Only trials with sufficiently long delay periods (≥ 2 s) were included in this analysis ($n = 2,921$ trials collected from 18 experiments). Solid and dashed lines represent the slopes and 95% CI of a fit regression model. The models’ coefficients of determination (R^2) and slopes, and their differences between the two tasks (both $p < 0.001$, difference greater than zero, one-sample t -test), are printed in the top left. **d**, The proportion of Cue trials where the delay CD_{CUE} activity correctly classified the preceding Cue stimulus, split by task. Delay periods were split based on whether they preceded a Correct Rejection (CR) or False Alarm (FA) to the subsequent Cue ($p < 0.001$, $n = 3,580$ trials for the WM task, $p = 0.96$, $n = 5,782$ trials for Discrimination task, Fisher’s exact test). Error bars represent 95% CI. **e-h**, as in **a-d** but for area M2 experiments. **g**, $n = 1,882$ trials from 13 experiments. Model coefficients of determination (R^2) and slopes differences between the two tasks ($p < 0.001$ and $p < 0.05$, respectively, difference greater than zero, one-sample t -test). **h**, Pre-CR and pre-FA difference, $p = 0.33$ for Discrimination task and $p < 0.05$ for WM task, $n = 3,670$ and $n = 2,575$ trials from 13 experiments, Fisher’s exact test.

We have also identified two mistakes in the manuscript figures.

- 1) The figure panels Fig. 2h and Fig. 2j, showing the low-dimensional trajectory separation between the Working Memory and Discrimination tasks over the course of the delay and stimulus periods, were plotted incorrectly. We have updated these two panels. The statistical analyses and the associated figure legends were not affected and remain unchanged.
- 2) The number of significance stars in panel Fig. 4m was incorrect (was 2 instead of 1), for both delay durations in area AM silencing experiments. This has been fixed. The reported statistics and the associated figure legends were correct and remain unchanged.

We address all other reviewer comments below, in order:

Reviewer #1 (Remarks to the Author):

This study provides evidence for high-dimensional representation of working memory (WM) that is maintained across distributed cortical regions in mice. The authors develop a novel behavioral task that requires mice to switch between a visual discrimination task and a delayed non-match-to-sample task with similar behavioral statics that allow dissociation of WM from motor preparation and reward expectation. Focal optogenetic inactivation of visual areas and M2 during WM reveal distributed involvement of these cortical regions. 2-photon calcium imaging shows that WM information is embedded in high-dimensional representations, which differ from visual representations during stimulus presentation. Simultaneous optogenetic inactivation and axonal imaging show that WM representations are impacted by silencing of specific cortical regions within reciprocally coupled corticocortical loops, suggesting that reciprocal interactions between regions are required to maintain WM.

This is an exciting and technically impressive study that combines several cutting-edge approaches in a well-controlled behavior to gain insights into the network mechanisms of WM. Experiments and analyses are carefully done. Statistics are appropriate and descriptions of p values/error bars are accurate. The study is significant because it provides evidence for potential alternative mechanisms of WM (differing from classic persistent activity dynamics) reflected in high-dimensional dynamics that require distributed cortical regions. The results are powerful because they demonstrate a link between the discovered WM activity dimensions and behavior, even under conditions where activity is optogenetically perturbed. I have a few comments which could be addressed through additional analysis.

Major:

1) The nature of WM. A key premise of the interpretation is that the WM engaged in this study is the "maintenance of mnemonic sensory information". However, what information is maintained in WM is not completely clear from the analysis.

a. Surprisingly, the authors found that top activity modes in visual areas and M2 do not differentiate WM and discrimination task. Instead, the difference in delay activity is found in high-dimensional dynamics (quantified as activity projections on a task dimension, CD_task).

However, the across-task comparison of activity dimensions encoding the stimulus, CD_{cue} , is not examined. Is the stimulus information present during the delay in the discrimination task, but just orthogonal to task dimension (as implied in Fig 3a)? If stimulus information is equally persistent during the delay and the activity in the two tasks only differ in the task dimension, this could warrant a different interpretation.

We have further explored the maintenance of prior sensory information during the delays of the Discrimination task in the first part of the rebuttal, see above (addressed together with major comment 3 of reviewer #2). We found that stimulus information during the delay, operationalized as CD_{cue} activity, was largely absent during the Discrimination task.

b. If the delay activity in WM differs between correct reject and false alarm (FA) (suggested by Fig 3k and n), the activity difference may reflect a degradation of stimulus information maintenance or reflect an impending motor response. Can the authors discuss whether the two possibilities can be differentiated?

The presence of premotor activity during the delay, confounding the WM maintenance related activity (i.e. CD_{task} or CD_{cue} activity), and leading to the observed degradation of CD_{task} and CD_{cue} activity prior to false alarm (FA) responses, is indeed an interesting possibility. One observation that speaks against this possibility is that the relationship between CD_{task} and CD_{cue} activity with subsequent FAs was not present during the Discrimination task (Fig. 3e, h, k, n), suggesting that task-agnostic premotor activity is not sufficient to explain this result. We have added a new **Extended Data Fig. 12** to the manuscript which shows this difference more clearly by plotting average trial scores (activity projected onto the CD_{cue} and CD_{task}), per experiment, with statistics calculated accordingly.

Extended Data Fig. 12 | Working memory subspace population activity predicted correct behavioural responses during the working memory task. **a**, Z-scored CD_{task} delay activity of individual experiments from area AM ($n = 18$), split by behavioural outcome (CR and FA), and averaged across Discrimination task (blue) and WM task (red) trials ($p < 0.001$ for the WM task, $p = 0.21$ for the Discrimination task, signed-rank tests). Horizontal bars represent medians. **b**, Proportion of trials with their task correctly classified using CD_{task} delay activity prior to CRs or FAs, as in Fig. 3.e, plotted for individual experiments from area AM ($n = 18$, $p < 0.01$ for the WM task, $p = 0.79$ for the Discrimination task, signed-rank tests). **c-d**, as in **a-b**, for area M2 experiments ($n = 13$; Z-scored CD_{task} delay activity differences $p < 0.05$ and $p = 0.79$ for WM and Discrimination task trials, respectively, and task classification accuracy differences $p < 0.01$ and $p = 0.91$ for

WM and Discrimination task trials, respectively, signed-rank tests). **e-h**, As in **a-d**, for CD_{CUE} activity during the WM task. **e-f**, For area AM experiments ($n = 18$; Z-scored CD_{CUE} delay activity differences $p < 0.01$ for both Cue A and Cue B trials, and Cue classification accuracy difference $p < 0.01$, signed-rank tests). **g-h**, For area M2 experiments ($n = 13$; Z-scored CD_{CUE} delay activity differences $p < 0.001$ and $p = 0.19$ for Cue A and Cue B trials, respectively, and Cue classification accuracy difference $p < 0.001$, signed-rank tests).

To investigate this possibility with a more direct approach, we took a similar decoding strategy to the one used for CD_{CUE} and CD_{TASK} identification in order to find a linear subspace of activity which could predict FAs in either task, and tested if this subspace could generalize across tasks (i.e. identify a premotor signal in the Discrimination task which predicts FAs during the WM task, and vice versa). However, somewhat surprisingly, we were not able to predict the majority of FAs with such a linear decoder (experiment-averaged cross-validated test accuracy of 57% and 60% during the WM task in areas AM and M2, respectively; 50% chance accuracy; **Rebuttal Fig. 1**), although it should be noted that as a minority of trials were FAs in our data, this limited the ability for our models to avoid overfitting, and regularization of the models was required.

Rebuttal Fig. 1 | Predicting upcoming behavioural responses from task-agnostic delay activity. **a**, Decoding of FA responses to stimuli using population activity from area AM experiments ($n = xx$) during the immediately preceding interstimulus delay periods. Circles are individual experiments' average cross-validated classification accuracies. The first column shows decoding results following training of the decoder on Discrimination task trials and testing on Discrimination task trials (leave-one-out), the second column for training and testing on WM task trials, the third column for training on Discrimination task trials and testing on WM task trials, and the fourth column for training on WM task trials and testing on Discrimination task trials. Only training and testing on WM task trials showed a significant decoding performance across experiments ($p < 0.01$, as compared to 50% chance accuracy, signed-rank test). **b**, The eigenspectrum of the covariance of population activity during the delay, prior to CRs (grey line) and FAs (black line), for area AM experiments ($n = 18$, $p = 0.38$, signed-rank test). Shaded regions are the 95% CI across experiments. **c-d**, as in **a-b** for area M2 experiments ($n = 13$). **c**. Only decoding from WM trials following WM training showed significant classification accuracy ($p < 0.05$). **d**, No significant difference in the dimensionality of delay activity prior to CRs and FAs; $p = 0.07$, signed-rank test.

Furthermore, training such a 'FA decoder' exclusively on Discrimination task trials, and testing on WM task trials, resulted in a chance level classification accuracy (signed-rank test against a 50% chance level; **Rebuttal Fig. 1a, c**), and the dimensionality of the population activity (estimated as the eigenspectrum of the covariance of the delay activity) did not significantly differ between delays prior to correct rejections and FAs (**Rebuttal Fig. 1b, d**; this is further elaborated upon in our response to your comment 3c below). These results give us more confidence that the relationship between CD_{TASK} and CD_{CUE} activity with subsequent behavioural responses, despite being relatively weak, is nevertheless quite significant, and that this relationship does not reflect any clearly dominant (or at least linearly decodable) confounding premotor signal.

We introduced the possibility of a common premotor signal corrupting WM-related population activity, and argued against it, in the relevant section of the Results:

“Reduction of delay period CD_{TASK} activity did not predict incorrect responses during the Discrimination task, nor were behavioural responses associated with changes in the dimensionality of the population activity during the delay (Extended Data Fig. 9), suggesting that the relationship between delay-period CD_{CUE} and CD_{TASK} activity and subsequent behavioural responses was not explained by task-agnostic premotor activity.”

2) Interpretation of FA in WM task. In the behavioral experiments (Fig 1), a key behavioral feature of WM is an increase in FA rate with increased delay duration, which implies a degradation of WM maintenance over time. However, the stimuli used in the WM task (+45 and -45 deg) are the rewarded stimuli from the discrimination task. This presents a potential confound, because another interpretation could be that mice are progressively more likely to respond in blocks where a rewarding stimulus is repeatedly presented.

Thank you for raising the possibility that a positive relationship between FA rate and delay duration in WM blocks could be a byproduct of the increased rate of the presentation of previously rewarded stimuli during the WM blocks. We believe that our data speaks against this interpretation. We observed that responses to Probe stimuli during the WM task, despite having the same reward contingency as the Cues, were unaffected by the delay duration in the WM blocks (**Extended Data Fig. 3**). This precludes a simpler ‘progressively more likely to respond’ phenotype, and instead indicates that the effect of the delay duration was specific to *WM performance*. Such degradation of performance over time, whether by decay or interference, is a hallmark of working memory (e.g. Pasternak & Greenlee, 2005, 10.1038/nm1603; Barrouillet et al., 2018, 10.1080/17470218.2017.1358293), as there is no reason why WM performance, specifically, should be dependent on delay duration other than due to a degradation of the maintained WM information over time.

We have updated the relevant sentence in the Results section which addresses this concern:

“Crucially, this effect was specific to the maintenance of mnemonic sensory information, as responses to the Probe stimuli during the WM task, despite having the same reward contingency as the Cues, did not depend on the preceding delay length (Fig. 1d and Extended Data Fig. 3b; $n = 9$ mice, all $p > 0.05$).”

Can the data dissociate WM degradation from an urge to respond to the rewarding stimulus? One possible solution could be to examine the WM task performance when an unrewarded stimulus is used as the cue. For example, the behavioral data from imaging experiments using rotated visual stimuli, which use non-rewarded stimuli as cues in some blocks, could clarify this question.

Thank you for suggesting this analysis. We first measured the relationship between the delay duration and subsequent correct responses during the WM task in our imaging data, and then tested whether this relationship was different depending on if the stimuli during the WM task were matched or rotated away from the Target stimuli in the preceding Discrimination task. The delay length-WM performance relationship for individual experiments from the imaging dataset is shown in **Rebuttal Fig. 2a**. We found that the strength of this relationship in WM task blocks, which followed Discrimination task blocks that were of the same rotation block (i.e. shared Target and Cue stimuli) or of the rotated away rotation block (i.e. different Target and Cue stimuli), was not significantly different ($p < 0.05$; **Rebuttal Fig. 2b**). This was also true when correcting for any differences in the initial Discrimination task delay duration effects (data in Purple, calculated by simply subtracting the

Discrimination task effect from the subsequent WM task effect). This analysis indicates that the effect of delay duration on WM performance was present even when the WM Cues were not rewarded in the preceding Discrimination task blocks.

Rebuttal Fig. 2 | Orientation of rewarded stimuli in the preceding Discrimination task was not related to the delay duration dependence of WM performance. **a**, A summary of delay duration effects across all imaging experiments (individual lines), in both tasks (WM red, Discrimination blue), as measured by the proportion of correct responses to stimuli following short (<1600 ms) and long (>1600 ms) delay durations. Thick lines are averages across experiments. Percentage of correct responses was significantly reduced during the WM task ($p < 0.001$), and did not differ during the Discrimination task ($p = 0.31$). **b**, Effect of delay duration on correct responses (measured as the difference between the two columns in **a**), per experiment (individual circles), for the WM (red) and Discrimination (blue) tasks, and the difference between the two tasks (purple), plotted separately for experiments when the Discrimination task block that was preceding the WM task block was of the same stimulus rotation (*not rotated*, i.e. the WM Cues were matched to the Discrimination task Targets), and different rotation (*rotated out*, i.e. when the WM Cues were of a different orientation to the Discrimination task Targets). There was no significant difference between these conditions ($p = 0.56$, $p = 0.91$, and $p = 0.79$ for the Discrimination task, WM task, and difference between tasks, respectively, rank-sum tests).

3) High-dimensional WM representation. The finding that informative dimensions for task and stimulus decoding are higher during WM than during stimulus presentation is very interesting (Fig 3). A few points could be better clarified.

a. How are the principal components and variance explained calculated in Fig 3b? Presumably these are calculated based on single trial activity, but the description is missing.

Principal components were calculated separately for each experiment, with the single trial activity simply concatenated in time (which is effectively the raw population timeseries with data from unused trials ignored). The decoding accuracies from PCs of this population activity and the variance explained by the PCs are accordingly averages across experiments, and shaded error bars are CIs across experiments. The text in the results section (line 293) currently directs the reader to the Methods section, where these details are provided (line 1053).

We have updated the text in the Results section to be more clear:

“In order to assess the sparsity of this population encoding and to dissociate the number of cells required for decoding from the total activity variance explained, we performed a similar

decoding sweep with each experiment's population activity projected onto the Principal Components (PCs) of the population activity (PCs were calculated independently per experiment, from all of the single-trial population activities concatenated in time; see Methods)."

b. Does the higher dimensionality result from single cells responding sparsely and in uncorrelated manner as suggested by the activity-silent WM storage hypothesis? This is implied in several places in the text but could be better shown or explained.

The activity-silent WM storage hypothesis (Stokes, 2015, 10.1016/j.tics.2015.05.004) does predict high-dimensional population activity patterns reflective of WM which are similar to what we have observed in our data. However, as we additionally observe 'dominant' and sustained population activity patterns that are not related to WM, it is difficult to disambiguate whether the sparsity of WM-subspace specific signals (i.e. single cell contributions to single trial CD_{TASK} and CD_{CUE} activity) is arising from structured 'hidden' network states (e.g. short term plasticity or functional connectivity states) or, for example, from local recurrent dynamics which are obscured by (potentially input driven) low-dimensional dynamics (e.g. Galgali et al, 2021, 10.1101/2021.07.19.452951). Single cell stimulation experiments are perhaps the most promising avenue for disambiguating the physiological basis of WM-specific high-dimensional activity patterns.

The related 'Hot-Coal' theory of WM (Lundquist et al, 2021, 10.1101/2020.12.30.424833) is likewise supported by the relegation of WM representations to the high-dimensional modes of population activity that we observed, but further work using electrophysiological measurements is required for us to identify the hallmarks of such WM models, such as WM informative events being restricted to gamma bursts.

We have added these links to the Discussion section of the manuscript:

"Instead, our results provide support for an alternative representational scheme for working memory maintenance, whereby dispersed cell populations encode working memory with trial-to-trial variable and uncorrelated delay activity patterns. Such high-dimensional population codes have recently been associated with 'hot-coal' (Lundqvist et al., 2021) or 'activity-silent' (Stokes, 2015) theories of working memory maintenance, although further investigations with electrophysiology and targeted causal network perturbations, respectively, will be necessary to draw direct comparisons."

c. The results show that stimulus information is maintained in high-dimensional dynamics during WM and collapsed activity along CD_{cue} predicts behavioral errors. Is the collapsed activity due to a reduction in dimensionality of delay activity? Related, does the dimensionality of stimulus activity differ between the WM and discrimination task?

This is a great question. We have generated a new **Extended Data Fig. 9** with a thorough analysis of the eigenspectrum of the covariance of the population activity during the delay and stimulus periods, across tasks, and prior to correct behavioural responses and errors. Interestingly, we did not find any differences in the dimensionality of the data between tasks or between different behavioural outcomes in either task.

We have updated our Results text with the following:

"Furthermore, the dimensionality of the population activity (eigenspectrum of population activity covariance) did not differ significantly between tasks (Extended Data Fig. 9)."

“Reduction of delay period CD_{TASK} activity did not predict incorrect responses during the Discrimination task, nor were behavioural responses associated with changes in the dimensionality of the population activity during the delay (Extended Data Fig. 9), suggesting that the relationship between delay-period CD_{CUE} and CD_{TASK} activity and subsequent behavioural responses was not explained by task-agnostic premotor activity.”

Extended Data Fig. 9 | Dimensionality of population activity across tasks and behavioural responses. a-c, The eigenspectrum of the covariance of the population activity during the interstimulus delay period, shown as the variance explained by the first 20 Principal Components (PCs). PCs were calculated separately for individual experiments similarly to **Fig. 3b** (see Methods). Thick line is the average and shaded region is the 95% CI across experiments ($n = 18$). **a**, Trials split by task (Discrimination task in blue, WM task in red). There was no significant difference between tasks in the variance explained by the first PC ($p = 0.46$, signed-rank test). **b-c**, Trials split by behavioural response to subsequent stimulus during the Discrimination (**b**) and WM (**c**) tasks (CR grey, FA black). CR trials were randomly selected to match the number of FA trials per experiment, and this was permuted 100 times and the results averaged per experiment. There was no significant difference between tasks in the variance explained by the first PC in either task ($p = 0.33$ and $p = 0.54$ for Discrimination and WM task, respectively, signed-rank tests). **d-f**, as in **a-c** but for area M2 experiments ($n = 13$, $p = 0.16$, $p = 0.19$, and $p = 0.77$, for differences between task, behavioural response during Discrimination task, and behavioural response during WM task, respectively). **g-i**, as in **a-c** but for population activity during the stimulus ($n = 18$ experiments, $p = 0.68$, $p = 0.64$, $p = 0.85$, differences between task, behavioural response during Discrimination task, and behavioural response during WM task, respectively). **j-l**, as in **d-f** but for population activity during the stimulus ($n = 13$ experiments, $p = 0.85$, $p = 0.42$, $p = 0.42$, differences between task, behavioural response during Discrimination task, and behavioural response during WM task, respectively). **a-l**, Delay activity was significantly lower dimensional than stimulus activity in area AM (26% variance versus 19% variance explained, respectively, by the first PC, averaged across 18 experiments, $p < 0.001$, signed-rank test) and area M2 (20% variance versus 14% variance explained by the first PC, respectively, averaged across 13 experiments, $p < 0.05$, signed-rank test).

Minor:

1) Do the authors have an explanation for why M2 inactivation only transiently affects

behavior, but AM inactivation induces persistent effect? Slower timescale activity is usually associated with frontal cortical regions. Here, it seems the results break that expectation.

We had few *a priori* expectations regarding differences between areas in the transience of behavioural effects following optogenetic inactivation, as such differences could stem from multiple overlapping mechanisms. For example, if the timescales of the activity patterns observed in the inactivated areas determined the effects on behaviour, then our data would indeed break expectations, but if the transience was caused by the re-entrainment of local recurrent dynamics by ongoing long range cortical input following the cessation of inactivation, then the recovery of behaviour would be dependent on a given area's long-range connectivity patterns. Another potential mechanism could be the degree to which 'activity-silent' mechanisms (e.g. short term plasticity states; not disrupted by optogenetic inactivation) vs. population dynamics are recruited to maintain WM.

Interestingly, the fact that behaviour recovered following area M2 inactivation is consistent with observations from another frontal cortical area, ALM, following delay inactivation when mice were performing a motor preparation task (Li et al., 2016, 10.1038/nature17643). In light of their model, our results would suggest that transcallosal input from the contralateral hemisphere would function differently in posterior visual cortical areas (i.e. not being able to re-entrain local AM dynamics).

We added a section in the Discussion section to further address this result.

“Furthermore, the behavioural and neural consequences of inactivation differed in their transience, with different cortical areas showing varying degrees of recovery following inactivation. The ability for behaviour and neural activity to recover following area M2, but not area AM, inactivation during the early delay period can be contrasted with the timescales of activity patterns observed locally in these areas, which have been found to be slower in anterior regions (Murray et al., 2014), but is consistent with recent reports of the re-entrainment of population activity in the frontal cortex following inactivation (Li et al. 2016).”

2) For axon imaging + target-region inactivation experiment, is the disruption of activity limited to CD_cue or does the inactivation also affect the CD_task activity?

While our axonal imaging and inactivation experiments were conducted with the same behavioural task design as the cell-body imaging experiments (i.e. with alternating task and rotation blocks), we sometimes had difficulty tracking individual boutons across task blocks (which were separated in time by around 20 minutes) due to issues such as axial drift. As further analyses of cross-task activity (i.e. CD_{TASK}) would have required generating a second heavily curated data set which only included stable axonal imaging experiments, we chose not to include such analyses in this manuscript, and limited our analyses to within WM blocks.

3) Line 352: typo. "... following area AM inactivation (Fig. 4k)". This is likely referring to Fig 4e.

Fixed, thank you.

4) Line 432-434, "The selective disruption of working memory representations in corticocortical feedback projections by the inactivation of ongoing feedforward activity provides direct evidence for the mechanistic role of functional cortical feedback loops for the maintenance of internally generated cognitive representations." I did not fully follow this conclusion. The results show that the inactivated cortical region plays a role in maintaining WM, and the inactivation

also affects downstream cortical regions that project back to the inactivated region, which suggests that their interactions are required. But it is not clear if the affected cortical feedback signal itself is required for WM maintenance. Can the authors more clearly explain their interpretation?

We believe a key factor that makes the dependence of feedback activity on ongoing feedforward activity ‘direct evidence for’ the *functional* role of cortical feedback loops is the reciprocal nature of our observations – area M2 feedback carrying WM representations required area AM activity and area AM feedback required area M2 activity. As such, although our experiments do not test precisely if the feedback WM representations were causally maintaining the target area’s WM representations (testing this would require subspace specific axonal manipulations – currently technologically unfeasible), our experiments do confirm a specific experimental prediction which has been proposed by some of the theoretical studies which we cited (e.g. Friston, 2010, 10.1038/nrn2787; Lee & Mumford, 2003, 10.1364/josaa.20.001434). Specifically, we confirmed that, consistent with the role of feedback loops maintaining and coordinating internal cognitive representations, the feedback activity of WM between areas in a recurrent loop is dependent on activity in their targets. Accordingly we believe ‘provides direct evidence for’ is sufficiently tempered.

Reviewer #2 (Remarks to the Author):

This study is an excellent combination of a unique, highly challenging behavioral paradigm and best-of-class physiology and activity perturbation. In brief, the authors designed a novel task, in which mice alternated between a working memory task and a discrimination task allowing to match stimuli, movement and reward statistics across tasks. This novel task helps disentangle neural representations of working memory from other behavioral variables. The authors carried out multiple perturbation experiments and showed that working memory is distributed across areas of neocortex. Simultaneous axonal recordings during inactivation of a distal cortical area further shed light on the role of interareal reciprocal interactions in the maintenance of working memory. This is highly valuable, important and timely work whose results will be of great interest to the neuroscience community. I feel that some analyses could be performed in a more straightforward way and more clearly presented, which would further improve the manuscript.

Major comments:

1. The task is great. Very impressive.

2. In the analysis of population data, the use of “high-dimensional representations” and the related conceptual model was confusing to me. “High dimensional representations” could mean representations that live in higher dimensions, or it could mean representations that are themselves high dimensional. It felt to me that on occasion the manuscript switches between those two meanings. To step back, there are two clear findings: (i) Task related information is present in population activity. (ii) Using only the first PCs that capture a large part of the variance is insufficient to fully capture this information. What this means is that the task related information is present in dimensions of activity space that don’t capture much variance.

In their decoding analysis the authors find that a one dimensional decoder performs well. Therefore, the task-related dynamics themselves are not necessarily high dimensional. As an example, one can have a model in which the task-relevant dynamics are one (or low) dimensional and orthogonal to another set of dynamics that are not task relevant and capture a large amount of variance, which would be inline with the findings. The third finding is that a large number of neurons is needed to arrive at good decoding. It is related to dimensionality but not in a very straightforward way since the representation may be low dimensional yet a large number of neurons may just be necessary to estimate the dynamics in the low variance part of the population activity that allows to decode working memory.

Your appraisal of our data and results is very much in line with what we wished to convey. Perhaps an important differentiation that could help with some confusion is that the *intrinsic* dimensionality of the population activity (alternatively referred to as the dimensionality of the *dynamics*) is a concept which we have not explored in this manuscript. The fact that a one dimensional linear decoder could identify WM representations implies that the dynamics of these representations are either one dimensional or kept orthogonal through time (e.g. rotational), but says nothing about the *embedding* dimensionality of these representations (the number of dimensions of the ‘ambient’ neural space explored by these representations) – the type of dimensionality which we explored in this manuscript. Embedding dimensionality is an important metric as it gives insights into the local network operations (Jazayeri & Ostojic, 2021, 10.48550/arXiv.2107.04084), how the information can be decoded by downstream brain areas, and has been a focus in recent theories of WM mechanisms (e.g. Lundqvist et al., 2021, 10.1101/2020.12.30.424833).

Furthermore, as you pointed out, the fact that most cells have some information about WM (i.e. the representation of WM being dispersed in the population) is not sufficient as a measure of the embedding dimensionality, as, for example, if this signal was highly correlated across all cells within the population then it would imply different neural mechanisms and read out strategies (e.g. subsampling from the population being a reasonable strategy). As such, we used ‘high-dimensional embedding’ to summarize two properties of the WM representations we observed; that a large proportion of cells carried WM representations and that the correlated modes of this population activity were not sufficient to capture these WM representations.

I believe the authors have a different model in mind, one in which (I state it here simplified) working memory is distributed across many neurons, but a neuron doesn't have information in every trial, but rather only in a small subset of trials. In this scenario, trial-averaging would average away this decoding contribution, pushing it into low variance components and unless one has access to many neurons, most trials will be un-decodable, consistent with their findings (as a side note there is a potential problem in this model: since the decoder is fixed it has to assign the same weight to a neuron whether it is in a trial in which that neuron happens to be informative or not, but if the signal in informative trials is strong enough the decoder will still work). I think it is hard (and unnecessary) to argue for this model and it is best to revise statements in the text to clarify the specific findings and the possible interpretations more generally, and not just in relation to this model.

We agree that our data does not exclusively lead to this particular model for WM representations. We referenced this model in our text as it is an existing relevant model for WM representations in the

field that is *consistent* with our data (as opposed to, e.g. persistent sensory activity a la Romo et al., 1999, 10.1038/20939). We have qualified our statements further in the Discussion:

“Instead, our results provide support for an alternative representational scheme for working memory maintenance, whereby dispersed cell populations encode working memory with trial-to-trial variable and uncorrelated delay activity patterns. Such high-dimensional population codes have recently been associated with ‘hot-coal’ (Lundqvist et al., 2021) or ‘activity-silent’ (Stokes, 2015) theories of working memory maintenance, although further investigations with electrophysiology and targeted causal network perturbations, respectively, will be necessary to draw direct comparisons.”

We hope that the changes made to the text as part of this revision do emphasise the specific findings. If there are any sections that are still too interpretative, please point them out, and we’d be happy to revise further.

3. In terms of the CD_task, I also feel that the text and analysis follow one particular interpretation for these dynamics when the full picture might be more complex. The authors state that it “captures the activity introduced by working memory engagement by contrasting the delay activity in the two task blocks following identical sensory input”. While this is a possible interpretation, another possible interpretation is that there is a context-signal that “tells” the circuit which task it is in, unrelated to the specific computations necessary for remembering the cue. The way CD_task was defined may in fact favor such a signal, as memory of the cue may be present in the discrimination blocks of the task despite the fact that it isn’t behaviorally useful. I realize that the memory cannot be quite the same due to the differential results of length-of-delay on behavior and the differential perturbation effects (and that the projection has complex dynamics), but these results are not enough to justify assuming no memory. The reason it is potentially important to consider these different scenarios is in the interpretation of projections on CD_task. This interpretation differs whether one has in mind a context-signal or a reflection of the actual process of working memory. One way to get more quantitative purchase on this question is to directly test differences in the preservation of cue information through the delay epoch between the Discrimination task and the WM task (see also comment 7 below). Specifically, one possible analysis could be calculating a CD_cue that contrasts between the -15° cue and +15° cue in the Discrimination task and then compare the activities in their respective CD_cue in the Discrimination task and WM task. This comparison can be done for CD_cue both at the delay epoch and at the stimulus period. It would be interesting to see whether there is any consistent difference, for example, activity in the discrimination blocks along CD_cue may decay faster than in the WM task. Or perhaps there is no significant difference between the blocks which would suggest that working memory is also maintained in the Discrimination task, despite it being unnecessary for successful completion of the trial.

This is a great point, and we addressed it at the onset of this rebuttal (combined with response to major comment #1 of reviewer #1). We found that stimulus-related activity during the delay period (CD_{CUE}) was largely absent during the Discrimination task, even though the stimulus period CD_{CUE} activity (i.e. response to the stimulus itself) was similar across tasks.

Nevertheless we do appreciate that whether CD_{TASK} activity reflects a “context-signal or a reflection of the actual process of working memory” is a question which we do not address. One observation

which lends support to CD_{TASK} being related to WM processes (and not simply a context signal for which task the mouse is in) is that CD_{TASK} activity was more robust and persistent during the WM task as compared to the Discrimination task (**Fig. 3d, g**), and predicted correct behavioural responses only during the WM task (**Fig. 3e, h**). Further insights would perhaps require ensemble specific (i.e. CD_{TASK} subspace specific) single-cell optogenetic perturbations in future experiments to be adequately answered.

We have expanded the relevant section of the results:

We found that both the slope and R^2 of the CD_{TASK} activity was higher when the mice were performing the WM task as compared to the Discrimination task (Fig. 3d, g), suggesting that CD_{TASK} activity reflected representations integral to WM instead of WM-agnostic context signals such as those that reflect the task the animals are engaged in, and supporting the role of attractor-like dynamics for the maintenance of working memory representations (Murray et al., 2017; Inagaki et al., 2019, Koyluoglu et al., 2017).

4. The presentation of the results of the population analysis could be made more straightforward. In particular, Fig. 3e, k, h, n are the main quantitative findings in their section and are in my opinion unnecessarily confusing. The text states: "We observed that in both areas AM and M2, disruption of the CD_{cue} activity or CD_{task} activity during the WM task, as measured by the failure to correctly classify a given trial's cue or task identity, was predictive of incorrect responses to the subsequent stimulus (Fig. 3e, h, k, n)". It is possible I missed something, but the way the data is summarized in these plots seems unnecessarily difficult to absorb. Since the cue direction is built as a decoder of cue, the state of the decoder (on one side or another of an assumed threshold) should relate to the represented cue and a switch to the other side of the threshold should correlate with errors. It therefore seems more straightforward to directly compare the state of the decoder to the action at the end of the delay. While there is an imbalance in trial types since 80% of the trials are match trials, the analysis still seems relevant in its simpler form. The current choice of comparing decoder accuracy separately in CR and FA trials and the claim that a reduction is expected seems a very complex presentation of this statement. I believe the idea is similar, that a mistake in decoder indicates a switch to the opposite cue and this should happen more in FA trials, and a decrease in accuracy means more errors, but the multiple inversions make it harder to understand, in my opinion.

Thank you for pointing this out. We used the proportion of trials correctly classified (%) as a measure because it allowed us to pool data across experiments (as all trials are assigned either a correct or incorrect), and perform a high power Fisher's exact test for significance. We have now evaluated the ability of population activity along WM coding dimensions (CD_{TASK} and CD_{CUE}) to predict subsequent behavioural responses in a manner closer to the data, by looking at individual experiments' population activity projections onto the WM coding dimensions prior to correct rejections and false alarms, and performed the appropriate statistics (signed-rank tests across experiments). These analyses are compiled in a new **Extended Data Fig. 12**. These analyses confirm that CD_{TASK} and CD_{CUE} activity during the delay period was able to predict correct behavioural responses to the subsequent Cues during the WM task.

Extended Data Fig. 12 | Working memory subspace population activity predicted correct behavioural responses during the working memory task. **a**, Z-scored CD_{TASK} delay activity of individual experiments from area AM ($n = 18$), split by behavioural outcome (CR and FA), and averaged across Discrimination task (blue) and WM task (red) trials ($p < 0.001$ for the WM task, $p = 0.21$ for the Discrimination task, signed-rank tests). Horizontal bars represent medians. **b**, Proportion of trials with their task correctly classified using CD_{TASK} delay activity prior to CRs or FAs, as in Fig. 3.e, plotted for individual experiments from area AM ($n = 18$, $p < 0.01$ for the WM task, $p = 0.79$ for the Discrimination task, signed-rank tests). **c-d**, as in **a-b**, for area M2 experiments ($n = 13$; Z-scored CD_{TASK} delay activity differences $p < 0.05$ and $p = 0.79$ for WM and Discrimination task trials, respectively, and task classification accuracy differences $p < 0.01$ and $p = 0.91$ for WM and Discrimination task trials, respectively, signed-rank tests). **e-h**, As in **a-d**, for CD_{CUE} activity during the WM task. **e-f**, For area AM experiments ($n = 18$; Z-scored CD_{CUE} delay activity differences $p < 0.01$ for both Cue A and Cue B trials, and Cue classification accuracy difference $p < 0.01$, signed-rank tests). **g-h**, For area M2 experiments ($n = 13$; Z-scored CD_{CUE} delay activity differences $p < 0.001$ and $p = 0.19$ for Cue A and Cue B trials, respectively, and Cue classification accuracy difference $p < 0.001$, signed-rank tests).

Especially given the text which conflates the activity and the decoder and uses the word disruption which can be easily confused with an active perturbation. In addition, the use of "preceding" in the axis labels is ambiguous. It could mean that the preceding trial was a CR or FA, not that the delay period preceded a CR or FA outcome.

We agree that our language was ambiguous. We have updated the relevant Results section text.

"We observed that in both areas AM and M2, a reduction of CD_{CUE} activity or CD_{TASK} activity during the WM task delays, as measured by the failure to correctly classify a given trial's cue or task identity, was predictive of incorrect responses to the subsequent stimulus (Fig. 3e, h, k, n; Extended Data Fig. 12 shows these results in terms of individual experiments' CD_{TASK} and CD_{CUE} activity)."

We have also updated the behaviour prediction axis labels of Figure 3 to be more clear.

5. The analysis of the time course of single trials in lines 296-308 is presented as highly informative of activity underlying working memory. "The slope of this relationship is indicative of the persistence of working memory representations over the course of the delay, and the R^2 is indicative of their robustness over time". While it is true that single-trial dynamics during the delay period are interesting, and the slope and R^2 of a comparison between the first and

second part of the projection on a decoder are relevant metrics, they are not as central as presented. For instance, in a straightforward decoding model of a comparison to threshold, the decoder projection values can become completely scrambled between the first and second part of the delay period (yielding zero R² between the first and second parts of the delay period), yet as long as each trial doesn't cross the threshold there is no loss in performance. It is possible that I misunderstood the analysis, and the way the different trial types were treated means that a zero R² would necessarily mean complete mixing between condition types, but I don't think that is the case given the way data is plotted in Fig. 3 (see also next comment). If this is true then there are simpler ways to show and describe this effect.

Thank you for pointing out this potential source of confusion. While you have correctly understood the reasoning behind our use of R² and slope for analysing single trial dynamics, we did not intend for these measures to be used to gauge the ability of the neural population to predict either Tasks or Cues. The 'mixing of condition types' within the neural population is instead measured by the cross-validated performance of our decoding, which is presented in **Fig. 3b** and **Extended Data Fig. 11** (the horizontal lines representing the experiment averaged classification accuracy of left-out trials). The R² and slope of early/late delay coding dimension activity of single trials is instead a measure of the *stability* of the population activity along the calculated coding dimensions.

We have revised the relevant Results section text to be more clear.

“The slope of this relationship is indicative of the persistence of working memory representations over the course of the delay, and the R² is indicative of their robustness over time. Importantly, these measures are agnostic of whether the Cue or Task identities of the trials were correctly predicted (i.e. the classification accuracy).”

6. In general, it was difficult for me to follow the precise details of the population analysis. The first part of the analysis is based on a pooled pseudo-population, with trial-averaging, but then the analysis switches to single-trial dynamics and decoding (with no trial averaging). The mechanics of how single trial analysis relates to sessions (each with different trial numbers) wasn't clearly described. I believe the authors used single session analysis that leaves one trial out and then just pooled these left out trials across all sessions, but it wasn't clear whether the values given would be changed if averages across sessions are taken, and I am not sure I correctly followed what was performed. Clearer descriptions in the main text would be useful as well as longer explanations in the methods (and/or putting code in a code repository).

We believe some confusion arises from what is being plotted in **Fig. 3b** and **Extended Data Fig. 11**. These are single experiment classification accuracies averaged across experiments, with the error bars accordingly representing the 95% CI across experiment. Cross validation *within* each experiment was done by taking one trial out and calculating the coding dimensions from the remaining trials, and then classifying the left out trial (i.e. correct or incorrect Task or Cue), and repeating this procedure for all of the trials, once per experiment. For analyses of single trial coding dimension dynamics, the same procedure was done but with the left-out trial's population activity projected onto the coding dimensions identified from the remaining trials. For some analyses, to pool across experiments, we either pooled the Z-scored projection strength of all trials within each experiment (i.e. when they were left out for calculating the coding dimensions) for **Fig. 3.d, g, j, m**, or simply pooled each trial's correct/incorrect classification (when it was left out) for **Fig. 3.e, h, k, n**. A version of **Fig. 3.e, h, k**,

n, but with the trial-averaged Z-scored projection strengths (i.e. calculated separately for each experiment) is presented in the new **Extended Data Fig. 12** (see our reply to your comment 4 above).

We have expanded the relevant Methods section to clarify our analyses.

“All reported Task or Cue decoding accuracies were the average cross-validation (leave-one-out) test accuracies, calculated by averaging each trial’s prediction of Task or Cue given the coding dimensions derived from the respective experiment’s remaining trials (i.e. one classification accuracy was derived per experiment). All projections of the neural population activity onto the respective coding dimensions (e.g. Fig. 3c, f, i, l, and Fig. 4e, k) are likewise the projections of the activity of left-out single trials onto the coding dimensions calculated from their respective experiment’s remaining trials.”

The analysis code is available in an online repository: github.com/ivan-voitov/loops

7. Regarding the point that dynamics in the high variance components of the two task blocks are very similar, one possibility (as alluded to in comment 3) is that since the animals were trained on both tasks, they adopt a strategy in which they memorize the cue in all block types. This puts the experiments in a bit of a catch 22: in order to compare the dynamics one has to train the same animal on both tasks, but if one trains the animal on both tasks, the animal could memorize the cue in all task conditions. In principle this could perhaps be addressed by animals first learning the discrimination task and then adding the working memory task (perhaps also vice versa in separate animals), while performing longitudinal imaging. I believe these experiments would be too time consuming, especially given that the study already goes above and beyond the standard in terms of behavioral training. I therefore don't suggest the authors actually collect this data, but if the authors have existing data for the discrimination task alone (in separate animals) they could argue for its similarity to the full task data to make the scenario of working memory training changing the dynamics for both block types less plausible. The authors should also raise this option, and/or argue against it, in the discussion section.

This is an interesting line of thought. We believe our data supports the view that the population encoding of WM is specific to the WM task, given that CD_{CUE} activity is largely absent during the Discrimination task (new analysis presented in **Extended Data Fig. 13**), and that the CD_{TASK} activity, although present in the Discrimination task (by virtue of its derivation), is more robust and more persistent during the WM task (**Fig. 3**).

Nevertheless, the activity patterns that are shared across tasks may potentially not be observed if the mice were trained only on the Discrimination task, and emerge only following WM task training, perhaps due to the increased difficulty or training time required for the WM task (e.g. the Discrimination task alone, being Pavlovian in nature, may be ‘performed’ using quicker neural structures than the neocortex, as suggested by our reaction time results and optogenetic inactivation experiments). In such a framework, these shared activity patterns may be related to aspects of the task such as timing, reward expectation, and motor preparation. In support of this idea, several recent studies have suggested a role for low-dimensional dynamics which is agnostic of the information kept in WM, such as the maintenance of a timing signal (Meirhaeghe et al., 2021, 10.1101/2021.11.08.467806), or to support the orthogonalization of sensory input with respect to premotor activity (Libby & Buschman, 2021, 10.1038/s41593-021-00821-9) – processes which may be useful even in simple tasks (i.e. to maximize the reliability of behavioural responses), but not normally recruited under simpler (e.g. Pavlovian) learning experience. Interestingly, our findings

somewhat contradict a recent preprint, Arlt et al. (2021; 10.1101/2021.12.10.472106), that training mice on a more difficult task would increase the necessity of the neocortex for a previously trained simpler task, as we found no requirement of the neocortex for the Discrimination task during the delay even when blocked together with the WM task (of course, however, the tasks used for these experiments are not interchangeable).

Unfortunately we do not have recordings from mice which were only trained on the Discrimination task, as the clarity of our cranial windows and viral expression times limited our flexibility in data collection, and as such we always made sure to record both tasks in each mouse as soon as was possible.

We have added a section in the Discussion section to bring up this point.

Similar two-task designs have previously been used to disambiguate the neural correlates of specific cognitive processes by isolating neural representations of interest from ‘condition-independent’ neural activity (Tajima et al., 2017; Panichello & Buschman, 2021). One potential drawback of the two-task design is that neural activity may be recruited which would otherwise be absent if the mice were only trained on one task. Accordingly, although we identified shared neural activity patterns between our Discrimination and WM tasks, the extent to which such WM-independent signals would arise if mice were trained only on the Discrimination task is unknown. Nevertheless, ethological behaviour is characterized by flexible switching between a vast repertoire of previously learned behaviours, and two-task designs therefore impose a reasonably conservative control for investigating neural correlates of cognitive processes.

8. Going back to the role of CD_task as in comment 3 above, I was wondering whether one can try to analyze trials in which dynamics go into the wrong state in terms of task and see whether there is a specific prediction to be made regarding different relevance for the state of CD_cue, i.e., what should one expect assuming that there might be a mistake in the assignment of the task. For instance, would a crossing of threshold in both CD_task and CD_cue compensate for each other? I believe it is not so simple since the relation between the cue and probe are different in discrimination and WM blocks, but perhaps something can be done.

Thank you for this interesting suggestion. We performed the analysis as you suggested, where we took trials which had their Task correctly or incorrectly classified during the WM task blocks, and then looked at the Cue classification accuracy of these trials. We further split trials into those preceding CRs and FAs. We found that there was no compensation of Task information with Cue information, and, in fact, there was a mutual disruption of both Task and Cue encoding, wherein when population activity had difficulty predicting the Task, it also failed to predict the preceding Cues. A simple interpretation of this could be that these two types of WM representations are co-dependent. There was a modest correlation (Pearson’s r) between CD_{TASK} and CD_{CUE} activity of 0.26 ± 0.18 for area AM experiments and 0.18 ± 0.26 for area M2 experiments (mean \pm 95% CI across experiments).

Extended Data Fig. 14 | Relationship of Cue and task representations. **a**, Delay period averaged population activity, per WM task trial, from an example experiment in area AM, projected onto the CD_{TASK} (horizontal axis) and CD_{CUE} (vertical axis). Black points represent individual trials' delay period activity prior to CRs and red points prior to FAs. CD_{TASK} and CD_{CUE} were identified independently for each trial using all remaining trials (i.e. leave-one-out cross-validated). Points below zero correspond to trials in which the task (horizontal axis) or Cue (vertical axis) was incorrectly classified. **b**, The prediction of preceding Cues from the population activity during the delay (measured as classification accuracy, i.e. the percentage of trials below zero on the vertical axis of **a**), when the Task was incorrectly (left column) or correctly (right column) classified, for all trials pooled from area AM experiments ($n = 3,452$ CR trials and $n = 405$ FA trials pooled from 18 experiments; $p < 0.001$ and $p < 0.01$ for CR and FA trials, respectively, Fisher's exact tests). Error bars represent 95% CI. **c**, The Pearson's correlation coefficient (r) between CD_{TASK} and CD_{CUE} delay activity, per area AM experiment ($n = 18$; greater than zero, $p < 0.001$, one-sample signed rank test). Horizontal bar represents median. **d**, As in **b**, but for area M2 experiments ($n = 2,378$ CR trials and $n = 395$ FA trials pooled from 13 experiments; $p < 0.001$ and $p < 0.01$ for CR and FA trials, respectively, Fisher's exact tests). **e**, as in **c**, but for area M2 experiments ($n = 13$, $p < 0.05$, one-sample signed rank test).

We have added the corresponding text to our Results section:

“Notably, when the delay activity failed to predict the current task (i.e. CD_{TASK} misclassification), the Cue information encoded in the delay activity (measured as CD_{CUE} classification accuracy) was concomitantly lower (Extended Data Fig. 14), revealing a link between these two coding dimensions (Pearson's $r = 0.26 \pm 0.18$ for area AM experiments and 0.18 ± 0.26 for area M2 experiments; mean \pm 95% CI across experiments).

Related to this point, it wasn't clear to me whether CD_{task} and CD_{cue} were explicitly orthogonalized or they were just generally found to be orthogonal.

No we did not orthogonalize them. They were found to be (roughly) orthogonal. The correlation coefficient (r) of individual trial scores (delay-averaged population activity of single trials projected onto the CD_{TASK} and CD_{CUE}) was 0.20 ± 0.16 (mean \pm 95% CI across all experiments). We added the relevant text in the Fig. 3 legend.

“Note that the CD_{CUE} and the CD_{TASK} were not explicitly orthogonalized.”

9. In the simultaneous imaging and perturbation of figure 4, even when activity remained perturbed, the 95% CI mostly don't overlap for most of the short delay, in particular for AM (as the shortest short delay is 0.8 seconds). This may imply that one shouldn't expect a behavioral effect. It is possible that the read out is more graded, but it is worth commenting on that.

Interpreting the very onset of CD_{CUE} activity during the delay may be tricky because boutons with progressively later onsets sum with the tails of the calcium transients of previously active boutons, leading to a slight illusory ‘ramping’ of population activity (deconvolution of the boutons' calcium

signals was not possible due to a low signal-to-noise ratio). Nevertheless it is true that there is never a perfect separation of baseline activity and silencing activity, which may reflect only a partial elimination of CD_{CUE} activity. An incomplete disruption of CD_{CUE} activity may in turn be consistent with the behavioural effects also being graded in nature. We have added a sentence to address this in the Results section.

“Although feedback CD_{CUE} activity was not completely eliminated following target area inactivation, this may reflect the partial impact of optogenetic inactivation on WM-specific behavioural effects (Fig. 1i)”.

In addition, how the CIs were calculated is not shown.

The trial-averaged CD_{CUE} activity projections of individual experiments were averaged for these plots, and the 95% CI are accordingly across experiments. We have added this information to the **Fig. 4c, e**, legends.

10. I was confused by how cue rotation figured into the task block structure. If I understood correctly, the mice can switch task blocks quickly because the absence/presence of a vertical grating is informative of the task block (line 793-794). But if the vertical grating can now be rotated and appear in both tasks, do mice recognize the task because there is only one cue type?

As all stimuli were rotated together during the rotation blocks (e.g. +15°), such that the Discrimination task vertical grating was still oriented at an angle equidistant between the two WM Cues (e.g. +15° when the Cues were -15° and +45°). Accordingly, the mice were still informed of the task switch by the abrupt presentation of a ‘near-vertical’ (±15°) stimulus that was oriented in between the Cues of the preceding WM task (i.e. the Discrimination task ‘vertical’ stimulus only matches Cues from opposing rotation blocks).

I was also confused by the statement in line 808-809 that “in between two rotation blocks, the stimulus orientation angles were changed slowly...” Does that mean the angles of the grating changed continuously? It would help to clarify these issues.

Yes, the angles were changed continuously.

We have updated the Methods section with a more detailed procedure.

“In between two rotation blocks, the stimulus orientation angles were changed slowly in a continuous fashion (averaging ~10 minutes for a full 30° rotation), such that the mouse performance was not disrupted. No previous training was required for the mice to perform these rotation blocks, and there was minimal interference with the mice’s ability to alternate task blocks as the sudden presence or absence of a stimulus in between the two Cues in the WM task remained an abrupt indicator of a task block switch.”

11. In the section “Cortical feedback loops maintain distributed working memory representations”, it is suggested that the representation of working memory disrupted by distal inactivation might be recovered if the delay is long enough. It would be useful to generate a version of Fig. 1h in which trials are grouped according to delay length (as in Fig. 4m) and check for the effect there as well.

Thank you for suggesting this analysis. We have generated these plots as **Rebuttal Fig. 3**. These plots demonstrate that with a different measure of performance, $[100\% - \text{Hit rate} (\%) - \text{FA rate} (\%)]$, and restricting the analysis to the optogenetics and behaviour dataset, the differential robustness of AM and M2 silencing was maintained. We chose not to include this metric in the manuscript as (1) a large amount of variability is introduced by the relatively rare Hit trials (**Fig. 4m** measures this same effect as $(\%)$ of correct trials, which weighs all trials equally), and (2) it mixes the effects of inactivation on Cues, Probes, and Targets (**Fig. 1i** and **Extended Data Fig. 6** show behaviour and optogenetics dataset statistics split by trial type). We would, however, be happy to include this figure into the manuscript if you think it would be useful for the reader.

Rebuttal Fig. 3 | Reduction in performance by optogenetic inactivation depended on delay period duration. **a**, A schematic of the cortical areas targeted for optogenetic silencing (top), and the optogenetic silencing protocol (bottom). The optogenetic silencing light was flashed for 400 ms at 3 mW, followed by a linear ramp down to 0 mW over 200 ms, at either the onset of the delay or stimulus periods. **b**, An overview of the silencing effects on task performance in trials with delay periods less than 1.6 seconds in length ($n = 143,764$), split by silencing onset (delay onset, left, and stimulus onset, right), task (Discrimination task, top, and WM task, bottom), and area (shaded circles). Performance was defined as $[100\% - \text{FA rate} (\%) - \text{Miss rate} (\%)]$. The shade of the circles is the differences in performance between control and silencing trials. **c**, As in **b**, for trials with delay periods greater or equal to 1.6 seconds in length ($n = 97,984$).

Minor comments:

1. Statistics for behavioral analysis and perturbations were not clear.

We have updated the Methods section with further details of how the statistical analyses of behaviour and optogenetic inactivation effects were performed. The relevant statistics are included in **Extended Data Fig. 6**.

“Statistical analyses of optogenetic inactivation effects (Fig. 1i and Extended Data Fig. 6) were done by pooling all trials from 9 mice ($n = 173,432$ trials) and performing a Fisher’s exact test, separately for Cue, Probe, and Target trials, and split by task. Significance levels were accordingly adjusted for multiple comparisons. Bar plot values were the trial-averaged optogenetic silencing effects subtracted from the control trials (where no silencing occurred), and error bars represent the 95% CI of the silencing trials (i.e. binomial confidence intervals).”

2. Line 166 states “... but were more diverse in their temporal profiles (Fig. 2c)”. It was not quite clear to me what is meant here. The two blocks in Fig. 2c look quite similar.

Sorry, we mistakenly referenced **Fig. 2c** instead of **Fig. 2a-b**, which may have led to some confusion. This sentence refers to differences between delay and stimulus responses, not between the two task blocks. We have updated the text to read:

“Single cells in both areas AM and M2 were as likely to be responsive during the inter-stimulus delay period as compared to during the presentation of the stimulus, but had more varied response onset times during the delay (Fig. 2a, b; Extended Data Fig. 8).”

3. Line 171: “Surprisingly, however, working memory engagement did not alter the temporal profile of the trial-averaged activity of individual cells” I am not sure what is the precise claim. Do no neurons have a different response, even by chance? Perhaps the statement related to changes above some chance level?

Yes, it is compared to chance, using a *t*-test between peak response times of all delay or stimulus responsive cells. The relevant statistics are described in the legend of **Extended Data Fig. 8**. The sentence now reads:

“Surprisingly, however, working memory engagement did not alter the temporal profile of the trial-averaged activity of individual cells as compared to chance (Extended Data Fig. 8; $n = 805$ cells, $p = 0.43$), nor the magnitude of cell-averaged delay period activity (Fig. 2d, f; $n = 805$ cells, $p = 0.67$).”

4. Line 225-7: “Furthermore, this analysis did not reveal any clear subpopulations of cells whose delay activity was selective to either task.”What exactly does this mean?

We were commenting on the fact that in the scatter plots in **Extended Data Fig. 10**, there were no clusters of cells which were clearly Task or Cue selective (i.e. off-diagonal), implying that although the distribution of average firing rates was higher than chance, this was not due to strong tuning biases of a small number of cells.

5. Line 313: “(Fig. 3e, h, k, n,)” unnecessary comma at the end.

Fixed, thank you.

6. Title EDF 7: “Stimulus rotations ensured identical sensory inputs across the two tasks”, correct to sensory.

Fixed, thank you.

7. Lines 334-5: I believe a reference to Fig. 4b and Fig. 4h should be added.

Added.

8. in Fig. 1a the reward sign only appears in the WM task but not the Discrimination task, while this contingency is simpler, it is still worth having the reward sign.

We've updated **Fig. 1a** with this change.

9. For Fig. 2g and 2i, I expected to see three different trajectories, one for each delay group, but it seems that in the initial part of the trajectories the different delays combine into one line. Is this a plotting effect and there are actually three lines there? Or were they combined artificially? It would be good to clarify in the legends.

The initial component of each of the three delay trajectories is shared across all trials, as activity from trials which were *at least* as long as a given time point were plotted (i.e. all trials were at least 800 ms long and so were included in the initial points of all three trajectories). Accordingly, each of the three trial groups' trajectories diverge only when they are no longer in the previous trial group's delay range (e.g. the 2.4 – 3.2 second delay group shared the delay trajectory up to 2.4 seconds, and split off after that). We have updated the figure legend to clarify this.

“Trials were split by task and into three groups of stimulus onset times (0.8-1.6 s, 1.6-2.4 s, and 2.4-3.2 s), such that at each point in time, data from trials with delays which were at least as long as their respective stimulus onset time group were plotted (i.e. trajectories were shared among stimulus onset time groups until 1.6 or 2.4 seconds following delay onset).”

10. Line 352: in “...following area AM inactivation (Fig. 4k; ...)”. Should “Fig. 4k” be “Fig. 4e”?

Yes, fixed, thank you.

11. Line 976: the word “first” is in italics. This seems unnecessary.

Agreed, changed.

12. I found the vertical axes in Extended Data Fig. 10 g and h and their description confusing. Please clarify.

There was a mistake in the labelling of these axes, sorry. These plots represent the proportion of total variance explained by the individual PCs of the population activity (i.e. the PCs which were used for decoding in the preceding panels). We have updated the axes labels.

Reviewer Reports on the First Revision:

Referees' comments:

Referee #1 (Remarks to the Author):

The authors have done an excellent job revising the manuscript, adding new analyses that clarify the nature of working memory representations and their relationship to behavior. All my comments are satisfactorily addressed. This study provides compelling evidence for distributed high-dimensional representation of working memory (WM) that is maintained across reciprocally coupled cortical regions in mice. The findings will have a substantial impact in the field.

Typo: Fig 3b legend (line 261) "Extended Data Fig. 10" should be updated to EDF11. Same for Methods line 1064.

Nuo Li

Referee #2 (Remarks to the Author):

The authors put together a serious and thorough response to the comments. The paper is acceptable for publication in my opinion.